# Automatic Auxiliary Task Selection and Adaptive Weighting Boost Molecular Property Prediction

**Zhiqiang Zhong**[1,2,3]     **Davide Mottin**[1]

[1]Department of Computer Science, Aarhus University
[2]Institute for Advanced Studies, University of Luxembourg
[3]Faculty of Science, Technology and Medicine, University of Luxembourg
**Contacts:** zhiqiang.zhong@uni.lu,   davide@cs.au.dk

## Abstract

Recent studies in *Machine Learning* (ML) for biological research focus on investigating molecular properties to accelerate drug discovery. However, limited labeled molecular data often hampers the performance of ML models. A common strategy to mitigate data scarcity is leveraging auxiliary learning tasks to provide additional supervision, but selecting effective auxiliary tasks requires substantial domain expertise and manual effort, and their inclusion does not always guarantee performance gains. To overcome these challenges, we introduce *Automatic Auxiliary Task Selection* (AUTAUT), a fully automated framework that seamlessly retrieves auxiliary tasks using large language models and adaptively integrates them through a novel *gradient alignment* weighting mechanism. By automatically emphasizing auxiliary tasks aligned with the primary objective, AUTAUT significantly enhances predictive accuracy while reducing negative impacts from irrelevant tasks. Extensive evaluations demonstrate that AUTAUT outperforms *10* auxiliary task-based approaches and *18* advanced molecular property prediction models.

## 1 Introduction

*Machine Learning* (ML) continues to drive advancements across diverse scientific disciplines, with biology standing out as a key beneficiary [28, 2, 11, 63]. Despite these advances, the limited availability of annotations in molecule datasets constrains the performance of ML models for molecular property prediction [50, 61, 26]. This limitation underscores the importance of strategies that enhance learning efficiency with minimal annotated data, *e.g.*, semi-supervised learning [66, 12], transfer learning [49, 7], and few-shot learning [10, 46].

Among these strategies, a common approach is to introduce auxiliary learning tasks, such as predicting molecular solubility to help estimate toxicity levels, so as to provide additional supervision from related tasks with easily obtainable labels that improve primary molecular prediction tasks [41, 15]. By incorporating chemical, physical, structural, and toxicological profiles, these tasks help ML models represent underlying data structures more effectively and enhance generalisation [44, 57]. This approach has shown benefits in molecular property prediction, where the objective is to determine key characteristics *e.g.*, bioactivity, and organ-specific drug effects [32, 51].

However, the process of constructing high-quality auxiliary tasks is complex and resource-intensive. These tasks are typically derived from existing datasets [52, 20] or manually designed by domain experts to align with the primary dataset [27, 13]. In disciplines such as biology and chemistry, these challenges are particularly pronounced due to the scarcity, cost, and time-intensive nature of acquiring domain-specific knowledge [63]. As datasets grow in scale and complexity, manual evaluation of task relevance becomes impractical, creating scalability bottlenecks [67]. Furthermore, effectively incorporating auxiliary tasks into training frameworks remains an open problem [42, 27]. A common

39th Conference on Neural Information Processing Systems (NeurIPS 2025).

Table 1: Model comparison of AUTAUT and related work. A model is self-contained if it does not require additional datasets.

| MODEL | AUTOMATIC TASK RETRIEVAL | AUTOMATIC TASK SELECTION | ADAPTIVE WEIGHTING | SELF-CONTAINED | NECESSARY INPUTS |
|---|:---:|:---:|:---:|:---:|:---:|
| UNWEIGHTED AVERAGES | ✗ | ✗ | ✗ | ✓ | AUXILIARY TASKS |
| TAG [8] | ✗ | ✗ | ✓ | ✓ | AUXILIARY TASKS |
| TASK2VEC [1] | ✗ | ✗ | ✓ | ✓ | AUXILIARY TASKS & DESCRIPTIONS |
| MTDNN [25] | ✗ | ✗ | ✗ | ✗ | RELEVANT AUXILIARY DATASETS & DESCRIPTIONS |
| GRADNORM [3] | ✗ | ✗ | ✓ | ✓ | AUXILIARY TASKS |
| GS-META [68] | ✗ | ✓ | ✗ | ✗ | AUXILIARY TASKS & RELATION GRAPH |
| MOLGROUP [17] | ✗ | ✗ | ✓ | ✗ | RELEVANT AUXILIARY DATASETS & DESCRIPTIONS |
| INSTRUCTMOL [51] | ✗ | ✗ | ✓ | ✗ | RELEVANT AUXILIARY DATASETS & DESCRIPTIONS |
| AUTAUT (OURS) | ✓ | ✓ | ✓ | ✓ | NONE |

approach is joint training, where the model is trained simultaneously on both primary and auxiliary task labels to leverage additional supervision. The challenge lies in balancing the contributions of auxiliary tasks such that they reinforce rather than interfere with the learning objectives of the primary task [18]. Auxiliary tasks should provide useful inductive biases without introducing noise or misguiding the model [29, 17]. Poorly chosen or improperly integrated auxiliary tasks can dilute learning signals, exacerbate overfitting, or degrade the primary task's performance.

In this paper, we introduce *Automatic Auxiliary Task Selection* (AUTAUT), a novel framework that automates the retrieval, selection, and adaptive integration of auxiliary task labels to enhance molecular property prediction. AUTAUT leverages the capabilities of *Large Language Models* (LLMs) to identify relevant auxiliary task labels and employs an adaptive weighting strategy to dynamically adjust their contributions to the primary task. Specifically, AUTAUT consists of three main components: *(1) Auxiliary Task Retrieval:* AUTAUT employs the information retrieval capabilities of LLMs to collect candidate auxiliary task labels from domain-specific databases or online resources. *(2) Auxiliary Task Selection:* Using a multi-step prompting mechanism [9], AUTAUT investigates the relevance of each candidate auxiliary task label to the primary task, ensuring that only the most relevant tasks are selected. *(3) Joint Training Strategy:* The selected auxiliary task labels are incorporated into the training pipeline through a dynamic weighting strategy. This approach adjusts the contribution of each auxiliary task during training, ensuring that they provide proper supervision without overshadowing the primary task's objectives. Importantly, by automating the process of auxiliary task selection, AUTAUT eliminates the need for manual intervention and domain expertise, making it particularly suitable for molecular property prediction tasks with limited annotations.

We evaluate AUTAUT on *9* molecular property prediction datasets, demonstrating its superiority over *10* auxiliary task-based methods and *18* state-of-the-art property prediction models. Ablation studies further validate the effectiveness of AUTAUT's task selection and adaptive reweighting in improving robustness, while also analyzing the impact of hyperparameters and different LLM choices.

## 2 Related Work

**ML for Molecular Property Prediction.** ML models have become integral to molecular property prediction, leveraging their capacity to encode structural information and capture complex relationships in molecular data [24, 11]. State-of-the-art methods include graph neural networks [53, 62] and transformers [45, 39], which utilize molecular SMILES strings, structures, and other molecular descriptors. Despite their success, these models often depend on large, annotated datasets, limiting their applicability in data-scarce domains [63, 26]. To overcome this limitation, approaches like semi-supervised learning [66, 12], and transfer learning [49, 7] have been proposed.

**Additional Supervision from Auxiliary Tasks.** Auxiliary tasks have proven to be an effective strategy for enhancing the performance of the primary task by leveraging related, easier-to-obtain labels [52, 27]. In the context of molecular property prediction, auxiliary tasks have included predicting molecular fingerprints, physicochemical properties, and toxicity profiles [13, 44, 36]. These tasks enrich representation learning and improve generalization, particularly in data-scarce scenarios [8, 40, 57]. However, current approaches often depend on manually curated auxiliary tasks informed by domain knowledge [27, 63, 5]. This manual curation introduces biases and limits scalability, posing challenges when adapting to larger datasets or emerging domains where expert annotations are unavailable [20, 67].

**Integration of Auxiliary Tasks.** Furthermore, integrating auxiliary tasks into training frameworks remains a significant challenge. Balancing their contributions and avoiding interference with the primary task is complex [42, 18]. For example, Lyle *et al.*[35] investigate the effect of auxiliary tasks on learned representations, and Liu *et al.*[29] address the issue of conflicting gradients between auxiliary tasks, which can hinder convergence. Poorly chosen auxiliary tasks can lead to adverse outcomes such as overfitting or optimization divergence [17].

**Novel Capabilities of LLMs.** The advent of LLMs has unlocked new possibilities for ML across various domains, including molecular property prediction [64, 38]. LLMs, with their vast pre-trained knowledge and advanced reasoning capabilities, have been leveraged to augment feature extraction, data annotation, and knowledge integration [4, 19]. Their ability to adapt across diverse contexts makes them particularly appealing for applications where traditional ML approaches struggle with data sparsity or require extensive domain expertise [21].

**Discussion.** However, leveraging LLMs for auxiliary task retrieval is not only underexplored but also inherently challenging. Identifying relevant auxiliary tasks requires reasoning over complex domain-specific relationships, integrating structured and unstructured knowledge, and ensuring that selected tasks contribute meaningfully to the primary learning objective. The non-trivial nature of this problem stems from the need to balance task relevance, diversity, and generalization while avoiding spurious correlations. Addressing these challenges is a key focus of our work.

Table 1 summarizes the key advantages of the proposed AUTAUT, comparing it with several state-of-the-art methods. Notably, AUTAUT *is the only fully automated framework that eliminates the need for human intervention in designing or selecting auxiliary tasks and curating additional datasets. It is entirely self-contained, relying solely on the LLM without requiring any external data.* A more detailed discussion is provided in Appendix E due to space constraints..

## 3 Proposed Framework: AUTAUT

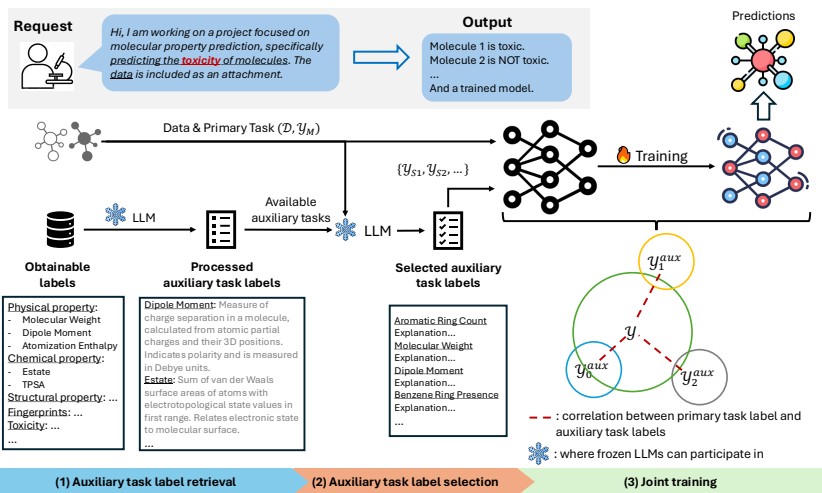

Figure 1: **Overview of the AUTAUT framework.** There are three main steps: (*i*) *Auxiliary task retrieval:* AUTAUT leverages the information retrieval capabilities of LLMs to retrieve a diverse set of obtainable auxiliary tasks from online resources or domain-specific databases. (*ii*) *Auxiliary task selection*: AUTAUT guides LLMs to evaluate the relevance of each auxiliary task to the primary task and select the most relevant ones. (*iii*) *Joint training:* AUTAUT employs a dynamic weighting strategy to adjust the contribution of each auxiliary task during training, ensuring that they provide meaningful supervision without diluting the primary task signal.

## 3.1 Preliminary

Let $\mathcal{D}$ denote the primary dataset, partitioned into training ($\mathcal{D}_{\text{TRAIN}}$), validation ($\mathcal{D}_{\text{VALID}}$), and test ($\mathcal{D}_{\text{TEST}}$) subsets. Each molecule in $\mathcal{D}$ is represented as $\mathcal{M} = (X, Y)$, where $X$ denotes the molecular representation, such as a SELFIES string, or graph structure, and $Y$ corresponds to its associated property label, such as water solubility or toxicity.

The primary task, $\mathcal{T}_{\text{M}}$, aims to learn a mapping function $f_\theta : \mathcal{X} \to \mathcal{Y}_{\text{M}}$ that predicts molecular properties using the training dataset $\mathcal{D}_{\text{TRAIN}}$. The set of target molecular properties is denoted as $\mathcal{Y}_{\text{M}} = \{Y_{\text{M}_1}, Y_{\text{M}_2}, \dots\}$. Beyond the primary task, we consider a set of auxiliary tasks, $\mathcal{T}_{\text{A}}$, where each task is defined by an auxiliary label $Y_{\text{A}_i}$ associated with the same representations as in $\mathcal{D}_{\text{TRAIN}}$. The corresponding training datasets, denoted as $\mathcal{D}^i_{\text{TRAIN}}$, share molecular representations with $\mathcal{D}_{\text{TRAIN}}$ but have different property labels. The objective is to optimize the model parameters $\theta$ to improve primary task performance by leveraging both primary task labels $\mathcal{Y}_{\text{M}}$ and auxiliary task labels $\mathcal{Y}_{\text{A}}$.

However, selecting useful auxiliary tasks is non-trivial. Not all auxiliary labels contribute positively to the primary task, and in practical scenarios, predefined auxiliary task labels $\mathcal{Y}_{\text{A}}$ may not be available. Thus, we tackle two challenges. *The first challenge is to automatically identify and select beneficial auxiliary task labels $\mathcal{Y}_{\text{S}} \subseteq \mathcal{Y}_{\text{A}}$. The second challenge is to develop an effective joint training strategy* that integrates the selected auxiliary tasks into the primary learning process.

## 3.2 Overview of the Framework

Our proposed framework, AUTAUT, is illustrated in Figure 1. Given a user request specifying the primary task (*e.g.*, predicting molecular toxicity) and the primary dataset, which includes the training dataset with labels, AUTAUT operates in three stages: *First*, AUTAUT automatically retrieves potential auxiliary task labels from online resources or domain-specific databases. *Second*, AUTAUT generates a brief summary describing the correlation between auxiliary task labels and the primary task and selects a subset of auxiliary task labels based on the generated summary. *Finally*, it applies a joint training strategy to integrate the selected auxiliary task labels into the primary task training process, improving predictive performance. By following these steps, AUTAUT enables molecular toxicity prediction with auxiliary task enhancement, without requiring manual efforts.

## 3.3 Automated Auxiliary Task Retrieval and Selection

This section outlines the methodology for designing prompts that enable LLMs to perform automated auxiliary task label retrieval and selection. We emphasize that *prompt engineering is not the primary focus of this work, and the prompts used here are intentionally simple and minimally refined*. See Appendix A for our complete prompts and obtained outputs.

**Auxiliary Task Retrieval.** Given a molecular dataset $\mathcal{D}$ and the primary task labels $\mathcal{Y}_{\text{M}}$, a prompt is designed to instruct LLMs to retrieve potential auxiliary task labels $\mathcal{Y}_{\text{A}}$. The objective is to construct a query $\mathcal{Q}$ that enables the LLM ($f_{\text{LLM}}$) to generate a structured response $\mathcal{A}$, where $\mathcal{A} = f_{\text{LLM}}(\mathcal{Q})$. The response $\mathcal{A}$ represents a set of candidate auxiliary task labels $\mathcal{Y}_{\text{A}}$ relevant to the primary molecular property prediction task.

The prompt consists of two main components: (1) *Instruction:* Provides general guidance to the LLM, specifying its role in the retrieval process (*e.g.*, expertise in chemistry, finance, biology, *etc.*). (2) *Message:* A direct and clear request for the LLM to identify potential auxiliary task labels based on the given context (*e.g.*, molecular properties or computational tools). This approach yields a computable auxiliary task label set $\mathcal{Y}_{\text{A}}$. An example prompt and a partial response from the LLM are provided in Appendix A.1, demonstrating that even basic prompts can generate meaningful and relevant auxiliary tasks. We provide the details of collected molecular properties in Table 4.

**Auxiliary Task Selection.** After retrieving the potential auxiliary task label set $\mathcal{Y}_{\text{A}}$, the next step evaluates the relevance of each auxiliary task label to the primary task labels $\mathcal{Y}_{\text{M}}$. To achieve this, AUTAUT employs a multi-step prompting mechanism [48, 59, 9] that refines the LLM's responses and summarizes the correlations to quantify the alignment between the objectives of the auxiliary and primary task labels.

The multi-step prompting mechanism proceeds as follows: *(S1)* The LLM retrieves relevant information about the auxiliary task labels $\mathcal{Y}_{\text{A}}$ from online resources or domain-specific databases, ensuring that essential domain-specific details are gathered. *(S2)* The LLM processes the retrieved information and generates a brief summary of each auxiliary task label. These summaries incorporate domain-specific knowledge or information from the previous step. *(S3)* The LLM selects the top $K$ auxiliary task labels $\mathcal{Y}_{\text{S}} \subseteq \mathcal{Y}_{\text{A}}$ that are most relevant to the primary task labels $\mathcal{Y}_{\text{M}}$, ensuring that only the most useful auxiliary tasks are integrated into training. An example prompt and a partial response from the LLM are presented in Appendix A.2. To improve the consistency of the LLM's behavior during auxiliary task selection, we set the LLM's hyperparameter temperature to 0.2. The final set of selected auxiliary tasks is shown in Table 5.

The selected auxiliary task labels $\mathcal{Y}_{\text{S}}$ form the foundation for enhancing primary task performance in the subsequent joint training phase. By automating this process, AUTAUT identifies and selects auxiliary tasks without requiring manual curation or extensive domain-specific expertise.

## 3.4 Learning to Learn from Selected Auxiliary Tasks

The primary task aims to learn a function $f_\theta$ with parameters $\theta$ that predicts molecular properties $\mathcal{Y}_{\text{M}}$ from molecular representations. At its core, this is an optimization problem where the objective is to maximize the likelihood of the primary task:

$$\mathcal{L}_{\text{M}}(\theta) = -\log p(\mathcal{D}_{\text{TRAIN}}|\theta) - \log p(\theta), \tag{1}$$

where $p(\mathcal{D}_{\text{TRAIN}}|\theta)$ represents the likelihood of the training data, and $p(\theta)$ is prior knowledge over model parameters.

Besides, selected auxiliary task labels $\mathcal{Y}_S$ provide additional signals by acting as informative constraints on the optimization landscape. Given an auxiliary task $k$, its corresponding objective is:

$$\mathcal{L}_{\text{S}}^k(\theta) = -\log p(\mathcal{D}_{\text{TRAIN}}^k|\theta) - \log p(\theta). \tag{2}$$

By incorporating auxiliary tasks, the overall training objective is redefined as:

$$\mathcal{L}(\theta, \boldsymbol{\alpha}) = \mathcal{L}_{\text{M}}(\theta) + \sum_{k=1}^{K} \alpha_k \mathcal{L}_{\text{S}}^k(\theta), \tag{3}$$

where $\alpha_k$ represents the contribution of auxiliary task $k$, and $\boldsymbol{\alpha}$ are trainable parameters that dynamically adjust during training. Since auxiliary tasks vary in their usefulness, their weights should reflect their contribution to improving the primary task. A well-chosen auxiliary task should exhibit gradient alignment with the primary task. Auxiliary tasks with higher alignment are assigned greater weight during training, allowing the model to focus on tasks that reinforce relevant learning signals.

**Theorem 1** (Subspace Alignment for Optimization). *If the gradient of the primary task loss $\nabla \mathcal{L}_{\text{M}}(\theta)$ lies entirely within the subspace $\mathcal{S}$ spanned by auxiliary task gradients, then the auxiliary tasks fully support primary task optimization:*

$$\left\| \nabla \mathcal{L}_{\text{M}}(\theta) - \sum_{k=1}^{K} \alpha_k \nabla \mathcal{L}_{\text{S}}^k(\theta) \right\| = 0. \tag{4}$$

This result ensures that when auxiliary tasks are properly weighted, they can serve as effective surrogates for the primary task and contribute directly to its optimization (proof in Appendix B.1).

*Optimizing auxiliary task weights.* To dynamically update the auxiliary weights $\boldsymbol{\alpha}$, we minimize the Fisher divergence between the primary task gradient and the surrogate prior induced by the auxiliary tasks:

$$\min_{\boldsymbol{\alpha}} \mathbb{E}_{\theta \sim p^J} \| \nabla \log p(\mathcal{T}_{\text{M}}|\theta) - \nabla \log p_{\boldsymbol{\alpha}}(\theta) \|_2^2. \tag{5}$$

This optimization ensures that the auxiliary task contributions are continuously refined based on their alignment with the primary task.

**Theorem 2** (Generalization Bounds with Auxiliary Tasks). *Incorporating well-aligned auxiliary tasks reduces the hypothesis class complexity, which we formalize via the Rademacher complexity $\mathcal{R}_n(\mathcal{H})$. This leads to improved generalization:*

$$\mathcal{E}(\theta) \leq \hat{\mathcal{E}}(\theta) + c \cdot \mathcal{R}_n(\mathcal{H}_{\boldsymbol{\alpha}}), \tag{6}$$

*where $\hat{\mathcal{E}}(\theta)$ is the empirical error, $\mathcal{H}_{\boldsymbol{\alpha}}$ is the hypothesis class augmented with auxiliary tasks, and $c$ is a constant.*

This bound highlights that by adjusting auxiliary task weights effectively, we constrain the hypothesis space, leading to better generalization and improved predictive performance (proof in Appendix B.2).

*Training procedure.* The training process consists of three phases. First, in the *weight initialization* phase, the auxiliary task weights $\boldsymbol{\alpha}$ are initialized based on affinity scores derived from gradient alignment. The model is then trained on $\mathcal{D}_{\mathrm{TRAIN}}$ to establish a baseline. Next, during *dynamic weight adaptation*, the auxiliary task weights are iteratively updated using gradient alignment and validation performance on $\mathcal{D}_{\mathrm{VALID}}$. The update follows the rule:

$$\alpha_k^{(t+1)} = \alpha_k^{(t)} - \beta \nabla_{\alpha_k} \|\nabla \log p(\mathcal{T}_{\mathrm{M}}|\theta) - \nabla \log p_{\boldsymbol{\alpha}}(\theta)\|_2^2, \tag{7}$$

where $\beta$ is the learning rate for task weights. Finally, in the *model fine-tuning* phase, after the task weights stabilize, the model parameters $\theta$ are optimized with fixed auxiliary task weights. The final model is evaluated on $\mathcal{D}_{\mathrm{TEST}}$ to assess improvements in molecular property prediction. Algorithm 1 provides an overview of the AUTAUT's complete pipeline. Further implementation details are available in Appendix C. Appendix D provides a computation complexity analysis and some training information. Overall, AUTAUT achieves a balance between improved learning efficiency and computational cost, making it a scalable approach for molecular property prediction tasks.

## 4 Experiments

### 4.1 Datasets

This paper selects various datasets from a widely used benchmark, MoleculeNet [52], to examine the effectiveness of our algorithm for molecular property prediction. We include 6 representative datasets for the property classification tasks, BBBP, BACE, CLINTOX, TOX21, TOXCAST, and SIDER, and 3 for the property regression tasks, ESOL, FREESOLV, and LIPO. Note that these datasets do not have features duplicated with the auxiliary tasks collected in Table 4. We follow the previous work to adopt scaffold splitting [17] to divide the datasets into training, validation, and test sets, where molecules are partitioned based on their core scaffolds to ensure structurally dissimilar compounds appear in different splits. Appendix F presents additional details about the datasets and splits.

### 4.2 Competing Models

We compare AUTAUT with 10 auxiliary tasks selection approaches, using 3 base ML models (including GCN [23], GIN [55] and GRAPHORMER [60]). The 10 auxiliary task selection competing methods including beam search method [37], grouping-based methods and methods (including TAG [8], TASK2VEC [1], and MOLGROUP [17]) follow the pre-training and fine-tuning pipeline. Moreover, we select Unweighted Averages(UA), GRADNORM [3], MTDNN [25], GS-META [68], and INSTRUCTMOL [51]. Furthermore, we present the results of the Pretrain-Finetune (PF) strategy [15], where the model is first trained on PCQM4Mv2 [14] and then finetuned on the downstream dataset. In addition, we compare AUTAUT with 18 advanced molecular property prediction models to demonstrate its effectiveness and generalization. Such as GCN, GIN, GRAPHORMER, D-MPNN [58], ATTENTIONFP [54], MGCN [34], N-GRAM [31], PRETRAINGNN [15], GPT-GNN [16], GROVER$_{\mathrm{BASE}}$ [39], GROVER$_{\mathrm{LARGE}}$, 3D-INFOMAX [43], GRAPHMVP [33], MOL-CLR [47], UNI-MOL [65] GEM [6], PIN-TUNING [30] and LAC [56]. Details about these competing methods and other experimental settings can be found in Appendix E- F. Our code and data are available at `https://github.com/zhiqiangzhongddu/AUTAUT`.

### 4.3 Main Results

**AUTAUT surpasses auxiliary task selection methods.** Table 2 compares AUTAUT with 10 auxiliary task selection methods across 9 molecular property prediction tasks and three ML models. The results consistently highlight AUTAUT's superior performance, showcasing its ability to effectively select and integrate auxiliary tasks to enhance downstream predictions.

AUTAUT achieves the *highest performance on classification tasks*, delivering a $6.4\%$ improvement on BBBP with GCN and a $5.1\%$ improvement on BACE with GRAPHORMER. These significant

Table 2: Performance of ML models with different auxiliary task selection methods on molecular prediction tasks. For classification tasks, we calculate the ROC-AUC, while for regression tasks, we use RMSE as the evaluation metric. The number in the bracket is the standard deviation of 5 runs.

| | CLASSIFICATION (ROC-AUC % ↑) | | | | | | REGRESSION (RMSE ↓) | | |
|---|---|---|---|---|---|---|---|---|---|
| DATASETS
# MOLUCULES
# TASKS | BBBP
2,039
1 | BACE
1,513
1 | CLINTOX
1,478
2 | TOX21
7,831
12 | TOXCAST
8,575
617 | SIDER
1,427
27 | ESOL
1,128
1 | FREESOLV
642
1 | LIPO
4,200
1 |
| **GCN** | $63.45_{\pm0.05}$ | $74.83_{\pm0.18}$ | $56.28_{\pm0.09}$ | $74.63_{\pm0.06}$ | $65.38_{\pm0.30}$ | $62.24_{\pm0.27}$ | $3.165_{\pm0.007}$ | $3.752_{\pm0.013}$ | $1.672_{\pm0.008}$ |
| +BEAM SEARCH | $66.08_{\pm0.02}$ | $68.42_{\pm0.03}$ | $58.32_{\pm0.08}$ | $75.30_{\pm0.01}$ | $66.45_{\pm0.22}$ | $62.87_{\pm0.25}$ | $3.205_{\pm0.020}$ | $3.415_{\pm0.018}$ | $1.685_{\pm0.012}$ |
| +TAG | $64.65_{\pm0.02}$ | $72.10_{\pm0.02}$ | $57.12_{\pm0.06}$ | $71.78_{\pm0.01}$ | $65.08_{\pm0.21}$ | $62.63_{\pm0.23}$ | $3.340_{\pm0.035}$ | $3.885_{\pm0.055}$ | $1.722_{\pm0.014}$ |
| +TASK2VEC | $68.25_{\pm0.01}$ | $75.38_{\pm0.03}$ | $49.55_{\pm0.07}$ | $70.20_{\pm0.01}$ | $63.62_{\pm0.20}$ | $62.25_{\pm0.23}$ | $3.278_{\pm0.042}$ | $3.437_{\pm0.030}$ | $1.742_{\pm0.016}$ |
| +MTDNN | $66.63_{\pm0.02}$ | $70.12_{\pm0.03}$ | $53.25_{\pm0.08}$ | $71.95_{\pm0.01}$ | $64.53_{\pm0.25}$ | $61.95_{\pm0.26}$ | $3.172_{\pm0.055}$ | $3.420_{\pm0.040}$ | $1.715_{\pm0.015}$ |
| +UA | $60.38_{\pm0.01}$ | $62.48_{\pm0.03}$ | $51.95_{\pm0.08}$ | $70.95_{\pm0.01}$ | $62.75_{\pm0.24}$ | $60.18_{\pm0.22}$ | $3.428_{\pm0.072}$ | $4.210_{\pm0.060}$ | $1.810_{\pm0.017}$ |
| +GRADNORM | $61.55_{\pm0.01}$ | $65.12_{\pm0.04}$ | $53.45_{\pm0.07}$ | $59.88_{\pm0.05}$ | $62.05_{\pm0.23}$ | $59.20_{\pm0.22}$ | $3.558_{\pm0.082}$ | $4.410_{\pm0.065}$ | $1.855_{\pm0.018}$ |
| +PF | $57.12_{\pm0.03}$ | $65.22_{\pm0.05}$ | $56.45_{\pm0.04}$ | $51.12_{\pm0.02}$ | $60.88_{\pm0.21}$ | $58.62_{\pm0.20}$ | $3.658_{\pm0.075}$ | $4.508_{\pm0.072}$ | $1.902_{\pm0.021}$ |
| +MOLGROUP | $68.42_{\pm0.02}$ | $77.68_{\pm0.02}$ | $60.18_{\pm0.03}$ | $76.12_{\pm0.01}$ | $66.95_{\pm0.23}$ | $63.12_{\pm0.25}$ | $3.125_{\pm0.025}$ | $3.245_{\pm0.020}$ | $1.702_{\pm0.012}$ |
| +GS-META | $66.72_{\pm0.02}$ | $75.71_{\pm0.41}$ | $63.23_{\pm0.06}$ | $75.23_{\pm0.08}$ | $65.94_{\pm0.21}$ | $63.22_{\pm0.60}$ | $3.035_{\pm0.013}$ | $3.444_{\pm0.061}$ | $1.839_{\pm0.027}$ |
| +INSTRUCTMOL | $68.50_{\pm0.04}$ | $80.21_{\pm0.18}$ | $64.02_{\pm0.05}$ | $76.69_{\pm0.04}$ | $66.94_{\pm0.48}$ | $64.30_{\pm0.72}$ | $2.612_{\pm0.034}$ | $2.108_{\pm0.046}$ | $1.252_{\pm0.018}$ |
| +AUTAUT | $\mathbf{69.72}_{\pm0.02}$ | $\mathbf{83.10}_{\pm0.02}$ | $\mathbf{81.00}_{\pm0.03}$ | $\mathbf{77.55}_{\pm0.01}$ | $\mathbf{68.05}_{\pm0.19}$ | $\mathbf{64.62}_{\pm0.24}$ | $\mathbf{1.885}_{\pm0.028}$ | $\mathbf{1.975}_{\pm0.022}$ | $\mathbf{1.285}_{\pm0.010}$ |
| **GIN** | $66.82_{\pm0.09}$ | $77.45_{\pm0.21}$ | $56.48_{\pm0.18}$ | $75.32_{\pm0.17}$ | $62.35_{\pm0.05}$ | $60.25_{\pm0.20}$ | $2.895_{\pm0.012}$ | $3.865_{\pm0.018}$ | $1.692_{\pm0.011}$ |
| +BEAM SEARCH | $67.78_{\pm0.02}$ | $81.92_{\pm0.04}$ | $76.95_{\pm0.12}$ | $77.05_{\pm0.02}$ | $65.28_{\pm0.22}$ | $62.57_{\pm0.22}$ | $2.943_{\pm0.028}$ | $3.375_{\pm0.022}$ | $1.688_{\pm0.014}$ |
| +TAG | $61.25_{\pm0.02}$ | $72.15_{\pm0.03}$ | $58.25_{\pm0.06}$ | $72.08_{\pm0.01}$ | $64.38_{\pm0.21}$ | $61.75_{\pm0.21}$ | $3.275_{\pm0.060}$ | $4.002_{\pm0.048}$ | $1.752_{\pm0.015}$ |
| +TASK2VEC | $68.48_{\pm0.01}$ | $75.10_{\pm0.03}$ | $48.62_{\pm0.04}$ | $69.12_{\pm0.01}$ | $63.62_{\pm0.24}$ | $60.38_{\pm0.26}$ | $3.232_{\pm0.045}$ | $3.452_{\pm0.032}$ | $1.775_{\pm0.017}$ |
| +MTDNN | $66.92_{\pm0.03}$ | $70.35_{\pm0.03}$ | $53.28_{\pm0.07}$ | $72.32_{\pm0.01}$ | $64.58_{\pm0.25}$ | $61.92_{\pm0.26}$ | $3.182_{\pm0.052}$ | $3.468_{\pm0.040}$ | $1.722_{\pm0.014}$ |
| +UA | $60.75_{\pm0.01}$ | $62.18_{\pm0.03}$ | $52.18_{\pm0.08}$ | $71.22_{\pm0.01}$ | $62.58_{\pm0.23}$ | $59.82_{\pm0.22}$ | $3.442_{\pm0.065}$ | $4.315_{\pm0.053}$ | $1.822_{\pm0.019}$ |
| +GRADNORM | $61.68_{\pm0.02}$ | $65.38_{\pm0.04}$ | $53.58_{\pm0.08}$ | $60.25_{\pm0.05}$ | $62.08_{\pm0.23}$ | $58.55_{\pm0.23}$ | $3.602_{\pm0.080}$ | $4.508_{\pm0.062}$ | $1.878_{\pm0.020}$ |
| +PF | $57.38_{\pm0.03}$ | $65.58_{\pm0.05}$ | $56.38_{\pm0.04}$ | $51.38_{\pm0.02}$ | $60.92_{\pm0.22}$ | $58.38_{\pm0.21}$ | $3.695_{\pm0.078}$ | $4.608_{\pm0.070}$ | $1.932_{\pm0.021}$ |
| +MOLGROUP | $69.88_{\pm0.01}$ | $83.12_{\pm0.02}$ | $81.25_{\pm0.04}$ | $77.72_{\pm0.01}$ | $67.55_{\pm0.22}$ | $64.28_{\pm0.24}$ | $1.882_{\pm0.030}$ | $1.952_{\pm0.025}$ | $1.288_{\pm0.011}$ |
| +GS-META | $66.46_{\pm0.09}$ | $79.46_{\pm0.00}$ | $73.54_{\pm0.12}$ | $76.27_{\pm0.08}$ | $66.14_{\pm0.31}$ | $62.96_{\pm0.25}$ | $2.060_{\pm0.012}$ | $3.848_{\pm0.043}$ | $1.434_{\pm0.041}$ |
| +INSTRUCTMOL | $69.87_{\pm0.02}$ | $81.99_{\pm0.04}$ | $76.16_{\pm0.06}$ | $77.91_{\pm0.02}$ | $68.44_{\pm0.14}$ | $64.25_{\pm0.62}$ | $1.878_{\pm0.019}$ | $1.960_{\pm0.022}$ | $1.289_{\pm0.026}$ |
| +AUTAUT | $\mathbf{70.58}_{\pm0.01}$ | $\mathbf{83.62}_{\pm0.02}$ | $\mathbf{82.52}_{\pm0.04}$ | $\mathbf{78.32}_{\pm0.01}$ | $\mathbf{69.38}_{\pm0.22}$ | $\mathbf{65.08}_{\pm0.25}$ | $\mathbf{1.870}_{\pm0.025}$ | $\mathbf{1.900}_{\pm0.020}$ | $\mathbf{1.270}_{\pm0.010}$ |
| **GRAPHORMER** | $67.05_{\pm0.04}$ | $79.18_{\pm0.13}$ | $78.42_{\pm0.12}$ | $75.55_{\pm0.18}$ | $67.12_{\pm0.05}$ | $70.08_{\pm0.21}$ | $2.102_{\pm0.012}$ | $1.795_{\pm0.010}$ | $1.282_{\pm0.010}$ |
| +BEAM SEARCH | $67.78_{\pm0.03}$ | $81.12_{\pm0.03}$ | $78.22_{\pm0.09}$ | $77.08_{\pm0.01}$ | $67.82_{\pm0.21}$ | $70.55_{\pm0.24}$ | $2.225_{\pm0.040}$ | $1.825_{\pm0.015}$ | $1.322_{\pm0.014}$ |
| +TAG | $61.52_{\pm0.02}$ | $72.25_{\pm0.04}$ | $59.35_{\pm0.07}$ | $72.55_{\pm0.01}$ | $66.25_{\pm0.23}$ | $69.25_{\pm0.25}$ | $2.285_{\pm0.060}$ | $1.865_{\pm0.020}$ | $1.375_{\pm0.018}$ |
| +TASK2VEC | $68.52_{\pm0.01}$ | $75.18_{\pm0.04}$ | $48.85_{\pm0.03}$ | $69.25_{\pm0.01}$ | $65.12_{\pm0.22}$ | $67.55_{\pm0.27}$ | $2.275_{\pm0.040}$ | $1.852_{\pm0.020}$ | $1.362_{\pm0.015}$ |
| +MTDNN | $67.08_{\pm0.02}$ | $70.52_{\pm0.03}$ | $53.52_{\pm0.07}$ | $72.45_{\pm0.01}$ | $65.55_{\pm0.25}$ | $68.12_{\pm0.26}$ | $2.202_{\pm0.042}$ | $1.875_{\pm0.018}$ | $1.335_{\pm0.012}$ |
| +UA | $60.82_{\pm0.01}$ | $62.22_{\pm0.03}$ | $52.52_{\pm0.08}$ | $71.52_{\pm0.01}$ | $63.52_{\pm0.22}$ | $66.72_{\pm0.24}$ | $2.455_{\pm0.050}$ | $2.055_{\pm0.020}$ | $1.432_{\pm0.016}$ |
| +GRADNORM | $62.02_{\pm0.02}$ | $65.52_{\pm0.04}$ | $53.82_{\pm0.07}$ | $60.52_{\pm0.05}$ | $62.82_{\pm0.23}$ | $65.52_{\pm0.25}$ | $2.605_{\pm0.070}$ | $2.205_{\pm0.040}$ | $1.525_{\pm0.020}$ |
| +PF | $57.52_{\pm0.03}$ | $65.82_{\pm0.05}$ | $56.52_{\pm0.04}$ | $51.52_{\pm0.02}$ | $61.22_{\pm0.22}$ | $64.82_{\pm0.23}$ | $2.655_{\pm0.080}$ | $2.305_{\pm0.050}$ | $1.625_{\pm0.018}$ |
| +MOLGROUP | $69.72_{\pm0.01}$ | $83.15_{\pm0.02}$ | $81.52_{\pm0.04}$ | $77.75_{\pm0.01}$ | $68.85_{\pm0.22}$ | $70.55_{\pm0.27}$ | $1.902_{\pm0.030}$ | $1.805_{\pm0.010}$ | $1.292_{\pm0.010}$ |
| +GS-META | $68.12_{\pm0.00}$ | $81.06_{\pm0.10}$ | $80.00_{\pm0.11}$ | $77.29_{\pm0.09}$ | $67.90_{\pm0.20}$ | $70.14_{\pm0.25}$ | $2.064_{\pm0.035}$ | $1.881_{\pm0.016}$ | $1.330_{\pm0.019}$ |
| +INSTRUCTMOL | $68.99_{\pm0.02}$ | $83.52_{\pm0.02}$ | $81.41_{\pm0.02}$ | $78.05_{\pm0.04}$ | $69.03_{\pm0.07}$ | $70.49_{\pm0.23}$ | $1.901_{\pm0.014}$ | $1.811_{\pm0.002}$ | $1.230_{\pm0.009}$ |
| +AUTAUT | $\mathbf{70.32}_{\pm0.01}$ | $\mathbf{83.72}_{\pm0.02}$ | $\mathbf{82.55}_{\pm0.04}$ | $\mathbf{78.52}_{\pm0.01}$ | $\mathbf{69.38}_{\pm0.21}$ | $\mathbf{71.55}_{\pm0.26}$ | $\mathbf{1.872}_{\pm0.025}$ | $\mathbf{1.772}_{\pm0.015}$ | $\mathbf{1.262}_{\pm0.010}$ |

gains highlight AUTAUT's ability to identify meaningful task relationships and leverage them effectively for prediction. For regression tasks, AUTAUT consistently achieves the lowest RMSE values across all datasets, reflecting its *strong predictive performance* and ability to generalize effectively across diverse molecular datasets. Moreover, the low standard deviations observed across five runs underscore AUTAUT's *robustness* and *reliability*. Its adaptive weighting strategy effectively balances contributions from auxiliary tasks, mitigating overfitting and ensuring consistent improvements in prediction accuracy.

**AUTAUT demonstrates superior molecular property prediction performance.** Table 3 benchmarks AUTAUT against 18 state-of-the-art molecular property prediction methods on 9 datasets. The results consistently demonstrate AUTAUT's superiority, establishing it as an effective and practical solution for molecular property prediction. AUTAUT achieves competitive performances across all datasets. These results emphasize AUTAUT's strong generalization capabilities and its ability to outperform competitive baselines. In addition, AUTAUT exhibits remarkable robustness, as evidenced by its low standard deviation across tasks, further reinforcing its reliability in practical applications. These characteristics, combined with its consistently superior performance, establish AUTAUT as an effective solution for diverse molecular property prediction tasks.

## 4.4 Analysis

**LLM-selected auxiliary tasks are chemically meaningful and stable across runs.** Table 5 in the Appendix presents the auxiliary tasks selected by the LLM across datasets. We observe that key molecular descriptors, such as LogP, TPSA, and Molecular Weight, are the tasks of choice across repeated runs and diverse prediction tasks. This indicates that LLMs can infer chemically meaningful and task-relevant patterns from textual cues, without relying on handcrafted templates or molecular priors. The selections are also sensitive to the target task. For instance, polarity-related properties are more frequently selected for solubility prediction highlighting task-specific adaptation. To reduce

Table 3: Performance on molecular property prediction tasks. For classification tasks, we calculate the ROC-AUC, while for regression tasks, we use RMSE as the evaluation metric. The number in the bracket is the standard deviation of 5 runs. ↑ indicates higher is better and ↓ indicates lower is better.

| | CLASSIFICATION (ROC-AUC % ↑) | | | | | | REGRESSION (RMSE ↓) | | |
| **DATASETS** | **BBBP** | **BACE** | **CLINTOX** | **TOX21** | **TOXCAST** | **SIDER** | **ESOL** | **FREESOLV** | **LIPO** |
|---|---|---|---|---|---|---|---|---|---|
| GCN | $63.45_{\pm0.05}$ | $74.83_{\pm0.18}$ | $56.28_{\pm0.09}$ | $74.63_{\pm0.06}$ | $65.38_{\pm0.30}$ | $62.24_{\pm0.27}$ | $3.165_{\pm0.007}$ | $3.752_{\pm0.013}$ | $1.672_{\pm0.008}$ |
| GIN | $66.82_{\pm0.09}$ | $77.45_{\pm0.21}$ | $56.48_{\pm0.18}$ | $75.32_{\pm0.17}$ | $62.35_{\pm0.05}$ | $60.25_{\pm0.20}$ | $2.895_{\pm0.012}$ | $3.865_{\pm0.018}$ | $1.692_{\pm0.011}$ |
| GRAPHORMER | $67.05_{\pm0.04}$ | $79.18_{\pm0.13}$ | $78.42_{\pm0.12}$ | $75.55_{\pm0.18}$ | $67.12_{\pm0.05}$ | $70.08_{\pm0.21}$ | $2.102_{\pm0.012}$ | $1.795_{\pm0.010}$ | $1.282_{\pm0.010}$ |
| D-MPNN | $70.83_{\pm0.42}$ | $81.19_{\pm0.51}$ | $90.95_{\pm0.63}$ | $76.24_{\pm0.47}$ | $64.97_{\pm0.24}$ | $57.31_{\pm0.45}$ | $1.045_{\pm0.070}$ | $2.080_{\pm0.090}$ | $0.685_{\pm0.015}$ |
| ATTENTIONFP | $64.15_{\pm1.48}$ | $78.24_{\pm0.31}$ | $80.51_{\pm0.26}$ | $77.00_{\pm0.35}$ | $63.52_{\pm0.34}$ | $60.12_{\pm6.03}$ | $0.880_{\pm0.025}$ | $2.075_{\pm0.150}$ | $0.720_{\pm0.002}$ |
| MGCN | $65.27_{\pm0.43}$ | $73.01_{\pm0.61}$ | $89.51_{\pm1.22}$ | $77.15_{\pm0.50}$ | $66.03_{\pm0.48}$ | $58.02_{\pm1.93}$ | $1.100_{\pm0.060}$ | $2.800_{\pm0.100}$ | $0.730_{\pm0.012}$ |
| N-GRAM | $70.15_{\pm0.52}$ | $78.09_{\pm1.37}$ | $75.95_{\pm2.97}$ | $77.21_{\pm0.45}$ | $62.04_{\pm0.60}$ | $58.94_{\pm5.50}$ | $1.150_{\pm0.045}$ | $2.850_{\pm0.110}$ | $0.735_{\pm0.013}$ |
| PRETRAINGNN | $68.91_{\pm0.98}$ | $83.97_{\pm0.64}$ | $72.95_{\pm0.61}$ | $77.45_{\pm0.38}$ | $63.01_{\pm0.87}$ | $62.03_{\pm5.94}$ | $1.100_{\pm0.050}$ | $2.750_{\pm0.010}$ | $0.740_{\pm0.004}$ |
| GPT-GNN | $63.96_{\pm1.23}$ | $71.98_{\pm0.67}$ | $65.04_{\pm1.00}$ | $77.12_{\pm0.60}$ | $62.00_{\pm0.45}$ | $58.47_{\pm4.00}$ | $1.200_{\pm0.070}$ | $3.000_{\pm0.150}$ | $0.800_{\pm0.020}$ |
| GROVER$_{BASE}$ | $65.02_{\pm0.19}$ | $81.01_{\pm0.29}$ | $74.92_{\pm0.60}$ | $77.31_{\pm0.41}$ | $62.97_{\pm0.70}$ | $61.01_{\pm5.50}$ | $1.085_{\pm0.075}$ | $2.270_{\pm0.050}$ | $0.820_{\pm0.012}$ |
| GROVER$_{LARGE}$ | $60.04_{\pm0.31}$ | $80.99_{\pm0.36}$ | $74.01_{\pm0.55}$ | $77.10_{\pm0.45}$ | $63.99_{\pm0.50}$ | $60.02_{\pm6.00}$ | $0.980_{\pm0.080}$ | $2.180_{\pm0.045}$ | $0.815_{\pm0.009}$ |
| 3D-INFOMAX | $67.02_{\pm1.23}$ | $84.01_{\pm1.31}$ | $78.03_{\pm1.43}$ | $77.25_{\pm0.32}$ | $65.96_{\pm0.80}$ | $62.04_{\pm5.90}$ | $0.950_{\pm0.030}$ | $2.600_{\pm0.080}$ | $0.800_{\pm0.016}$ |
| GRAPHMVP | $71.98_{\pm1.48}$ | $83.02_{\pm0.79}$ | $85.03_{\pm0.90}$ | $77.40_{\pm0.25}$ | $65.94_{\pm0.60}$ | $63.01_{\pm5.30}$ | $0.890_{\pm0.030}$ | $2.330_{\pm0.250}$ | $0.830_{\pm0.006}$ |
| MOLCLR | $71.51_{\pm1.96}$ | $83.48_{\pm0.52}$ | $88.97_{\pm1.10}$ | $77.35_{\pm0.34}$ | $66.98_{\pm0.70}$ | $64.05_{\pm5.70}$ | $0.940_{\pm0.120}$ | $2.580_{\pm0.220}$ | $0.650_{\pm0.007}$ |
| UNI-MOL | $72.83_{\pm0.48}$ | $\mathbf{86.52}_{\pm0.39}$ | $89.97_{\pm1.00}$ | $77.50_{\pm0.22}$ | $67.03_{\pm0.60}$ | $65.94_{\pm5.50}$ | $0.785_{\pm0.100}$ | $1.620_{\pm0.050}$ | $0.605_{\pm0.011}$ |
| GEM | $72.21_{\pm0.34}$ | $85.49_{\pm0.97}$ | $91.98_{\pm1.45}$ | $77.55_{\pm0.15}$ | $66.98_{\pm0.41}$ | $66.97_{\pm0.48}$ | $0.800_{\pm0.025}$ | $1.870_{\pm0.090}$ | $0.660_{\pm0.010}$ |
| PIN-TUNING | $70.87_{\pm0.61}$ | $81.21_{\pm0.20}$ | $89.81_{\pm1.44}$ | $77.20_{\pm0.27}$ | $66.92_{\pm0.98}$ | $66.36_{\pm0.81}$ | $0.845_{\pm0.204}$ | $1.935_{\pm0.104}$ | $0.673_{\pm0.112}$ |
| LAC | $72.44_{\pm0.95}$ | $83.47_{\pm0.11}$ | $92.11_{\pm1.19}$ | $77.66_{\pm0.61}$ | $67.01_{\pm0.34}$ | $67.02_{\pm0.65}$ | $0.794_{\pm0.100}$ | $1.827_{\pm0.121}$ | $0.657_{\pm0.108}$ |
| AUTAUT | $\mathbf{73.04}_{\pm0.58}$ | $86.03_{\pm1.14}$ | $\mathbf{92.31}_{\pm1.23}$ | $\mathbf{78.20}_{\pm0.12}$ | $\mathbf{67.54}_{\pm0.76}$ | $\mathbf{67.53}_{\pm0.82}$ | $\mathbf{0.760}_{\pm0.035}$ | $\mathbf{1.600}_{\pm0.040}$ | $\mathbf{0.580}_{\pm0.012}$ |

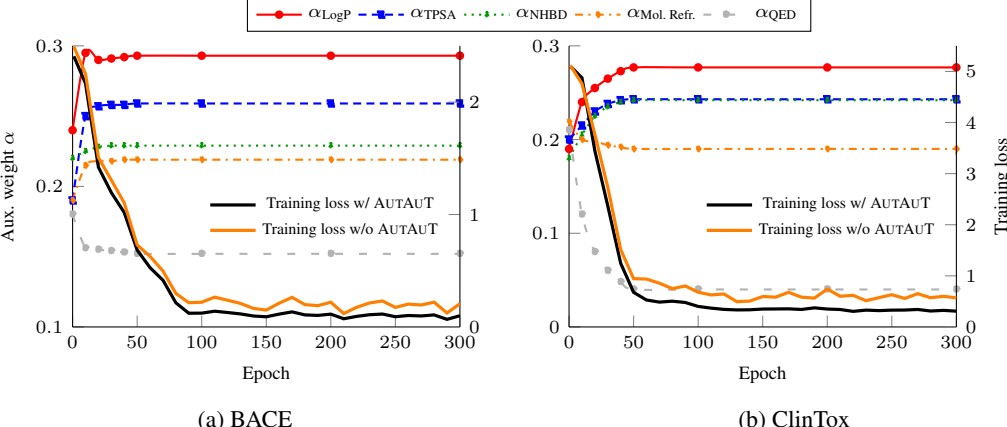

(a) BACE  (b) ClinTox

Figure 2: Auxiliary-task weights (left axes) and training-loss trajectories (right axes) for two datasets.

generation variance and improve reproducibility, we set the LLM's temperature to $0.2$, which helps the model produce consistent outputs while remaining responsive to different task contexts.

**Auxiliary task influence is correctly inferred.** Figure 2 shows the evolution of the learned auxiliary weights ($\alpha_k$) throughout the training process. AUTAUT progressively increases the contribution of auxiliary tasks that correctly align with the main prediction objective, while suppressing the influence of others. This results in more targeted supervision and faster convergence, as reflected in the consistently lower training loss compared to the baseline without adaptive integration. The adaptive weighting mechanism plays a critical role in filtering out noisy gradients and preventing negative transfer, which is particularly important when auxiliary tasks vary in relevance.

**Auxiliary task selection and adaptive weighting complement each other.** Naively introducing auxiliary tasks, especially without proper selection or weighting, can hamper model performance due to conflicting or irrelevant supervision signals. As UA draws auxiliary targets uniformly at random from Table 4 and attaches them to the backbone with equal weight. Its results in Table 2 support this observation: random auxiliary tasks significantly reduce performance, with regression tasks like FREESOLV and LIPO being particularly affected. Applying task-specific weighting (*e.g.*, GRAD-NORM) mitigates the drop by suppressing noisy gradients, but still fails to match the performance of our full method. Ablation analysis in Figure 3a confirms that both components are necessary: selection ensures the relevance of auxiliary tasks to the primary objective, while weighting adjusts their influence during training to prevent negative transfer. The strongest gain is achieved only when both mechanisms are applied together, highlighting their complementary nature.

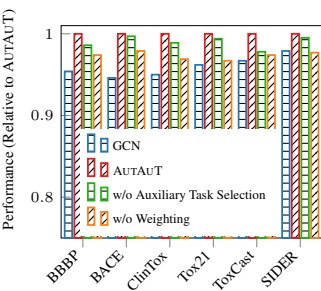
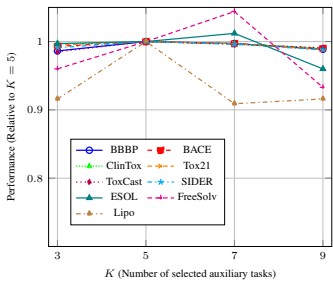
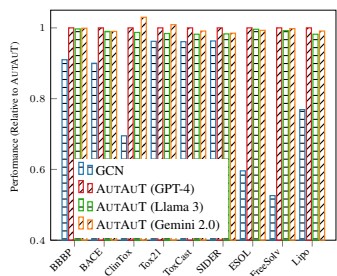

(a) Relative performance comparison of AUTAUT w/ and w/o task selection and adaptive weighting.

(b) Relative performance across datasets for $K \in \{3, 5, 7, 9\}$ on the Graphormer with AUTAUT.

(c) Relative performance comparison of AUTAUT and AUTAUT variants with different LLMs.

Figure 3: Ablation study.

**The number of selected auxiliary tasks ($K$) balance informativeness and noise.** The performance of AUTAUT depends on the number of selected auxiliary tasks, $K$. As shown in Figure 3b, using too few tasks (*e.g.*, $K = 3$) underutilizes auxiliary supervision, while larger values (*e.g.*, $K = 8$ or 10) introduce redundant or noisy signals. Yet, we find that $K = 5$ consistently yields the best results across datasets, providing a strong balance between task diversity and relevance. The sensitivity to $K$ is more pronounced in regression tasks such as FREESOLV, ESOL, and LIPO, where auxiliary signal quality is critical. In contrast, classification tasks like BBBP and BACE are more robust to variation in $K$. These results highlight that AUTAUT requires only minimal hyperparameter tuning tailored to the primary task in order to prevent underfitting or negative transfer.

**All major LLMs produce useful task suggestions and improve over baseline.** We evaluate the robustness of AUTAUT across three LLMs: GPT-4, Gemini 2.0, and LLaMA 3. As shown in Figure 3c, all variants consistently outperform the no-auxiliary-task baseline across all 9 datasets, confirming the general effectiveness of LLM-guided auxiliary supervision. GPT-4 yields the most stable performance overall and serves as our default configuration. Gemini 2.0 occasionally surpasses GPT-4, particularly on datasets like CLINTOX, TOX21, and FREESOLV, where its selected tasks, such as Eccentricity and TPSA, better align with regression or toxicity objectives. In contrast, LLaMA 3 tends to select a smaller and more conservative set of descriptors, achieving slightly lower but still competitive results. These findings show that while the choice of LLM can influence auxiliary task quality, our framework remains model-agnostic and effective across different foundation models—even in the absence of domain-specific pretraining or external chemical knowledge.

Furthermore, to isolate the effect of the LLM from that of the optimization procedure, we report the selected auxiliary tasks by different LLMs and the learned weights in Table 5. Across datasets, physicochemical descriptors such as *LogP*, *Topological Polar Surface Area*, and *Number of Hydrogen Bond Donors* are selected with both high frequency ($k$) and relatively large weights ($w$), indicating that the LLM repeatedly identifies domain-relevant features and that the optimizer further amplifies their contributions. Dataset-specific descriptors (e.g., *Balaban J Index* or *Labute Approximate Surface Area*) appear less frequently and receive smaller weights, suggesting complementary but secondary roles. The overall consistency of top-$k$ tasks across GPT-4, Llama 3, and Gemini 2.0 supports that the improvement observed in Figure 3a originates from the semantic alignment introduced by the LLM, rather than from random variation or optimization alone. These results further confirm that the LLM mainly contributes to identifying semantically relevant auxiliary tasks, while adaptive weighting refines their relative importance during training.

**Auxiliary tasks outperform static feature inclusion.** We further compare the effect of integrating selected molecular descriptors as static input features versus treating them as auxiliary prediction tasks. Table 7 in the Appendix shows that feature inclusion leads to marginal or inconsistent improvements, while modeling the same descriptors as supervised tasks with adaptive weighting yields substantial gains across all models and datasets. This highlights that auxiliary information is more effective when used as structured supervision rather than passive input, reinforcing our design choice to treat descriptors as tasks rather than features. More discussions are provided in Appendix G.

# 5 Concluding Remarks

This paper introduces AUTAUT, a fully automated framework that leverages LLMs to retrieve auxiliary tasks and integrates them via adaptive weighting to improve molecular property prediction. AUTAUT outperforms *10* auxiliary task methods and *18* advanced prediction models, with empirical analysis highlighting the complementary roles of task selection and adaptive weighting strategy. Future work will explore incorporating domain-specific knowledge into the retrieval process and extending AUTAUT to other scientific domains such as materials science.

**Broader Impact.** This work presents an automated framework for selecting and integrating auxiliary tasks in molecular property prediction using LLMs. By reducing reliance on manual feature engineering and expert-curated descriptors, AUTAUT lowers the barrier to applying auxiliary-task-based learning in data-scarce scientific domains. This may accelerate applications in drug discovery, materials design, and environmental modeling, particularly in settings where annotated data is expensive or limited. However, as AUTAUT depends on LLM-generated auxiliary task candidates, there is potential for biased or irrelevant suggestions, especially if the LLM has been exposed to skewed or incomplete chemical knowledge during pretraining. Misguided task selection could propagate into downstream predictions, especially in safety-critical domains like pharmacology. Responsible use of AUTAUT should include validation of selected tasks and awareness of LLM limitations.

**Limitations.** While AUTAUT achieves strong performance across molecular datasets, it has several limitations. First, it assumes that auxiliary tasks are computable and available in standard cheminformatics toolkits (*e.g.*, RDKit); tasks requiring experimental data or rare descriptors are excluded. Second, although the method is fully automated, it relies on LLM outputs, which may vary with prompt phrasing, temperature settings, or model version. Although we observe stability across five runs, reproducibility across future LLM updates may require prompt standardization or model pinning. Moreover, our experiments are limited to MoleculeNet-style benchmarks. The generalization of AUTAUT to other domains (*e.g.*, materials science, systems biology) remains to be validated. Lastly, the gradient alignment strategy assumes that auxiliary tasks can be meaningfully aligned with the primary task, which may not hold in more complex or noisy multi-modal settings.

## Acknowledgments

We gratefully acknowledge the EuroHPC Joint Undertaking for awarding computational GPU resources under the EuroHPC Benchmark Access and EuroHPC AI and Data-Intensive Applications Access Calls. Specifically, we thank EuroHPC for providing access to the MeluXina supercomputer hosted by LuxProvide in Luxembourg, which was essential for conducting our computational experiments. Additionally, this work is partly supported by the funding from the Institute for Advanced Studies of the University of Luxembourg, and the European Union's Horizon 2020 research and innovation program under the Marie Skłodowska-Curie Actions(GA #101081455).

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

# A  Prompts

## A.1  Auxiliary Task Retrieval Prompt

---

*Auxiliary task retrieval converzation:*

(For better visualization, complete converzation details can be found in our GitHub project.)

**Instruction:** You are now an expert in computational chemistry and machine learning for molecular property prediction. Your task is to provide detailed and accurate information about computable molecular properties.

**Message 1:** Find all available computable molecular properties.
**Response:** Computable molecular properties refer to properties of molecules that can be calculated or predicted using computational chemistry, cheminformatics, or machine learning techniques. These properties can be derived from the molecular structure and are used in various applications like drug discovery, material science, and chemical engineering. Below is a categorised list of computable molecular properties:

1. Physical and Chemical Properties
   - Molecular weight
   - LogP
   - ...
2. Geometric and Structural Properties
   - Radius of gyration
   - 3D molecular shape descriptors
   - ...
3. Topological Properties
   - Topological polar surface area (TPSA)
   - Shannon entropy
   - ...

If you are looking for tools to compute these properties, some commonly used software includes:

- RDKit (cheminformatics)
- MOPAC (semi-empirical quantum chemistry)
- ...

**Message 2:** I want to use RDKit to compute the properties of molecules. Find those computable properties using RDKit.
**Response:** RDKit provides a comprehensive set of functions to compute molecular properties. Here is a categorized list of computable properties using RDKit:

1. Physical and Chemical Properties
   - Molecular weight: rdkit.Chem.Descriptors.ExactMolWt
   - LogP: rdkit.Chem.Crippen.MolLogP
   - ...
2. Geometric and Structural Properties
   - Radius of gyration: rdkit.Chem.Descriptors3D.RadiusOfGyration
   - ...

You can compute these properties using Python by importing the relevant RDKit modules. For example: ...

---

Table 4: Details of collected molecular properties.

| Name | Category | Brief Description |
| --- | --- | --- |
| Molecular Weight | Constitutional | Total mass of a molecule calculated as the sum of atomic weights of all atoms, providing fundamental information about molecular size and mass distribution. |
| Heavy Atom Molecular Weight | Constitutional | Sum of atomic weights of all non-hydrogen atoms in the molecule, useful for comparing core molecular frameworks. |
| Number of Valence Electrons | Constitutional | Total number of electrons in the outer shells of all atoms, crucial for understanding chemical bonding and reactivity patterns. |
| Total Formal Charge | Constitutional | Sum of all formal charges on atoms in the molecule, indicating overall molecular charge state and ionic character. |
| Topological Polar Surface Area | Topological | Sum of surfaces of all polar atoms (mainly oxygen and nitrogen), correlating with drug absorption, including intestinal absorption and blood-brain barrier penetration. |
| Labute Approximate Surface Area | Topological | Approximate molecular surface area calculated using Labute's method, useful for predicting physical properties and molecular interactions. |
| Balaban J Index | Topological | Topological index based on molecular connectivity, indicating molecular branching and cyclicity. Higher values suggest more branched structures. |
| Bertz Complexity | Topological | Measure of molecular complexity considering both size and branching patterns. Higher values indicate more complex molecular structures. |
| LogP | Electronic | Logarithm of octanol-water partition coefficient, predicting molecular lipophilicity and membrane permeability. Key for drug absorption. |
| Molar Refractivity | Electronic | Measure of total polarizability of a molecule, related to molecular volume and electronic properties. Important for predicting optical behavior. |
| EState VSA1 | Electronic | Sum of van der Waals surface areas of atoms with electrotopological state values in first range. Relates electronic state to molecular surface. |
| Number of Hydrogen Bond Donors | Hydrogen bonding | Count of NH and OH groups capable of donating hydrogen bonds. Critical for predicting molecular interactions and drug-like properties. |
| NHOH Group Count | Hydrogen bonding | Total number of nitrogen and oxygen atoms with attached hydrogens, indicating potential hydrogen bond donor sites. |
| NO Group Count | Hydrogen bonding | Total count of nitrogen and oxygen atoms, regardless of hydrogen attachment. Indicates overall polar atom content. |
| Number of Aromatic Rings | Ring descriptors | Count of rings with conjugated pi-electron systems exhibiting aromaticity. Important for molecular stability and interactions. |
| Aliphatic Rings Count | Ring descriptors | Number of non-aromatic (saturated or partially saturated) rings in the molecule. Affects molecular flexibility and properties. |
| Heterocycles Count | Ring descriptors | Number of rings containing at least one heteroatom (non-carbon). Important for biological activity and drug-like properties. |
| Number of Rotatable Bonds | Molecular flexibility | Count of single bonds that can freely rotate, excluding those in rings. Key indicator of molecular flexibility and oral bioavailability. |
| Fraction of Csp3 Carbon Atoms | Molecular flexibility | Ratio of sp3 hybridized carbons to total carbon count. Higher values indicate more 3D character and better drug-like properties. |
| Molecular Volume in $Å^3$ | 3D descriptors | Three-dimensional space occupied by the molecule, calculated using van der Waals radii. Important for molecular packing and interactions. |
| Asphericity | 3D descriptors | Measure of how much molecular shape deviates from perfect sphere. Higher values indicate more elongated or irregular shapes. |
| Eccentricity | 3D descriptors | Ratio of the longest to shortest molecular axis. Indicates degree of molecular elongation in 3D space. |
| Radius of Gyration | 3D descriptors | Average distance of molecular mass from its center of mass. Describes molecular size and mass distribution in 3D. |
| Inertial Shape Factor | 3D descriptors | Ratio of principal moments of inertia, describing overall 3D shape and mass distribution. Distinguishes rod-like from disk-like molecules. |
| QED | Drug-likeness | Quantitative Estimate of Drug-likeness, combining multiple molecular properties into a single score (0-1). Higher values indicate more drug-like characteristics. |

## A.2 Auxiliary task selection Prompt

> *Auxiliary task selection prompt:*
> (For better visualization, converzation details can be found in our GitHub project. )
>
> **Instruction:** You are now an expert in computational chemistry and machine learning for molecular property prediction. Your task is to provide detailed and accurate information about computable molecular properties.
>
> **Message 1:** Search available information about these properties and write a brief summary for each property.
> **Response:** Here is a brief explanation of each property RDKit can compute:
>
> 1. Physical and Chemical Properties
>      - Molecular weight: Total mass of a molecule calculated as the sum of atomic weights of all atoms, providing fundamental information about molecular size and mass distribution.
>      - LogP: Logarithm of octanol-water partition coefficient, predicting molecular lipophilicity and membrane permeability. Key for drug absorption.
>      - ...
> 2. Geometric and Structural Properties
>      - Radius of gyration: Average distance of molecular mass from its center of mass. Describes molecular size and mass distribution in 3D.
>      - ...
> 3. Topological Properties
>      - Topological polar surface area (TPSA): Sum of surfaces of all polar atoms (mainly oxygen and nitrogen), correlating with drug absorption, including intestinal absorption and blood-brain barrier penetration.
>      - ...
>
> **Message 2:** For the primary task [TASK] (TASK DESCRIPTION), you assess the relevance between retrieved properties and the primary task with affinity scores, based on domain-specific knowledge and the summarised information.
> **Response:** ...
>
> **Message 3:** For the primary task [TASK] (TASK DESCRIPTION), you recommend which $[K]$ properties as auxiliary tasks to improve the machine learning model performance on the primary task. You provide: a list of $[K]$ selected auxiliary tasks and the affinity score of each auxiliary task to the primary task [TASK].
> **Response:** ...

## A.3 Discussion of selected auxiliary tasks

Table 5 presents the auxiliary tasks selected by different LLMs (GPT-4, LLaMA 3, and Gemini 2.0) across each molecular prediction dataset. Across all models, we observe consistent selection of chemically relevant descriptors such as `LogP`, `Topological Polar Surface Area`, and `Number of Hydrogen Bond Donors`, confirming their general importance for molecular property prediction.

For classification datasets like BBBP and TOX21, all three LLMs prioritize physicochemical and toxicity-related descriptors. Notably, `LogP` and `TPSA` appear in nearly all runs, underscoring their relevance to tasks involving membrane permeability and drug-likeness. In contrast, for regression datasets such as ESOL, FREESOLV, and LIPO, the selected auxiliary tasks increasingly include 3D structural properties like `Molecular Volume in Å`$^3$, `Radius of Gyration`, `Eccentricity`, and `Asphericity`, which are crucial for modeling continuous-valued properties.

Comparing across LLMs, we find that GPT-4 selections are more stable, with strong alignment to domain-relevant properties across all datasets. LLaMA 3 maintains good coverage of core features, but introduces additional diversity in its selections, occasionally replacing QED or TPSA with alternatives like `Balaban J Index` or `Heavy Atom Molecular Weight`. Gemini 2.0 shows

Table 5: Illustration of selected auxiliary tasks. The numbers in parentheses $(k, w)$ indicate the number of times each task was chosen out of five repeats $(k)$, and its average learned weight $(w)$.

| Dataset | $K$ | LLM | LLM Selected Auxiliary Tasks |
|---|---|---|---|
| BBBP | 5 | GPT-4 | Topological Polar Surface Area (5, 0.28), LogP (5, 0.27), Number of Hydrogen Bond Donors (5, 0.24), Molecular Weight (5, 0.21), QED (5, 0.20) |
| | | Llama 3 | Topological Polar Surface Area (5, 0.26), LogP (5, 0.25), Number of Hydrogen Bond Donors (5, 0.21), Balaban J Index (4, 0.13), NO Group Count (3, 0.11), QED (2, 0.09), Molecular Weight (1, 0.04) |
| | | Gemini 2.0 | Topological Polar Surface Area (5, 0.27), LogP (5, 0.26), Number of Hydrogen Bond Donors (5, 0.23), Molecular Weight (4, 0.13), QED (3, 0.10), Balaban J Index (2, 0.07), Heavy Atom Molecular Weight (1, 0.03) |
| BACE | 5 | GPT-4 | LogP (5, 0.291), Topological Polar Surface Area (5, 0.259), Number of Hydrogen Bond Donors (5, 0.229), Molar Refractivity (5, 0.219), QED (5, 0.151) |
| | | Llama 3 | LogP (5, 0.26), Topological Polar Surface Area (5, 0.25), Molar Refractivity (5, 0.22), Heavy Atom Molecular Weight (3, 0.13), EState VSA1 (2, 0.08), Number of Hydrogen Bond Donors (2, 0.07), QED (1, 0.03) |
| | | Gemini 2.0 | LogP (5, 0.27), Topological Polar Surface Area (5, 0.26), Number of Hydrogen Bond Donors (4, 0.17), Molar Refractivity (4, 0.15), QED (3, 0.09), Total Formal Charge (1, 0.03), NO Group Count (1, 0.02) |
| ClinTox | 5 | GPT-4 | LogP (5, 0.277), Topological Polar Surface Area (5, 0.242), Number of Hydrogen Bond Donors (5, 0.243), QED (3, 0.190), Bertz Complexity (3, 0.150), Fraction of Csp3 Carbon (2, 0.110), Molecular Weight (1, 0.08), Number of Aromatic Rings (1, 0.06) |
| | | Llama 3 | LogP (5, 0.25), Topological Polar Surface Area (5, 0.23), Bertz Complexity (4, 0.13), NHOH Group Count (3, 0.11), Balaban J Index (3, 0.10), Number of Hydrogen Bond Donors (2, 0.09), QED (2, 0.09) |
| | | Gemini 2.0 | LogP (5, 0.26), Topological Polar Surface Area (4, 0.15), QED (3, 0.12), Bertz Complexity (3, 0.10), Molecular Weight (2, 0.08), Aliphatic Rings Count (2, 0.08), NHOH Group Count (1, 0.04) |
| Tox21 | 5 | GPT-4 | LogP (5, 0.23), Topological Polar Surface Area (5, 0.22), Number of Hydrogen Bond Donors (5, 0.21), Bertz Complexity (5, 0.19), NO Group Count (5, 0.18) |
| | | Llama 3 | LogP (5, 0.21), Topological Polar Surface Area (5, 0.20), NO Group Count (5, 0.18), Number of Hydrogen Bond Donors (4, 0.15), EState VSA1 (3, 0.10), Bertz Complexity (2, 0.07) |
| | | Gemini 2.0 | LogP (5, 0.22), Number of Hydrogen Bond Donors (5, 0.20), QED (3, 0.12), Eccentricity (2, 0.09), NO Group Count (2, 0.09), Fraction of Csp3 Carbon Atoms (2, 0.08), TPSA (1, 0.04) |
| ToxCast | 5 | GPT-4 | LogP (5, 0.21), Topological Polar Surface Area (5, 0.20), QED (5, 0.19), Number of Hydrogen Bond Donors (4, 0.14), Balaban J Index (3, 0.09), Number of Rotatable Bonds (3, 0.08) |
| | | Llama 3 | LogP (5, 0.22), Balaban J Index (4, 0.14), Labute Approximate Surface Area (4, 0.13), EState VSA1 (3, 0.11), Topological Polar Surface Area (2, 0.07), QED (2, 0.06), Rotatable Bonds (1, 0.03) |
| | | Gemini 2.0 | LogP (5, 0.21), Topological Polar Surface Area (4, 0.11), Balaban J Index (3, 0.08), Rotatable Bonds (3, 0.08), Labute Approximate Surface Area (2, 0.05), QED (2, 0.06), TPSA (1, 0.02) |
| SIDER | 5 | GPT-4 | LogP (5, 0.20), Topological Polar Surface Area (5, 0.18), QED (4, 0.12), Number of Hydrogen Bond Donors (4, 0.12), Number of Rotatable Bonds (4, 0.10), Molecular Weight (3, 0.08) |
| | | Llama 3 | LogP (5, 0.19), Topological Polar Surface Area (4, 0.12), Aliphatic Rings Count (4, 0.10), Fraction of Csp3 Carbon Atoms (3, 0.08), Number of Hydrogen Bond Donors (2, 0.07), QED (2, 0.06) |
| | | Gemini 2.0 | Topological Polar Surface Area (5, 0.19), LogP (5, 0.18), Number of Hydrogen Bond Donors (4, 0.12), QED (3, 0.09), Molecular Weight (2, 0.06), Aliphatic Rings Count (1, 0.03) |
| ESOL | 5 | GPT-4 | LogP (5, 0.19), Molecular Weight (5, 0.18), Topological Polar Surface Area (5, 0.17), Molecular Volume in Å$^3$ (5, 0.17), QED (1, 0.05), Fraction of Csp3 Carbon Atoms (1, 0.04), Radius of Gyration (1, 0.03), Eccentricity (1, 0.03), Asphericity (1, 0.03) |
| | | Llama 3 | LogP (5, 0.17), Molecular Weight (5, 0.16), Topological Polar Surface Area (5, 0.15), Molecular Volume in Å$^3$ (4, 0.10), Eccentricity (2, 0.05), Inertial Shape Factor (2, 0.04), QED (1, 0.03) |
| | | Gemini 2.0 | LogP (5, 0.16), Molecular Weight (5, 0.15), Topological Polar Surface Area (4, 0.10), Molecular Volume in Å$^3$ (3, 0.08), Radius of Gyration (2, 0.05), Eccentricity (2, 0.05), Asphericity (2, 0.05), Inertial Shape Factor (1, 0.03), QED (1, 0.03) |
| FreeSolv | 5 | GPT-4 | LogP (5, 0.19), QED (5, 0.18), Topological Polar Surface Area (4, 0.12), Molecular Volume in Å$^3$ (4, 0.12), Molar Refractivity (4, 0.10), Radius of Gyration (3, 0.08) |
| | | Llama 3 | LogP (5, 0.18), QED (4, 0.12), Topological Polar Surface Area (4, 0.12), Molecular Volume in Å$^3$ (3, 0.09), Molar Refractivity (3, 0.08), Eccentricity (2, 0.05) |
| | | Gemini 2.0 | LogP (5, 0.17), Molecular Volume in Å$^3$ (4, 0.11), QED (3, 0.08), Topological Polar Surface Area (3, 0.08), Fraction of Csp3 Carbon Atoms (2, 0.06), Molar Refractivity (2, 0.06), Radius of Gyration (1, 0.03) |
| Lipo | 5 | GPT-4 | LogP (5, 0.18), Molecular Weight (5, 0.17), Topological Polar Surface Area (4, 0.13), Number of Hydrogen Bond Donors (4, 0.13), Molar Refractivity (4, 0.12), Fraction of Csp3 Carbon (2, 0.06), Radius of Gyration (1, 0.03) |
| | | Llama 3 | LogP (5, 0.16), Molecular Weight (5, 0.16), Molar Refractivity (4, 0.11), Topological Polar Surface Area (3, 0.09), Number of Hydrogen Bond Donors (2, 0.06), Heavy Atom Molecular Weight (2, 0.05), Fraction of Csp3 Carbon Atoms (1, 0.03) |
| | | Gemini 2.0 | LogP (5, 0.15), Molecular Weight (5, 0.15), Molar Refractivity (4, 0.11), Topological Polar Surface Area (3, 0.08), Number of Hydrogen Bond Donors (2, 0.05), Fraction of Csp3 Carbon Atoms (2, 0.05), Eccentricity (1, 0.03) |

greater variability and less task consistency, particularly in regression settings such as FREESOLV and ESOL, where it selects more diverse and occasionally less relevant 3D shape descriptors. This aligns with the drop in performance observed in Figure 3c.

Despite the inherent stochasticity of LLM outputs, the consistency in repeatedly selected descriptors (shown by the frequencies in parentheses) confirms that all models—especially GPT-4—are capable of capturing task-relevant molecular knowledge. These results support our claim that LLM-guided auxiliary task selection is both feasible and chemically meaningful, and also highlight that LLM quality impacts the robustness of selection.

**Costs of LLM Usages.** Our automatic auxiliary–task pipeline made 137 calls to GPT-4, two retrieval and three selection requests for each of the 9 target datasets in each of the five training runs. And 135 additional calls to Gemini 2.0 for the ablation in Figure 3c. A GPT-4 request needed $21.3 \pm 2.4$ s on average, while a Gemini 2.0 request required $7.8 \pm 1.1$ s. Using the research-tier prices (\$ 0.06 / 1 K prompt tokens and \$ 0.12 / 1 K completion tokens for GPT-4; USD 0.02 / 1 K prompt tokens and USD 0.02 / 1 K completion tokens for Gemini 2.0) and the logged token counts of roughly 1 K prompt + 1 K completion tokens per call, the GPT-4 portion cost about \$23 and the Gemini portion about \$9. The total large-language-model expenditure is therefore approximately \$32, or \$3.6 per downstream dataset across all five runs. GPT-4 accounts for most of the bill because it retrieves and ranks all auxiliary tasks and their textual descriptions; Gemini 2.0 is used only to show that a less expensive model can replace GPT-4 without a measurable drop in accuracy. Llama 3 runs on our GPUs, so it incurs no additional cost beyond electricity.

# B  Proof

## B.1  Proof of Theorem 1

Let $\nabla \mathcal{L}_{\text{M}}(\theta) \in \mathbb{R}^d$ denote the gradient of the primary task loss, and let $\{\nabla \mathcal{L}_{\text{S}}^k(\theta)\}_{k=1}^K$ represent the gradients of the auxiliary task losses. These auxiliary gradients span a subspace $\mathcal{S} \subseteq \mathbb{R}^d$.

**Step 1: Gradient Representation.** Since $\mathcal{S}$ is the span of the auxiliary task gradients, any vector in $\mathcal{S}$ can be expressed as a linear combination of these gradients:

$$\sum_{k=1}^K \alpha_k \nabla \mathcal{L}_{\text{S}}^k(\theta) \in \mathcal{S}, \tag{8}$$

where $\boldsymbol{\alpha} = \{\alpha_1, \ldots, \alpha_K\}$ are the auxiliary task weights. If $\nabla \mathcal{L}_{\text{M}}(\theta) \in \mathcal{S}$, then there exist coefficients $\{\beta_k\}$ such that:

$$\nabla \mathcal{L}_{\text{M}}(\theta) = \sum_{k=1}^K \beta_k \nabla \mathcal{L}_{\text{S}}^k(\theta). \tag{9}$$

**Step 2: Exact Reconstruction of $\nabla \mathcal{L}_{\text{M}}(\theta)$.** The primary task gradient can be optimally reconstructed using the auxiliary task gradients if there exist weights $\{\alpha_k\}$ such that:

$$\sum_{k=1}^K \alpha_k \nabla \mathcal{L}_{\text{S}}^k(\theta) = \nabla \mathcal{L}_{\text{M}}(\theta). \tag{10}$$

In this case, the reconstruction error is zero:

$$\left\| \nabla \mathcal{L}_{\text{M}}(\theta) - \sum_{k=1}^K \alpha_k \nabla \mathcal{L}_{\text{S}}^k(\theta) \right\| = 0. \tag{11}$$

This implies that the auxiliary tasks fully capture the primary task gradient, leading to complete alignment.

**Step 3: Decomposition with Orthogonal Components.** If $\nabla \mathcal{L}_{\text{M}}(\theta) \notin \mathcal{S}$, it can be decomposed into two components:

$$\nabla \mathcal{L}_{\text{M}}(\theta) = \nabla \mathcal{L}_{\text{M}}^{\|}(\theta) + \nabla \mathcal{L}_{\text{M}}^{\perp}(\theta), \tag{12}$$

where:

- $\nabla \mathcal{L}_{\text{M}}^{\|}(\theta) \in \mathcal{S}$ is the projection of $\nabla \mathcal{L}_{\text{M}}(\theta)$ onto $\mathcal{S}$,
- $\nabla \mathcal{L}_{\text{M}}^{\perp}(\theta) \perp \mathcal{S}$ is the orthogonal component.

The best approximation of $\nabla \mathcal{L}_{\text{M}}(\theta)$ within $\mathcal{S}$ is given by its projection:

$$\sum_{k=1}^K \alpha_k^* \nabla \mathcal{L}_{\text{S}}^k(\theta) = \nabla \mathcal{L}_{\text{M}}^{\|}(\theta). \tag{13}$$

The reconstruction error in this case is:

$$\|\nabla\mathcal{L}_{\mathrm{M}}(\theta) - \sum_{k=1}^{K} \alpha_k \nabla\mathcal{L}_{\mathrm{S}}^k(\theta)\| = \|\nabla\mathcal{L}_{\mathrm{M}}^{\perp}(\theta)\| > 0. \tag{14}$$

The presence of $\nabla\mathcal{L}_{\mathrm{M}}^{\perp}(\theta)$ implies that the auxiliary task gradients cannot fully reconstruct the primary task gradient, reducing their effectiveness.

**Step 4: Implications for Optimization.** When $\nabla\mathcal{L}_{\mathrm{M}}(\theta) \in \mathcal{S}$, we obtain:

$$\|\nabla\mathcal{L}_{\mathrm{M}}(\theta) - \sum_{k=1}^{K} \alpha_k \nabla\mathcal{L}_{\mathrm{S}}^k(\theta)\| = 0. \tag{15}$$

Thus, the auxiliary tasks provide complete gradient information for optimizing the primary task, resulting in:

- Improved convergence rates, as optimization follows an aligned gradient trajectory.
- Reduced variance, since the primary task gradient is entirely captured by the auxiliary tasks.

Conversely, if $\nabla\mathcal{L}_{\mathrm{M}}^{\perp}(\theta) \neq 0$, the orthogonal component introduces misalignment in the optimization process, leading to:

- Slower convergence, as optimization is influenced by gradients that do not fully capture the primary task.
- Potential suboptimal solutions, since the primary task is not entirely represented in the auxiliary gradient space.

**Conclusion.** The auxiliary tasks contribute maximally to primary task optimization when $\nabla\mathcal{L}_{\mathrm{M}}(\theta) \in \mathcal{S}$, ensuring full alignment with the optimization trajectory. When this condition holds, the auxiliary tasks effectively serve as surrogates for the primary task, guiding learning efficiently. However, when $\nabla\mathcal{L}_{\mathrm{M}}^{\perp}(\theta) \neq 0$, the mismatch introduces misalignment, emphasizing the need for selecting auxiliary tasks whose gradients best align with the primary task gradient.

$\square$

### B.2 Proof of Theorem 2

**Step 1: Generalization Error Definition.** The generalization error $\mathcal{E}(\theta)$ is the difference between the expected loss on the data distribution and the empirical loss on the training set:

$$\mathcal{E}(\theta) = \mathbb{E}_{(X,Y)\sim\mathcal{D}_m}\left[\ell(f_\theta(X), Y)\right] - \frac{1}{n}\sum_{i=1}^{n}\ell(f_\theta(X_i), Y_i), \tag{16}$$

where $\ell(\cdot, \cdot)$ is the loss function, $\mathcal{D}_m$ represents the primary task data distribution, and $n$ is the number of training samples.

**Step 2: Auxiliary Tasks and the Hypothesis Class.** Incorporating auxiliary tasks $\mathcal{T}_s$ modifies the training objective by introducing a set of trainable weights $\boldsymbol{\alpha}$ that dynamically adjust the contribution of auxiliary task losses:

$$\mathcal{L}(\theta, \boldsymbol{\alpha}) = \mathcal{L}_{\mathrm{M}}(\theta) + \sum_{k=1}^{K} \alpha_k \mathcal{L}_{\mathrm{S}}^k(\theta). \tag{17}$$

This formulation does not change the hypothesis class itself but adjusts the optimization trajectory by guiding parameter updates through gradient alignment.

**Step 3: Rademacher Complexity with Dynamic Weighting.** The Rademacher complexity $\mathcal{R}_n(\mathcal{H})$ measures the expressive capacity of the function class:

$$\mathcal{R}_n(\mathcal{H}) = \mathbb{E}_\sigma\left[\sup_{f_\theta\in\mathcal{H}} \frac{1}{n}\sum_{i=1}^{n}\sigma_i f_\theta(X_i)\right]. \tag{18}$$

Unlike traditional multi-task learning, which may increase $\mathcal{R}_n(\mathcal{H})$ by introducing additional objectives, your framework does not expand the hypothesis space but rather optimizes within the same space using dynamically weighted gradients.

The key effect of auxiliary tasks in your framework is that it improves gradient alignment, ensuring that updates to $\theta$ are more consistent with the primary task objective. This leads to:

$$\mathcal{R}_n(\mathcal{H}_{\boldsymbol{\alpha}}) \leq \mathcal{R}_n(\mathcal{H}), \tag{19}$$

where $\mathcal{H}_{\boldsymbol{\alpha}}$ denotes the function class learned with dynamically weighted auxiliary tasks.

**Step 4: Generalization Bound.** From statistical learning theory, the generalization error satisfies:

$$\mathcal{E}(\theta) \leq \hat{\mathcal{E}}(\theta) + c \cdot \mathcal{R}_n(\mathcal{H}_{\boldsymbol{\alpha}}), \tag{20}$$

where:

- $\hat{\mathcal{E}}(\theta)$ is the empirical error minimized during training.
- $\mathcal{R}_n(\mathcal{H}_{\boldsymbol{\alpha}})$ is the Rademacher complexity of the function class trained with auxiliary task reweighting.
- $c$ is a constant dependent on the Lipschitz continuity of $\ell(\cdot, \cdot)$.

Since dynamic weighting ensures auxiliary tasks contribute positively through gradient alignment, it lowers the effective complexity of the hypothesis class while improving empirical performance.

**Conclusion.** The auxiliary task reweighting strategy in AUTAUT improves generalization by enhancing gradient alignment rather than modifying the hypothesis class itself. This leads to better generalization bounds without introducing unnecessary complexity.

$\square$

## C   Algorithm

The proposed algorithm, AUTAUT, optimizes the primary task by leveraging auxiliary tasks through a structured three-phase learning procedure. The key idea is to dynamically integrate auxiliary supervision to refine the learning process while ensuring that auxiliary tasks positively contribute to the primary task. To achieve this, AUTAUT systematically retrieves auxiliary tasks, adapts their influence during training, and fine-tunes the model for optimal performance.

AUTAUT begins by retrieving a set of potential auxiliary task labels using a pre-trained LLM. This step identifies molecular property prediction tasks that are related to the primary task, which are then ranked based on their gradient alignment with the primary task. Only the most relevant auxiliary tasks are retained, and their initial weights are set according to their alignment scores. This step ensures that the model does not incorporate noisy or misleading auxiliary signals.

During training, AUTAUT jointly optimizes the primary and auxiliary tasks. The model parameters are updated based on the primary task gradient, combined with weighted gradients from the selected auxiliary tasks. The auxiliary task weights are not fixed but rather adapted dynamically using a gradient-based Fisher divergence minimization strategy. This allows AUTAUT to progressively emphasize auxiliary tasks that contribute positively while down-weighting those that introduce conflicting gradients. The auxiliary task weights are constrained to a probability simplex, ensuring a balanced contribution of tasks.

Once training stabilizes, the final fine-tuning phase optimizes the model exclusively on the primary task while keeping the auxiliary task weights fixed. This prevents auxiliary supervision from dominating the optimization and ensures that the learned model parameters generalize well to unseen data.

The complete training procedure is outlined in Algorithm 1. The algorithm follows three structured phases:

1. Retrieval and Initialization, where relevant auxiliary tasks are selected based on LLM retrieval and gradient alignment.

**Algorithm 1** AUTAUT: Optimizing Primary Task with Auxiliary Task Integration

---

1: **Input:** Primary task dataset $\mathcal{D}_{\text{TRAIN}}$, auxiliary task dataset $\mathcal{D}_{\text{TRAIN}}^k$, pre-trained LLM $f_{\text{LLM}}$
2: **Parameters:** Learning rates $\epsilon_t$, $\beta$; convergence threshold $\delta$; maximum epochs $E$, fine-tuning epochs $E_{\text{fine}}$
3: **Output:** Optimized model parameters $\theta^*$
4: **Phase 1: Retrieval and Initialization**
5: Query LLM to retrieve potential auxiliary task labels:
6: $\mathcal{Y}_a \leftarrow f_{\text{LLM}}(\mathcal{D}_{\text{TRAIN}})$
7: Compute gradient alignment scores for tasks in $\mathcal{Y}_a$ w.r.t. $\mathcal{D}_{\text{TRAIN}}$
8: $\mathcal{Y}_s \leftarrow$ Top-K auxiliary task labels selected based on alignment scores
9: Initialize task weights $\boldsymbol{\alpha}$ proportionally to alignment scores
10: Initialize model parameters $\theta_0$
11: **Phase 2: Joint Training with Adaptive Weighting**
12: **for** epoch $e = 1$ to $E$ **do**
13:     **for** iteration $t = 1$ to $T$ **do**
14:         Compute primary task gradient: $g_p \leftarrow \nabla \log p(\mathcal{D}_{\text{TRAIN}}|\theta_{t-1})$
15:         Compute auxiliary task gradients: $g_k \leftarrow \nabla \log p(\mathcal{D}_{\text{TRAIN}}^k|\theta_{t-1}) \quad \forall k$
16:         Update model parameters: $\theta_t \leftarrow \theta_{t-1} - \epsilon_t\big(-g_p - \sum_{k=1}^{K} \alpha_k g_k\big)$
17:         **Adaptive Task Weight Optimization**
18:         **if** not converged($\boldsymbol{\alpha}$) **then**
19:           Update task weights via Fisher divergence minimization: $\boldsymbol{\alpha} \leftarrow \boldsymbol{\alpha} - \beta\nabla_{\boldsymbol{\alpha}}\|\nabla \log p(\mathcal{D}_{\text{TRAIN}}|\theta_t) - \nabla \log p_{\boldsymbol{\alpha}}(\theta_t)\|_2^2$
20:           Project $\boldsymbol{\alpha}$ onto simplex $\mathcal{A}$         ▷ Ensure $\alpha_k \geq 0$, $\sum_k \alpha_k = 1$
21:         **end if**
22:     **end for**
23:     Evaluate model on validation set $\mathcal{D}_{\text{VALID}}$
24:     **if** validation performance plateaus or shows marginal improvement **then**
25:         **break**
26:     **end if**
27: **end for**
28: **Phase 3: Fine-Tuning**
29: **for** epoch $e = 1$ to $E_{\text{fine}}$ **do**
30:     Fix auxiliary task weights $\boldsymbol{\alpha}$
31:     Update model parameters: $\theta \leftarrow \theta - \epsilon_t\big(-\nabla \log p(\mathcal{D}_{\text{TRAIN}}|\theta) - \sum_{k=1}^{K} \alpha_k \nabla \log p(\mathcal{D}_{\text{TRAIN}}^k|\theta)\big)$
32:     Evaluate model on validation set $\mathcal{D}_{\text{VALID}}$
33:     **if** validation performance plateaus or shows marginal improvement **then**
34:         **break**
35:     **end if**
36: **end for**
37: **Return:** Optimized model parameters $\theta^*$

---

2. Joint Training with Adaptive Weighting, where model parameters and auxiliary task weights are updated iteratively using gradient alignment and Fisher divergence minimization.

3. Fine-Tuning, where the model is optimized on the primary task with fixed auxiliary task weights to ensure stable generalization.

By integrating these steps, AUTAUT automatically learns an effective task weighting strategy, reducing the reliance on manual task selection while enhancing the predictive performance of the primary task.

## D Computational Complexity Analysis

AUTAUT consists of three phases: retrieval and initialization, joint training with adaptive weighting, and fine-tuning. The computational complexity is primarily determined by the auxiliary task selection process, the gradient-based joint training, and the adaptive weight optimization.

### D.1 Phase 1: Retrieval and Initialization

Retrieving auxiliary tasks using an LLM query has a complexity of $\mathcal{O}(1)$ if the LLM response time is independent of dataset size. However, if retrieval involves searching a database of $N_a$ potential auxiliary tasks, the complexity is $\mathcal{O}(N_a)$. Computing gradient alignment for auxiliary task selection requires evaluating the cosine similarity between the primary task gradient and each auxiliary task gradient, which results in a cost of $\mathcal{O}(N_a D)$, where $D$ is the number of model parameters. Selecting the top $K$ auxiliary tasks incurs a sorting cost of $\mathcal{O}(N_a \log N_a)$. Thus, the total complexity for retrieval and initialization is:

$$\mathcal{O}(N_a D + N_a \log N_a). \tag{21}$$

### D.2 Phase 2: Joint Training with Adaptive Weighting

Each training iteration involves computing gradients for both the primary and auxiliary tasks, requiring $\mathcal{O}(D)$ operations per task. Since there are $K$ selected auxiliary tasks, the total cost per iteration is $\mathcal{O}((K + 1)D)$. Updating model parameters using stochastic gradient descent (SGD) incurs an additional $\mathcal{O}(D)$ cost per step. Computing gradient alignment scores requires dot products between the primary task gradient and $K$ auxiliary gradients, adding an extra $\mathcal{O}(KD)$ per iteration. Updating auxiliary task weights via Fisher divergence minimization requires computing weight gradients and projecting onto a simplex, which contributes another $\mathcal{O}(KD)$. Over $E$ epochs with $T$ iterations per epoch, the total complexity for joint training is:

$$\mathcal{O}(ET(KD + D + KD)) = \mathcal{O}(ETKD). \tag{22}$$

### D.3 Phase 3: Fine-Tuning

Fine-tuning updates only the primary task parameters with fixed auxiliary task weights, resulting in a per-iteration complexity of $\mathcal{O}(D)$. Over $E_{\text{fine}}$ fine-tuning epochs, the total complexity is:

$$\mathcal{O}(E_{\text{fine}} TD). \tag{23}$$

### D.4 Overall Complexity

Summing the complexity of all phases, the total computational cost of AUTAUT is:

$$\mathcal{O}(N_a D + N_a \log N_a) + \mathcal{O}(ETKD) + \mathcal{O}(E_{\text{fine}} TD). \tag{24}$$

If $E_{\text{fine}} \approx E$, this simplifies to:

$$\mathcal{O}(N_a D + N_a \log N_a + ETKD). \tag{25}$$

### D.5 Comparative Analysis

For comparison, standard single-task learning has complexity $\mathcal{O}(ETD)$, while traditional multi-task learning with fixed auxiliary tasks has complexity $\mathcal{O}(ETKD)$ but lacks dynamic weighting. AUTAUT introduces additional costs from auxiliary task retrieval and weight adaptation but avoids the need for manual task selection.

### D.6 Scalability Considerations

If $K \ll D$ (*i.e.*, a small number of auxiliary tasks), AUTAUT remains comparable in complexity to standard multi-task learning. If $N_a$ is large, auxiliary task selection may be costly, but it is a one-time process and does not impact training iterations. Since gradient-based weight updates scale linearly with $K$, AUTAUT remains computationally feasible for moderate values of $K$.

In Table 6, we report the number of model parameters, GPU memory usage, and average execution time of ML models with and without AUTAUT across 5 runs. In practice, the additional computational burden from auxiliary task integration is minimal.

Overall, AUTAUT achieves a balance between improved learning efficiency and computational cost, making it a scalable approach for molecular property prediction tasks.

Table 6: Training information.

| Model | Dataset | #Param | GPU Mem (MB) | Time (min) Baseline | Time (min) +AUTAUT |
|---|---|---|---|---|---|
| **GCN** | BACE | 296,833 | 534 | 2.37 | 2.47 |
| | BBBP | 296,833 | 536 | 3.14 | 3.23 |
| | ClinTox | 296,962 | 536 | 2.30 | 2.37 |
| | Tox21 | 298,252 | 532 | 12.17 | 12.64 |
| | ToxCast | 376,297 | 538 | 18.75 | 19.44 |
| | SIDER | 300,187 | 562 | 2.42 | 2.50 |
| | ESOL | 296,833 | 508 | 1.77 | 1.84 |
| | FreeSolv | 296,833 | 506 | 1.00 | 1.04 |
| | Lipo | 296,833 | 536 | 6.44 | 6.68 |
| **GIN** | BACE | 496,005 | 546 | 2.31 | 2.39 |
| | BBBP | 496,005 | 548 | 3.08 | 3.19 |
| | ClinTox | 496,134 | 548 | 2.28 | 2.36 |
| | Tox21 | 497,424 | 542 | 12.00 | 12.47 |
| | ToxCast | 575,469 | 546 | 18.69 | 19.34 |
| | SIDER | 499,359 | 570 | 2.41 | 2.53 |
| | ESOL | 496,005 | 518 | 1.76 | 1.83 |
| | FreeSolv | 496,005 | 512 | 0.99 | 1.03 |
| | Lipo | 496,005 | 550 | 6.38 | 6.64 |
| **Graphormer** | BACE | 7,944,160 | 10,957 | 10.59 | 11.12 |
| | BBBP | 7,944,160 | 10,997 | 14.48 | 15.07 |
| | ClinTox | 7,946,740 | 10,997 | 10.45 | 10.88 |
| | Tox21 | 7,972,540 | 10,898 | 59.31 | 60.56 |
| | ToxCast | 9,533,440 | 10,998 | 92.52 | 94.28 |
| | SIDER | 8,011,240 | 11,478 | 11.02 | 11.57 |
| | ESOL | 7,944,160 | 10,417 | 7.81 | 7.96 |
| | FreeSolv | 7,944,160 | 10,338 | 3.90 | 4.05 |
| | Lipo | 7,944,160 | 11,018 | 31.02 | 32.50 |

# E  Related Work Discussion

Auxiliary task selection related work:

1. **Beam Search** [37]. Beam Search is a task selection method that incrementally builds a set of auxiliary tasks by scoring candidates based on intermediate performance gains. It selects the top-scoring sequence of tasks using beam width exploration.
   *Important settings:* It adopts all collected auxiliary tasks as shown in Table 4. Selection is performed by evaluating model performance on the validation set at each expansion step.

2. **TAG** [8]. TAG (Task Affinity Grouping) clusters tasks by computing pairwise gradient similarities and forming groups to minimize negative interference. It selects task groups with high internal compatibility.
   *Important settings:* It uses all collected auxiliary tasks as shown in Table 4 and forms a group that includes the primary task. Task affinity is computed via cosine similarity of gradient vectors.

3. **Task2vec** [1]. Task2vec represents tasks as embedding vectors derived from the Fisher Information Matrix and selects auxiliary tasks based on embedding proximity.
   *Important settings:* It adopts all collected auxiliary tasks as shown in Table 4 and uses embedding similarity to rank and select top-$K$ tasks closest to the primary task in representation space.

4. **MolGroup** [17]. MolGroup groups tasks by jointly optimizing group assignments and shared representations using a bi-level optimization framework.
   *Important settings:* It adopts all collected auxiliary tasks as shown in Table 4 and learns task groupings end-to-end during training.

5. **GS-Meta** [68]. GS-Meta (Graph Sampling-based Meta-Learning) learns task-specific initializations using meta-learning, adapting quickly to new tasks with few updates.
   *Important settings:* We adopt the auxiliary datasets and molecular property relation graph provided in their official implementation.

6. **InstructMol** [51]. InstructMol leverages instruction-based supervision by encoding auxiliary task semantics through natural language prompts during pretraining.
   *Important settings:* We adopt the auxiliary datasets provided in their official implementation.

Adaptive multi-task learning models:

1. **MTDNN** [25]. MTDNN (Multi-task Deep Neural Network) trains on all tasks simultaneously and learns to balance them via a shared backbone and a task discriminator.
   *Important settings:* It uses the same number of randomly selected auxiliary task set as AUTAUT (as shown in Table 5). The task discriminator is trained to identify relevant tasks dynamically during training.

2. **Unweighted Averages (UA)**. UA is a naive baseline where all auxiliary task losses are included with equal weight. It assumes uniform contribution from each auxiliary task.
   *Important settings:* It uses the same number of randomly selected auxiliary task set as AUTAUT (as shown in Table 5), but without adaptive weighting.

3. **GradNorm** [3]. GradNorm adjusts task-specific learning rates to balance gradient magnitudes across tasks, aiming for balanced training dynamics.
   *Important settings:* It uses the same number of randomly selected auxiliary task set as AUTAUT (as shown in Table 5) and dynamically reweights task losses by normalizing their gradient norms.

4. **Pretrain-Finetune (PF)** [15]. PF is a two-stage pipeline that first pretrains the model on a large source dataset and then fine-tunes it on the target task without auxiliary tasks.
   *Important settings:* The model is pretrained on PCQM4Mv2 [14] and then fine-tuned on each downstream dataset individually.

**Discussion.** Table 1 summarizes the key differences between AUTAUT and prior auxiliary-task-based methods. Most existing approaches, including TAG, TASK2VEC, GRADNORM, and MTDNN, rely on manually curated auxiliary tasks or require detailed task descriptions and additional datasets. While some methods (*e.g.*, GS-META, MOLGROUP) support automatic task selection or adaptive weighting, they still depend heavily on external supervision or relation graphs to guide selection.

In contrast, AUTAUT is the only method that achieves both automatic auxiliary task retrieval and automatic task selection without requiring any human-generated task lists, labels, or external datasets. Furthermore, it uniquely integrates selected tasks using a gradient alignment-based adaptive weighting mechanism, which dynamically adjusts the importance of each auxiliary task during training. This combination enables AUTAUT to operate entirely without manual input or domain knowledge, setting it apart as the first fully automated framework for auxiliary task discovery and integration in molecular property prediction.

This design significantly lowers the barrier for deploying auxiliary-task-based learning in domains where expert-curated auxiliary signals are unavailable or difficult to define, highlighting AUTAUT's scalability and practical utility.

# F   Experimental Settings

**Datasets.** We conduct experiments on the MoleculeNet benchmark [52], a widely used dataset collection designed to evaluate molecular property prediction models. MoleculeNet includes a diverse range of tasks spanning physicochemical properties, bioactivity, toxicity, and side effects, providing a comprehensive testbed for assessing the effectiveness of our algorithm. For our evaluation, we select 9 representative datasets covering classification and regression tasks, described as follows:

1. **BBBP**. The Blood-Brain Barrier Penetration (BBBP) dataset contains binary classification labels indicating whether a molecule can permeate the blood-brain barrier. This property is crucial for designing central nervous system drugs and understanding their pharmacokinetics.

2. **BACE**. The BACE dataset provides both quantitative (IC50 values) and qualitative (binary classification) binding data for inhibitors of human $\beta$-secretase 1 (BACE-1), a key target for Alzheimer's disease treatment.

3. **CLINTOX**. The ClinTox dataset compares FDA-approved drugs with those that failed clinical trials due to toxicity. It presents a binary classification challenge for identifying toxic compounds early in drug discovery.

4. **TOX21**. The "Toxicology in the 21st Century" (Tox21) initiative compiles a dataset of qualitative toxicity measurements for 12 biological targets, including nuclear receptors and stress response pathways. It serves as a multi-label classification problem for assessing compound toxicity.

5. **TOXCAST**. An extension of Tox21, the ToxCast dataset contains in vitro high-throughput screening data for thousands of environmental and pharmaceutical compounds, spanning over 600 toxicity-related assays.

6. **SIDER**. The Side Effect Resource (SIDER) dataset aggregates information on adverse drug reactions (ADRs) for marketed drugs. The dataset categorizes side effects into 27 system organ classes, making it a multi-label classification task.

7. **ESOL**. The ESOL dataset comprises solubility measurements (log solubility in mol/L) for small organic molecules in water. Predicting solubility is a fundamental task in drug formulation and molecular design.

8. **FREESOLV**. The FreeSolv dataset provides both experimental and calculated hydration-free energy values for small molecules in water. This dataset is valuable for studying solvation effects and molecular interactions in aqueous environments.

9. **LIPO**. Lipophilicity, quantified by the octanol/water distribution coefficient (logD at pH 7.4), is a critical property influencing drug absorption, membrane permeability, and solubility. This dataset provides experimentally measured logD values for a variety of drug-like compounds.

**Data Split.** To ensure a rigorous evaluation, we follow prior work [6, 65] and adopt scaffold splitting to divide datasets into training, validation, and test sets with an 80%-10%-10% ratio. Unlike random splitting, scaffold splitting is a more challenging and realistic partitioning strategy, as molecules in different subsets do not share structural scaffolds [65]. This setup better simulates real-world scenarios where models must generalize to structurally novel compounds [52]. Zhou *et al.*[65] also highlight that chirality considerations in RDKit scaffold generation significantly impact the resulting partitions. For fair comparisons, we select the checkpoint with the best validation loss for final evaluation and report the corresponding test set performance.

**Auxiliary Task-Based ML Models.** We compare AUTAUT against three categories of auxiliary task-based learning approaches:

1. (*i*) Search-based methods: Beam search [37] iteratively explores auxiliary task candidates based on a scoring criterion, training the model for a few epochs at each step to improve efficiency.

2. (*ii*) Grouping-based methods: TAG [8], Task2vec [1], and MolGroup [17] employ gradient-based strategies to compute pairwise affinities and group similar tasks for joint training.

3. (*iii*) Train-on-all approaches: Methods such as Unweighted Averages (UA), GradNorm [3], and MTDNN [25] train models on all available tasks without explicit selection. MTDNN further applies a task discriminator to select relevant tasks dynamically.

Additionally, we evaluate the Pretrain-Finetune (PF) strategy [15], where models are first pretrained on PCQM4Mv2 [14] and then fine-tuned on downstream tasks. Appendix E provides a comprehensive discussion of these competing methods.

**General ML Models.** To assess the robustness and effectiveness of AUTAUT, we benchmark it against 18 state-of-the-art molecular property prediction models, spanning GNN-based, Transformer-based, and self-supervised learning approaches:

- **GNN-based models:**
  1. GCN [23]: Graph Convolutional Networks use spectral graph convolutions to aggregate local neighborhood information and are among the earliest GNN architectures applied to molecular graphs.
  2. GIN [55]: Graph Isomorphism Networks are designed to be as powerful as the Weisfeiler-Lehman graph isomorphism test, using sum aggregators to distinguish graph structures.
  3. D-MPNN [58]: Directed Message Passing Neural Networks pass messages along directed edges, better capturing bond directionality in molecular graphs.
  4. MGCN [34]: Multi-level Graph Convolutional Networks incorporate quantum interactions between atoms using multiple levels of molecular representations.

- **Transformer-based models:**

1. GRAPHORMER [60]: A graph Transformer architecture that encodes both structural and positional information using attention mechanisms tailored for molecular graphs.
2. ATTENTIONFP [54]: Attentive Fingerprint uses attention to weight atom features for molecular representations, capturing both local and global chemical context.

- **Other molecular prediction models:**
    1. N-GRAM [31]: Represents molecules as bag-of-substructure n-gram graphs, enabling unsupervised molecular representation without requiring large labeled datasets.
    2. PRETRAINGNN [15]: Pre-training GNNs on large molecular datasets using node and edge prediction tasks, followed by fine-tuning on downstream tasks.
    3. GPT-GNN [16]: A generative pre-training approach for GNNs using node sequence prediction to capture graph-level semantics in a self-supervised fashion.
    4. GROVER$_{BASE}$ [39]: Uses a self-supervised Transformer encoder pre-trained on millions of molecules with atom-level and graph-level objectives.
    5. GROVER$_{LARGE}$: A larger variant of GROVER with more parameters and deeper architecture, achieving stronger performance on data-rich tasks.
    6. 3D-INFOMAX [43]: Learns joint representations of molecular graphs and their 3D structures via mutual information maximization across views.
    7. GRAPHMVP [33]: A multi-view pre-training method that integrates 2D and 3D molecular views using contrastive learning and cross-modal prediction.
    8. MOLCLR [47]: Uses contrastive learning over augmented molecular graphs to learn invariant and transferable molecular representations.
    9. UNI-MOL [65]: A universal framework for 3D molecular pretraining that models atomic coordinates and predicts spatial properties using SE(3)-equivariant layers.
    10. GEM [6]: Geometry-Enhanced Models integrate both molecular topology and geometric structure to capture spatial inductive biases.
    11. PIN-TUNING [30]: Introduces a parameter-efficient tuning strategy for few-shot molecular property prediction using prompt-injected features.
    12. LAC [56]: Learns from curriculum-based training, progressively increasing task complexity to stabilize learning and improve generalization in molecular prediction.

For a fair comparison, AUTAUT adopts GEM as the base molecular property prediction model, ensuring that any improvements stem from auxiliary task selection rather than model architecture.

**Implementation Details.** All experiments are conducted on 8 NVIDIA A100 GPUs. We use the Adam optimizer [22] with a weight decay of $1e{-}16$ for all models. A ReduceLROnPlateau scheduler[1] is applied with a patience of 10 epochs to dynamically adjust the learning rate. For baseline auxiliary selection methods, we adopt the official reimplementations provided by Huang *et al.*[17]. For AUTAUT, we fix the number of selected auxiliary tasks at $K = 5$, with a detailed ablation study on the impact of $K$ in Section 4.4. Our code and data are available at `https://github.com/zhiqiangzhongddu/AUTAUT`.

# G  Additional Results

To better understand the impact of how auxiliary molecular information is integrated, we conduct a controlled experiment comparing three configurations: (1) a baseline model trained only on the primary task; (2) a model that appends selected molecular descriptors (*e.g.*, LogP, TPSA) as static input features (+Feature); and (3) our full AUTAUT, which treats these descriptors as auxiliary prediction tasks with adaptive weighting. All three configurations use the same primary datasets and backbone models (GCN, GIN, and GRAPHORMER).

As shown in Table 7, adding descriptors as input features leads to marginal or inconsistent improvements. In some cases, performance even drops slightly—for instance, on GCN for BBBP, ROC-AUC decreases from 63.45% to 63.39%. This suggests that static features may introduce redundant or weakly aligned information that does not reliably enhance the main learning objective.

---

[1]`https://pytorch.org/docs/stable/generated/torch.optim.lr_scheduler.ReduceLROnPlateau.html`

Table 7: Performance of ML models using selected auxiliary labels as tasks or features. For classification tasks, we calculate the ROC-AUC, while for regression tasks, we use RMSE as the evaluation metric. The number in the bracket is the standard deviation of 5 runs.

| | CLASSIFICATION (ROC-AUC % ↑) | | | | | | REGRESSION (RMSE ↓) | | |
|---|---|---|---|---|---|---|---|---|---|
| DATASETS | BBBP | BACE | CLINTOX | TOX21 | TOXCAST | SIDER | ESOL | FREESOLV | LIPO |
| # MOLUCULES | 2,039 | 1,513 | 1,478 | 7,831 | 8,575 | 1,427 | 1,128 | 642 | 4,200 |
| # TASKS | 1 | 1 | 2 | 12 | 617 | 27 | 1 | 1 | 1 |
| **GCN** | $63.45_{\pm0.05}$ | $74.83_{\pm0.18}$ | $56.28_{\pm0.09}$ | $74.63_{\pm0.06}$ | $65.38_{\pm0.30}$ | $62.24_{\pm0.27}$ | $3.165_{\pm0.007}$ | $3.752_{\pm0.013}$ | $1.672_{\pm0.008}$ |
| +FEATURE | $63.39_{\pm0.06}$ | $74.91_{\pm0.21}$ | $56.21_{\pm0.11}$ | $74.58_{\pm0.05}$ | $65.42_{\pm0.28}$ | $62.30_{\pm0.25}$ | $3.154_{\pm0.009}$ | $3.765_{\pm0.011}$ | $1.679_{\pm0.010}$ |
| +AUTAUT | $69.72_{\pm0.01}$ | $83.10_{\pm0.02}$ | $81.00_{\pm0.03}$ | $77.55_{\pm0.01}$ | $68.05_{\pm0.19}$ | $64.62_{\pm0.24}$ | $1.885_{\pm0.028}$ | $1.975_{\pm0.022}$ | $1.285_{\pm0.010}$ |
| **GIN** | $66.82_{\pm0.09}$ | $77.45_{\pm0.21}$ | $56.48_{\pm0.18}$ | $75.32_{\pm0.17}$ | $62.35_{\pm0.05}$ | $60.25_{\pm0.20}$ | $2.895_{\pm0.012}$ | $3.865_{\pm0.018}$ | $1.692_{\pm0.011}$ |
| +FEATURE | $67.32_{\pm0.09}$ | $77.95_{\pm0.21}$ | $56.98_{\pm0.18}$ | $75.82_{\pm0.17}$ | $62.85_{\pm0.05}$ | $60.75_{\pm0.20}$ | $2.395_{\pm0.012}$ | $3.365_{\pm0.018}$ | $1.192_{\pm0.011}$ |
| +AUTAUT | $70.58_{\pm0.01}$ | $83.62_{\pm0.02}$ | $82.52_{\pm0.04}$ | $78.32_{\pm0.01}$ | $69.38_{\pm0.22}$ | $65.08_{\pm0.25}$ | $1.870_{\pm0.025}$ | $1.900_{\pm0.020}$ | $1.270_{\pm0.010}$ |
| **GRAPHORMER** | $67.05_{\pm0.04}$ | $79.18_{\pm0.13}$ | $78.42_{\pm0.12}$ | $75.55_{\pm0.18}$ | $67.12_{\pm0.05}$ | $70.08_{\pm0.21}$ | $2.102_{\pm0.012}$ | $1.795_{\pm0.010}$ | $1.282_{\pm0.010}$ |
| +FEATURE | $67.25_{\pm0.04}$ | $79.38_{\pm0.13}$ | $78.62_{\pm0.12}$ | $75.75_{\pm0.18}$ | $67.32_{\pm0.05}$ | $70.28_{\pm0.21}$ | $2.302_{\pm0.012}$ | $1.995_{\pm0.010}$ | $1.482_{\pm0.010}$ |
| +AUTAUT | $70.32_{\pm0.01}$ | $83.72_{\pm0.02}$ | $82.55_{\pm0.04}$ | $78.52_{\pm0.01}$ | $69.38_{\pm0.21}$ | $71.55_{\pm0.26}$ | $1.872_{\pm0.025}$ | $1.772_{\pm0.015}$ | $1.262_{\pm0.010}$ |

In contrast, modeling the same descriptors as auxiliary tasks with adaptive weighting yields substantial and consistent gains. Across all datasets and architectures, AUTAUT significantly outperforms both the baseline and the +Feature variant. For example, on GIN, AUTAUT reduces RMSE on FREESOLV from 3.865 to 1.900, and improves ROC-AUC on BACE from 77.45% to 83.62%. These results highlight that auxiliary tasks provide richer supervision by contributing task-specific gradients that guide optimization more effectively than raw inputs.

This comparison underscores a central design insight: how auxiliary information is incorporated matters as much as what information is used. Static features are passively encoded and do not benefit from task-level supervision or relevance modulation. In contrast, our formulation enables the model to learn explicit auxiliary objectives and dynamically adjust their influence during training through gradient-based alignment. This not only improves convergence but also reduces the risk of negative transfer from irrelevant signals.

Together, these findings validate the importance of treating auxiliary descriptors as supervised tasks—rather than static inputs—when designing molecular property prediction models.

