# OpenReview forum: "Automatic Auxiliary Task Selection and Adaptive Weighting Boost Molecular Property Prediction"
_NeurIPS.cc/2025/Conference — NeurIPS 2025 poster_

### Official Review · Reviewer_D2QM · 2025-06-29

**Clarity:** 3
**Significance:** 3
**Originality:** 3
**Rating:** 3
**Confidence:** 3

**Summary:**

The paper targets the problem of molecular property prediction, especially when labeled data is scarce. It proposes Automatic Auxiliary Task Selection (AUTAUT) to automatically select auxiliary tasks and adaptively weight them to boost the performance of molecular property prediction. Experiments on MoleculeNet benchmarks show the effectiveness of the proposed method.

**Questions:**

Please see pros & cons

**Ethical Concerns:**

["NO or VERY MINOR ethics concerns only"]

**Limitations:**

yes

**Quality:**

3

**Strengths And Weaknesses:**

pros:
*  Traditional methods use auxiliary learning tasks to supplement main tasks, improving performance when labeled molecular data is limited. However, selecting and integrating effective auxiliary tasks requires domain expertise, manual effort, and does not always guarantee better results. This work applies large language models to automatically select auxiliary tasks, which is novel and effective.
* Experiments on MoleculeNet benchmarks show the effectiveness of the proposed method.
* This work utilizes a multi-step LLM prompting process to assess and select the most relevant auxiliary tasks for a given primary molecular prediction goal.
* This work introduces a novel gradient alignment weighting mechanism that dynamically adjusts the influence of each auxiliary task during training, amplifying tasks aligned with the primary objective and minimizing negative interference.

cons:
* Why not directly use auxiliary task label as model input? It seems using auxiliary task label as model input is more straightforward and effective.

---

> ### Author Rebuttal · Authors · 2025-07-30
>
> We sincerely thank Reviewer D2QM for the constructive comments and recognition of the novelty and effectiveness of our proposed approach, AUTAUT, in molecular property prediction tasks. We appreciate the positive remarks on our use of LLMs and the introduction of a novel gradient alignment weighting mechanism. Below, we carefully address the specific concern raised by the reviewer.
>
> > Why not directly use auxiliary task labels as model input? It seems using auxiliary task labels as model input is more straightforward and effective.
>
> We appreciate this insightful question and acknowledge that directly including auxiliary task labels as input features might appear straightforward. We also explored this solution in the early stages of our project. However, empirical evidence from our extensive ablation studies (Section 4.4, Table 6, Appendix G) shows that directly integrating molecular descriptors as static input features results in marginal or inconsistent improvements. For example, using Graphormer, the ROC-AUC on BACE slightly improves from 79.18% to 79.38% when auxiliary labels are added as features, which is minimal compared to the significant boost to 83.72% obtained by our proposed adaptive weighting of auxiliary tasks.
>
> This observation arises because static inclusion of features does not benefit from the explicit task-level supervision provided by treating these descriptors as auxiliary prediction tasks. Moreover, static inputs do not dynamically adapt during training and thus fail to leverage gradient-based information alignment with the primary task. In contrast, modeling descriptors as auxiliary tasks enables explicit supervision signals and adaptive weighting based on gradient alignment, significantly enhancing predictive performance and preventing negative transfer.
>
> Thus, our adaptive weighting strategy ensures the auxiliary tasks positively contribute to the primary objective, dynamically filtering irrelevant or conflicting signals. This distinction is central to the effectiveness and robustness demonstrated by AUTAUT across multiple datasets and models.
>
> ---
>
> Since this was the only concern raised, and the relevant details and evidence were already included in the original manuscript, we kindly ask Reviewer D2QM to reconsider updating the rating in light of this clarification. We are happy to provide any further clarification if required.

---

> > ### Comment · Reviewer_D2QM · 2025-08-01
> > **question**
> >
> > Thank you for your detailed response! I still have a few more questions.
> > Incorporating molecular descriptors as static input features retains all their information, whereas using descriptors as auxiliary prediction tasks seemingly utilizes the information only partially. Why can “auxiliary prediction task training” outperform “using molecular descriptors as static input features”? Could you provide some intuition?

---

> ### Author Response · Authors · 2025-08-01
> **Response to Reviewer D2QM**
>
> Thank you for raising this interesting question.
>
> **In a nutshell**, (1) directly incorporating molecular descriptors in datasets with handcrafted features does not fully preserve descriptor information, (2) the inclusion of features uncorrelated with the specific tasks may introduce noise, (3) auxiliary tasks enhance the predictions by including additional supervision signals.
>
> **In more detail**:
>
> 1. **Static features do not fully preserve descriptor information.**
>    While it might seem that concatenating molecular descriptors as static input features retains all their information, in practice, this is not the case. Most datasets _already contain nine handcrafted features_ alongside learned structural representations. When descriptors are merged as additional inputs, the model learns a joint weight matrix that assigns weights across all features. As a result, only a portion of each descriptor’s signal may be effectively utilized, with the remainder often diluted or ignored during training.
>
> 2. **Direct feature inclusion may introduce task-irrelevant noise.**
>    Many molecular descriptors have only weak or inconsistent correlations with specific prediction tasks (e.g., toxicity or solubility). When added as static features, these irrelevant or weakly relevant descriptors can introduce noise, which may even degrade performance, a trend we observe in Table 6 and supported by recent studies [1-4].
>
> 3. **Auxiliary prediction tasks provide explicit, adaptive supervision.**
>    By contrast, modeling descriptors as auxiliary prediction tasks supplies the model with structured, task-specific supervision signals. Each auxiliary task generates its own loss gradient, steering the learned representations toward capturing information useful for both the auxiliary and primary objectives. Our framework further improves upon this by dynamically weighting each auxiliary task according to its gradient alignment with the main task (Theorem 1, Eq. 4). This ensures that only tasks providing helpful, aligned information are amplified during training, while less relevant or conflicting tasks are down-weighted. In effect, auxiliary tasks serve as "adaptive teachers," helping the model focus on features that truly benefit the primary task and suppressing noise.
>
> These insights are supported by both our empirical results and by related work in the literature, which consistently finds that auxiliary-task training provides superior or more robust gains compared to simple feature concatenation [1-4].
>
> Thank you again for your thoughtful question. We hope this clarifies our design rationale, and we are happy to provide further discussion if needed.
>
> [1] Enhancing molecular property prediction with auxiliary learning and task‑specific adaptation, Journal of Cheminformatics, 2024.
> (*Auxiliary-task learning improves ROC-AUC by 5–8% over feature concatenation.*)
> [2] Molecular property prediction in the ultra‐low data regime, Communications Chemistry, 2025.
> (*Reports that using molecular descriptors as static features can lead to negative transfer in multi-task learning, while auxiliary-task adaptation with dynamic weighting is more robust and effective in low-data settings.*)
> [3] PEMP: Leveraging Physics Properties to Enhance Molecular Property Prediction, ACM International Conference on Information and Knowledge Management, 2022.
> (*Shows that adding physical descriptors as auxiliary prediction tasks significantly improves model performance and generalization, compared to using them as input features.*)
> [4] Molecular Descriptors Property Prediction Using Transformer-Based Approach, International Journal of Molecular Sciences, 2024.
> (*Demonstrates that treating molecular descriptors as auxiliary tasks within transformer models leads to more stable training and higher accuracy than feature concatenation, especially when training data is limited.*)

---

> > ### Author Response · Authors · 2025-08-04
> >
> > We believe that we have addressed all of your concerns regarding our submission. As the discussion period is closing soon, we would highly appreciate any further feedback at this stage, as this would allow us to address any remaining issues in a timely manner.
> >
> > We look forward to hearing from you. Thank you again for your time and constructive comments.

---

> ### Author Response · Authors · 2025-08-06
>
> Dear Reviewer,
>
> It has been 5 days since our last response, and as the reviewer-author discussion phase is closing soon, we wanted to gently check in. If you have any further questions or comments, please let us know. We would greatly appreciate the opportunity to address any remaining issues in a timely manner.
>
> Thank you again for your time and consideration.

---

> ### Author Response · Authors · 2025-08-08
> **Additional Evidence and Clarifications for Reviewer D2QM - 1**
>
> **Additional Evidence and Intuition**
>
> **1. Training dynamics: why auxiliary tasks help more than static features.**
> To further illustrate why modeling descriptors as auxiliary tasks can outperform using them as static features, we present training-loss trajectories for the BACE and ClinTox datasets under three settings:
> (1) no auxiliary tasks,
> (2) descriptors as static input features, and
> (3) descriptors as auxiliary tasks with adaptive weighting (AUTAUT).
>
> **Key observation:**
> Across both datasets, static-feature curves are nearly identical to (or slightly worse than) the no-auxiliary baseline, while AUTAUT converges faster and achieves markedly lower final loss, consistent with its higher ROC-AUC.
>
> ---
>
> **BACE Dataset (ROC-AUC: No Aux 74.73, Input Feature 74.79, AUTAUT 83.12)**
>
> | Epoch | No Aux. Task | Input Feature | AUTAUT   |
> | ----- | ------------ | ------------- | -------- |
> | 1     | 2.492795     | 2.490306      | 2.407118 |
> | 10    | 2.250155     | 2.247905      | 2.165342 |
> | 20    | 1.507204     | 1.502698      | 1.420544 |
> | 30    | 1.306099     | 1.302181      | 1.195040 |
> | 40    | 1.103621     | 1.100910      | 1.017448 |
> | 50    | 0.729232     | 0.727804      | 0.684153 |
> | 60    | 0.627123     | 0.625496      | 0.527153 |
> | 70    | 0.497315     | 0.495823      | 0.415335 |
> | 80    | 0.297848     | 0.297253      | 0.214357 |
> | 90    | 0.215429     | 0.214808      | 0.121953 |
> | 100   | 0.219757     | 0.219537      | 0.122788 |
> | 110   | 0.264828     | 0.264166      | 0.139921 |
> | 120   | 0.237989     | 0.237751      | 0.128290 |
> | 130   | 0.209190     | 0.208981      | 0.115875 |
> | 140   | 0.165256     | 0.164891      | 0.097406 |
> | 150   | 0.149947     | 0.149797      | 0.090387 |
> | 160   | 0.211504     | 0.210892      | 0.114114 |
> | 170   | 0.263249     | 0.262462      | 0.133917 |
> | 180   | 0.198285     | 0.197891      | 0.107035 |
> | 190   | 0.189394     | 0.188820      | 0.102584 |
> | 200   | 0.219806     | 0.219586      | 0.113853 |
> | 210   | 0.119048     | 0.118929      | 0.072654 |
> | 220   | 0.174618     | 0.174444      | 0.093986 |
> | 230   | 0.213258     | 0.212832      | 0.108547 |
> | 240   | 0.230917     | 0.230455      | 0.114715 |
> | 250   | 0.173724     | 0.173376      | 0.090942 |
> | 260   | 0.203092     | 0.202486      | 0.101794 |
> | 270   | 0.195181     | 0.194592      | 0.097734 |
> | 280   | 0.220393     | 0.219733      | 0.106924 |
> | 290   | 0.124124     | 0.123876      | 0.067521 |
> | 300   | 0.206705     | 0.206285      | 0.099657 |
>
> ---
>
> **ClinTox Dataset (ROC-AUC: No Aux 56.29, Input Feature 56.21, AUTAUT 80.92)**
>
> | Epoch | No Aux. Task | Input Feature | AUTAUT   |
> | ----- | ------------ | ------------- | -------- |
> | 1     | 5.116870     | 5.119451      | 5.102034 |
> | 10    | 4.759021     | 4.761899      | 4.875935 |
> | 20    | 3.768138     | 3.770884      | 3.445754 |
> | 30    | 2.733912     | 2.736646      | 2.367761 |
> | 40    | 1.515509     | 1.518031      | 1.244938 |
> | 50    | 0.945310     | 0.947010      | 0.677387 |
> | 60    | 0.936845     | 0.938529      | 0.526620 |
> | 70    | 0.856020     | 0.857789      | 0.486200 |
> | 80    | 0.749437     | 0.751036      | 0.504105 |
> | 90    | 0.801679     | 0.803284      | 0.478655 |
> | 100   | 0.679061     | 0.680534      | 0.401384 |
> | 110   | 0.626799     | 0.628272      | 0.368379 |
> | 120   | 0.644937     | 0.646352      | 0.343168 |
> | 130   | 0.493670     | 0.494785      | 0.333210 |
> | 140   | 0.506356     | 0.507469      | 0.335651 |
> | 150   | 0.595040     | 0.596205      | 0.350179 |
> | 160   | 0.582802     | 0.584050      | 0.351743 |
> | 170   | 0.677697     | 0.678980      | 0.354124 |
> | 180   | 0.582438     | 0.583683      | 0.342028 |
> | 190   | 0.564039     | 0.565325      | 0.373568 |
> | 200   | 0.740695     | 0.742030      | 0.350339 |
> | 210   | 0.600196     | 0.601495      | 0.341287 |
> | 220   | 0.619227     | 0.620550      | 0.304817 |
> | 230   | 0.513168     | 0.514342      | 0.329035 |
> | 240   | 0.573193     | 0.574498      | 0.319584 |
> | 250   | 0.630665     | 0.632013      | 0.328793 |
> | 260   | 0.557530     | 0.558870      | 0.330512 |
> | 270   | 0.650799     | 0.652131      | 0.342014 |
> | 280   | 0.571755     | 0.573070      | 0.311787 |
> | 290   | 0.599481     | 0.600780      | 0.324066 |
> | 300   | 0.568881     | 0.570130      | 0.308966 |
>
> We will convert the above information into a figure like Figure 2 in the revised manuscript.

---

> ### Author Response · Authors · 2025-08-08
> **Additional Evidence and Clarifications for Reviewer D2QM - 2**
>
> **2. Representation analysis.**
> t-SNE projections of the learned molecule embeddings show that auxiliary-task training yields more distinct class clusters and clearer label separation, whereas the static-feature model produces more diffuse, overlapping clusters. This suggests auxiliary supervision actively structures the latent space to align with both auxiliary and main tasks. (Visualizations will be included in the revised manuscript.)
>
> **3. Intuition — “active teaching” vs. “passive hinting”.**
> Static features act as passive hints: they flow only through the main-task loss, so irrelevant components can persist. Auxiliary prediction tasks act as active teachers:
>
> * They produce their own gradients, shaping the encoder to capture descriptor-relevant structure.
> * Gradient-alignment weighting amplifies only components that help the main task and suppresses conflicting ones.
>   Thus, even if only part of the descriptor information is directly used, that part is precisely the most relevant to the main objective.
>
> **4. Literature support.**
> These trends are consistent with prior findings:
>
> * \[1], \[3] report +5–8% ROC-AUC gains from auxiliary-task learning over feature concatenation.
> * \[2] shows that static descriptors can cause negative transfer in low-data regimes, while adaptive auxiliary-task weighting avoids this.
> * \[4] finds that predicting descriptors in Transformer models yields more stable training and higher accuracy than adding them as features, especially with limited data.
>
> ***Unlike \[1–4], AUTAUT is fully automated and requires no manual task selection or domain expertise.***
>
> **5. Ablation evidence (Table 6).**
> For example, on BACE with Graphormer, static features raise ROC-AUC only from 79.18% → 79.38%, whereas auxiliary-task training raises it to 83.72%. Similar patterns hold across datasets and backbones.
>
> ---
>
> [1] Enhancing molecular property prediction with auxiliary learning and task‑specific adaptation, Journal of Cheminformatics, 2024.
> [2] Molecular property prediction in the ultra‐low data regime, Communications Chemistry, 2025.
> [3] PEMP: Leveraging Physics Properties to Enhance Molecular Property Prediction, ACM International Conference on Information and Knowledge Management, 2022.
> [4] Molecular Descriptors Property Prediction Using Transformer-Based Approach, International Journal of Molecular Sciences, 2024.

---

### Official Review · Reviewer_8GVh · 2025-06-30

**Clarity:** 3
**Significance:** 3
**Originality:** 3
**Rating:** 4
**Confidence:** 4

**Summary:**

The article presents AutAuT, an automation framework for improving the prediction of molecular properties. AutAuT uses large language models (LLM) for extraction, selection and dynamic integration of auxiliary tasks. The key contribution is full automation of the process (from task extraction to adaptive weighting of their contribution) and gradient alignment using Fischer minimization to reduce the negative impact of irrelevant tasks. Experiments on 9 datasets showed superiority over 10 methods based on auxiliary tasks and 18 state-of-the-art models.

**Questions:**

1. Due to the absence of certain files and directories (`data/ogbg_molbace/mapping/mol.csv.gz`, also, due to the absence of the `generate_description`, `generate_compressor_message`, `query_chatgpt` sections in the code directory and dataset in the shared directory), the following commands are not working:

- `python -m code.generate_description base_demo_test`
- `python -m code.generate_compressor_message base_demo_test`
- `python -m code.query_chatgpt base ogbg_molbace`
- `python -m main_GN base ogg_molbace scratch n.model.name`

Could you please attach the GitHub code for these files?

2. Regarding the prompts given in Appendix A.1 and A.2, were these used for experiments on all databases, or are they just examples? If they are just examples, please provide the prompts that were actually used for your experiments.

3. The method is positioned as a fully automated framework that searches for auxiliary tasks _"from domain-specific databases or online resources"_ (page 2, line 28), but experiments use only descriptors from Table 4. Nowhere is it said on what principle the list of descriptors in this table was formed. How is it formed and would the method work with the same accuracy if it searched for properties from the Internet?

4. Table 5 shows that lipophilicity and molecular weight of the molecule contribute to all sets. Would it be possible to exclude them from the task search and look to select 3 tasks instead of 5? In practice, such a configuration could reduce computational costs significantly. Is this possible in the current setup?

5. On page 4 lines 27-28 say, _“By following these steps, AutAuT enables molecular toxicity prediction with auxiliary task enhancement, without requiring manual efforts.”_ The authors seem to test the framework on databases with known toxicity, but in no way generalize the findings to other databases. Could you comment on that?

6. Taking a closer look at Table 5, I notice that many of the properties appear selected by virtually all LLMs in all datasets (e.g., Topological Polar Surface Area, Number of Hydrogen Bond Donors, and others). It is, therefore, uninformative to report the selection of those properties as a result. On the other hand, some properties appear selected at random – by some LLMs and not frequently (e.g., Radius of Gyration and Eccentricity). It is, therefore, unclear how important the prediction of such properties as an auxiliary task really is. Can you provide a table with high specificity properties selected as auxiliary tasks for each dataset along with the information on consistency of the selection and the measured impact on the prediction performance of the target property?

7. Can you provide some evidence and comment on the consistency of the auxiliary-task weights? Is it possible to compile a table with means and standard deviations of the converged weights for at least some datasets? In my understanding, this would support the overwhelming dominance of your method shown on Table 3.

8. Have you experimented with any of the Small Language Models?

9. Do you plan to release an open-source package or toolkit?

**Ethical Concerns:**

["NO or VERY MINOR ethics concerns only"]

**Final Justification:**

The authors have made a significant effort in addressing my questions and concerns, which I appreciate. This work has many strengths already outlined. Despite the reproducibility issues, I trust that the authors will address them in the camera-ready submission, as they promised. I would like to keep my positive evaluation.

**Limitations:**

The limitations are briefly discussed in Appendix H.

**Quality:**

3

**Strengths And Weaknesses:**

__Strengths:__

1. It is a very interesting topic and a potentially impactful contribution. AutAuT eliminates manual task selection by using LLM to extract and select tasks based on their relevance to the primary goal.
2. The authors, with rare exceptions, focus on articles no older than 5 years, which indicates the actuality and relevance of their work.
3. A detailed mathematical proof of the algorithm's operation has been carried out: dynamic weighting of gradients adaptively enhances the contribution of tasks aligned with the main goal and suppresses noise from irrelevant tasks.
4. The analysis of the computational complexity of the algorithm and its cost is carried out, which contributes to the transparency of the application of this algorithm.
5. Superiority over 28 methods in accuracy on 9 datasets was shown. Ablation studies have confirmed the importance of adaptive weighting and task selection.
6. The paper is very well organized and structured. I found answers to many of my questions either in the main text or in the Appendix.

__Weaknesses:__
1. A significant disadvantage of the work is the lack of reproducibility. The GitHub repo does not have the files necessary to run the code. The questions section clarifies which files are missing.
2. No experiments were conducted with searching for additional tasks on the Internet, they were selected by LLM from Table 4 (see question 4).
3. No open-source resource is advertised or announced in the paper, which limits the wide adoption of the proposed framework.

---

> ### Author Rebuttal · Authors · 2025-07-30
>
> We sincerely appreciate the thorough and constructive feedback provided by Reviewer 8GVh. Below, we systematically address each of your concerns and questions.
>
> > The GitHub repo does not have the files necessary to run the code.
>
> > Do you plan to release an open-source package or toolkit?
>
> We apologize for this omission. Due to an oversight, the anonymized repository was not kept fully synchronized with the active development repository, resulting in the absence of several utility scripts and data mapping files. As this year’s rebuttal process prohibits any addition or update to the repository, we are unable to upload the missing files at this stage. Upon acceptance, we will ensure that the complete codebase, all required files, and detailed instructions are made publicly available, so that the experiments can be reproduced in full.
>
> The data files, such as data/ogbg_molbace/mapping/mol.csv.gz, are automatically downloaded by the code when the corresponding dataset is accessed.
>
> > Regarding the prompts given in Appendix A.1 and A.2, were these used for experiments on all databases, or are they just examples?
>
> The prompts provided in Appendix A.1 and A.2 are precisely those used for auxiliary task retrieval and selection in all our experiments.
>
> > The method is positioned as a fully automated framework that searches for auxiliary tasks… but experiments use only descriptors from Table 4.
>
> The computable molecular properties in Table 4 are retrieved from the internet using LLMs. As detailed in Section 3.3 and Appendix A.1-A.2, the automated auxiliary task retrieval process begins with a standard prompt (“Find all available computable molecular properties.”), where LLMs are allowed to freely search molecular properties from relevant databases or online resources. There are no restrictions placed on this search process.
>
> For benchmarking and reproducibility, we focused on properties that are supported by RDKit, as these are both widely used in the literature and guaranteed to be computable for arbitrary molecules [1,2,3]. This ensures all auxiliary tasks can be implemented reliably in practice. Table 4 presents the full list of properties obtained through this automated, internet-based, retrieval process and their brief descriptions.
>
> Yet the framework is not inherently constrained by the use of any specific toolkit (e.g., RDKit). If additional computable properties are found on the internet, the method can incorporate them as well, provided they are compatible with the task setup. However, properties that are not computable for arbitrary molecules cannot be used as auxiliary task labels in this context.
>
> We hope this clarifies the automated, internet-based, nature of our auxiliary task retrieval process. We are happy to provide further details if needed.
>
> > Table 5 shows that lipophilicity and molecular weight of the molecule contribute to all sets. Would it be possible to exclude them from the task search.
>
> We have investigated the effect of varying the number of auxiliary tasks in our experiments. As illustrated in Figure 3b, reducing the number of selected auxiliary tasks from 5 to 3 does lead to a modest decrease in performance. However, the drop is relatively minor and may be acceptable in scenarios where computational resources are limited. The framework is flexible: users can adjust the number and selection of auxiliary tasks to balance accuracy and efficiency according to their specific requirements.
>
> Excluding specific properties, such as lipophilicity and molecular weight, is also supported in our setup. This can be achieved either by filtering these properties out from Table 4 or by modifying the LLM prompts to exclude them during auxiliary task selection.
>
> > On page 4 lines 27-28 say, “...” The authors seem to test the framework on databases with known toxicity, but in no way generalize the findings to other databases.
>
> The statement on page 4, lines 27-28, appears in the framework overview section, where molecular toxicity prediction is used as a representative example to illustrate the workflow (see lines 120-121). This _should not be interpreted as a limitation to toxicity prediction tasks_. In fact, _our evaluation covers 9 diverse datasets_ encompassing a broad range of molecular property prediction tasks. For example, the ESOL dataset is focused on predicting water solubility for small organic molecules; the SIDER dataset involves classification of adverse drug reactions grouped by organ system; and the BACE dataset concerns quantitative and qualitative binding affinity predictions for inhibitors of human β-secretase 1 (BACE-1). A full description of all datasets and their corresponding prediction objectives is provided in Appendix F.
>
> > Taking a closer look at Table 5, I notice that many of the properties appear selected by virtually all LLMs in all datasets (...). It is, therefore, uninformative to report the selection of those properties as a result.
>
> We report the selected auxiliary tasks in Table 5 to provide transparency regarding which properties are consistently chosen by AUTAUT across experiments. The frequent selection of descriptors such as LogP, TPSA, and Number of Hydrogen Bond Donors is scientifically meaningful, as these properties are well established in the chemical literature to be highly relevant for a wide range of molecular prediction tasks. This pattern not only validates the chemical soundness of the LLM-driven selection process but also demonstrates that AUTAUT is able to recover domain-relevant knowledge in a fully automated way.
>
> We report the actual selected tasks to provide clear evidence about what the framework learns and helps validate the scientific relevance of the output.
>
> >  Can you provide a table with high specificity properties selected as auxiliary tasks for each dataset along with the information on consistency of the selection and the measured impact on the prediction performance of the target property?
>
> We are not entirely clear on the motivation for emphasizing “high specificity properties” or the precise analysis expected for these properties. We would appreciate it if the reviewer could further explain what is meant by “a table with high specificity properties selected as auxiliary tasks for each dataset, along with the information on consistency of the selection and the measured impact on the prediction performance of the target property.” Additional details or an example would be helpful for us to address this request more precisely.
>
> Some information that might be relevant is that Table 5 reports the number of times each auxiliary task was selected across 5 runs (see the number within round brackets), and Appendix A.3 discusses the behavior of different LLMs in auxiliary task selection.
>
> > On the other hand, some properties appear selected at random - by some LLMs and not frequently.
>
> Figure 2 illustrates the evolution of auxiliary task weights during training for BACE and ClinTox. This shows that AUTAUT adaptively adjusts the influence of each auxiliary task according to its alignment with the main objective, leading to improved convergence and reduced training loss compared to baselines without adaptive weighting. As a result, even auxiliary tasks selected with lower frequency are dynamically weighted by their actual utility during training.
>
> > Can you provide some evidence and comment on the consistency of the auxiliary-task weights?
>
> Figure 2 presents the evolution of auxiliary-task weights (left axes) and training-loss trajectories (right axes) for the BACE and ClinTox datasets. These results demonstrate that the auxiliary-task weights converge consistently across training runs and that the weighting mechanism is stable. The weights increase for auxiliary tasks that align well with the primary objective, while less relevant tasks are automatically down-weighted. This adaptive behavior contributes to both improved convergence and robust performance, as shown in Table 3.
>
> > Have you experimented with any of the Small Language Models?
>
> We have experimented with several LLMs via API and local deployment. As discussed in Appendix A, our cost analysis indicates that this configuration remains practical for most applications. Based on these results, and given the stable performance achieved with these models, we did not extend our experiments to even smaller language models. However, our framework is model-agnostic and can be readily adapted for use with smaller models if desired.
>
> [1] Strategies for Pre-training Graph Neural Networks, ICLR.
> [2] Self-Supervised Graph Transformer on Large-Scale Molecular Data, NeurIPS.
> [3] Physical Consistency Bridges Heterogeneous Data in Molecular Multi-Task Learning, NeurIPS.

---

> > ### Comment · Reviewer_8GVh · 2025-08-02
> >
> > Thank you to the authors for the answers to my questions and concerns.
> > I would like to make a few additional points as a follow up.
> >
> > 1. It is unfortunate that updates to the code repository are prohibited this year. While the experimental results appear solid in the paper, I find it difficult to objectively assess the reproducibility of the findings without full access to the codebase.
> >
> > 2. I asked about the prompts in Appendix A.1 and A.2, because it is hard for me to believe that these exact prompts deliver top performance of the proposed method. Additionally, these "prompts" seemingly include the responses of the LLMs, which is very confusing (why reporting LLM responses as prompts?). In my understanding, prompt and context engineering are key aspects of LLM-based solutions. It is, therefore, crucial to tailor those to the application to ensure top performance. Instead, the authors write that _"even basic prompts can generate meaningful and relevant auxiliary tasks"_. This does not convince me.
> >
> > 3. I would like to reformulate question #5, because the authors might have misunderstood it. Can you report and discuss any general finding produced by your method related to toxicity prediction, since it showed such superior performance to virtually all methods across all datasets?
> >
> > 4. I would like to further clarify my question regarding high specificity properties selected as auxiliary tasks. In order to claim that your method can select auxiliary tasks delivering top performance in target prediction, you should be able to demonstrate that useful tasks are consistently selected and useless tasks are consistently ignored. Table 5 shows many examples of auxiliary tasks selected for almost all models and all datasets. This might speak to the consistency of selection of the useful tasks, but might also indicate the strong selection bias. Can you point me to the evidence of useless tasks being consistently ignored?
> >
> > 5. My question regarding consistency of the auxiliary-task weights arose from the assumption that Figure 2 shows two cherry-picked example (perhaps, well representative ones, but still). I was wondering if the authors also did some some statistical evaluations. Please clarify.
> >
> > 6. My question regarding Small Language Models was driven by sloppy references of the models used. Which versions of the three LLMs have you used in experiments? Assuming those were indeed large models, have you tried using smaller versions (e.g., Llama-3.1-8B)? Ultimately, have you tested the method with newer models of the same providers?

---

> ### Author Response · Authors · 2025-08-04
> **Response to Reviewer 8GVh - 1**
>
> **Q1.**
> While we are unable to update the official code repository during the review process, we are fully committed to transparency. In this round of reply, we include below a compact version of the auxiliary task selection code (requiring only `Python` and the `transformers` package), as well as several complete auxiliary task selection conversations, addressing your concerns Q2 and Q6 as well. If there is any further specific information you need for verification, we are happy to provide it promptly.
>
> **Q2.**
> We fully agree that prompt design is critical in many LLM applications. Surprisingly, though,  in our experiments, even relatively simple prompts were able to generate highly relevant and chemically meaningful auxiliary tasks. This suggests that the domain priors within state-of-the-art LLMs are already strong, which may explain our method’s robustness and strong empirical results. We appreciate your suggestion and now provide a complete example LLM conversation in the next box for full transparency. We hope this helps address your skepticism and enhances the credibility of our findings.
>
> We presented both the prompts and the LLM together to offer a comprehensive view, an approach also adopted in recent influential works [1,2,3,4,5].
>
> **Q3**
> Across the datasets involving toxicity prediction (BBBP, TOX21, TOXCAST, CLINTOX, SIDER), we observed several consistent findings:
>
> - **Consistent Auxiliary Task Selection:** The LLMs consistently selected physicochemical properties (e.g., LogP, TPSA, hydrogen bond donors/acceptors) as most relevant for toxicity, echoing well-established chemical knowledge about toxicophores and molecular transport.
> - **Performance Correlates with Task Alignment:** The greatest gains over baselines occurred when the selected auxiliary tasks had strong theoretical links to the primary endpoint (e.g., ADME properties for toxicity).
> - **Adaptive Weighting:** The adaptive weighting mechanism regularly up-weighted descriptors that literature identifies as toxicologically meaningful, and down-weighted less relevant ones, leading to both accuracy and robustness gains.
>
> These general findings suggest that our framework not only automates the selection of relevant auxiliary tasks but also successfully identifies domain-relevant predictors for toxicity, which likely explains the strong empirical results.
>
> **Q4**
> We agree that a scientifically valid auxiliary task selection method should not only prioritize useful properties but also avoid selecting descriptors that lack demonstrated relevance to the target property.
>
> In our experiments, properties such as Eccentricity, Asphericity, Inertial Shape Factor, and Radius of Gyration—being geometric or topological descriptors—are rarely or never selected.
> This pattern is consistent with the cheminformatics literature: large-scale benchmarking studies have shown that _physicochemical descriptors_ (e.g., LogP, Topological Polar Surface Area, hydrogen bond counts) have strong predictive value for molecular property prediction tasks such as toxicity, solubility, or drug-likeness, while _geometric descriptors_ play little to no role [6]. Best practices in QSAR modeling likewise emphasize that physicochemical descriptors are the most successful features for ADMET prediction, and geometric measures generally do not improve model performance [7]. Comprehensive surveys of QSAR methods further confirm that the most predictive models for biological activity consistently rely on physicochemical and topological features, with geometric descriptors such as radius of gyration or asphericity seldomly used [8].
>
> Thus, the fact that our method does not select these geometric/topological properties demonstrates alignment with established chemical understanding, rather than being the result of selection bias.
>
> **Q5**
> To address concerns about the representativeness of Figure 2, we systematically evaluated auxiliary-task selection and learned weights across five independent runs for each dataset and LLM. To further quantify this relationship, we computed the Spearman rank correlation between the frequency with which an auxiliary task is selected and its average adaptive weight. We found a strong positive correlation (ρ = 0.81, p < 0.01) across all datasets and LLMs, demonstrating that tasks chosen more frequently are also assigned higher weights by the model. This relationship is robust and statistically significant, supporting the conclusion that our framework reliably prioritizes functionally meaningful auxiliary tasks.
>
> The table below summarizes the auxiliary tasks selected, where the numbers in parentheses (k, w) indicate the number of times each task was chosen out of five repeats (k), and its average learned weight (w).

---

> ### Author Response · Authors · 2025-08-04
> **Response to Reviewer 8GVh - 2**
>
> | Dataset | LLM | LLM Selected Auxiliary Tasks |
> | ----- | ----- | ----- |
> | BBBP | GPT-4 | Topological Polar Surface Area (5, 0.28), LogP (5, 0.27), Number of Hydrogen Bond Donors (5, 0.24), Molecular Weight (5, 0.21), QED (5, 0.20) |
> | | Llama 3 | Topological Polar Surface Area (5, 0.26), LogP (5, 0.25), Number of Hydrogen Bond Donors (5, 0.21), Balaban J Index (4, 0.13), NO Group Count (3, 0.11), QED (2, 0.09), Molecular Weight (1, 0.04) |
> | | Gemini 2.0 | Topological Polar Surface Area (5, 0.27), LogP (5, 0.26), Number of Hydrogen Bond Donors (5, 0.23), Molecular Weight (4, 0.13), QED (3, 0.10), Balaban J Index (2, 0.07), Heavy Atom Molecular Weight (1, 0.03) |
> | BACE | GPT-4 | LogP (5, 0.291), Topological Polar Surface Area (5, 0.259), Number of Hydrogen Bond Donors (5, 0.229), Molar Refractivity (5, 0.219), QED (5, 0.151) |
> | | Llama 3 | LogP (5, 0.26), Topological Polar Surface Area (5, 0.25), Molar Refractivity (5, 0.22), Heavy Atom Molecular Weight (3, 0.13), EState VSA1 (2, 0.08), Number of Hydrogen Bond Donors (2, 0.07), QED (1, 0.03) |
> | | Gemini 2.0 | LogP (5, 0.27), Topological Polar Surface Area (5, 0.26), Number of Hydrogen Bond Donors (4, 0.17), Molar Refractivity (4, 0.15), QED (3, 0.09), Total Formal Charge (1, 0.03), NO Group Count (1, 0.02) |
> | ClinTox | GPT-4 | LogP (5, 0.277), Topological Polar Surface Area (5, 0.242), Number of Hydrogen Bond Donors (5, 0.243), QED (3, 0.190), Bertz Complexity (3, 0.150), Fraction of Csp3 Carbon (2, 0.110), Molecular Weight (1, 0.08), Number of Aromatic Rings (1, 0.09) |
> | | Llama 3 | LogP (5, 0.25), Topological Polar Surface Area (5, 0.23), Bertz Complexity (4, 0.13), NHOH Group Count (3, 0.11), Balaban J Index (3, 0.10), Number of Hydrogen Bond Donors (2, 0.09), QED (2, 0.09) |
> | | Gemini 2.0 | LogP (5, 0.26), Topological Polar Surface Area (4, 0.15), QED (3, 0.12), Bertz Complexity (3, 0.10), Molecular Weight (2, 0.08), Aliphatic Rings Count (2, 0.08), NHOH Group Count (1, 0.04) |
> | Tox21 | GPT-4 | LogP (5, 0.23), Topological Polar Surface Area (5, 0.22), Number of Hydrogen Bond Donors (5, 0.21), Bertz Complexity (5, 0.19), NO Group Count (5, 0.18) |
> | | Llama 3 | LogP (5, 0.21), Topological Polar Surface Area (5, 0.20), NO Group Count (5, 0.18), Number of Hydrogen Bond Donors (4, 0.15), EState VSA1 (3, 0.10), Bertz Complexity (2, 0.07) |
> | | Gemini 2.0 | LogP (5, 0.22), Number of Hydrogen Bond Donors (5, 0.20), QED (3, 0.12), Eccentricity (2, 0.09), NO Group Count (2, 0.09), Fraction of Csp3 Carbon Atoms (2, 0.08), TPSA (1, 0.04) |
> | ToxCast | GPT-4 | LogP (5, 0.21), Topological Polar Surface Area (5, 0.20), QED (5, 0.19), Number of Hydrogen Bond Donors (4, 0.14), Balaban J Index (3, 0.09), Number of Rotatable Bonds (3, 0.08) |
> | | Llama 3 | LogP (5, 0.22), Balaban J Index (4, 0.14), Labute Approximate Surface Area (4, 0.13), EState VSA1 (3, 0.11), Topological Polar Surface Area (2, 0.07), QED (2, 0.06), Rotatable Bonds (1, 0.03) |
> | | Gemini 2.0 | LogP (5, 0.21), Topological Polar Surface Area (4, 0.11), Balaban J Index (3, 0.08), Rotatable Bonds (3, 0.08), Labute Approximate Surface Area (2, 0.05), QED (2, 0.06), TPSA (1, 0.02) |
> | SIDER | GPT-4 | LogP (5, 0.20), Topological Polar Surface Area (5, 0.18), QED (4, 0.12), Number of Hydrogen Bond Donors (4, 0.12), Number of Rotatable Bonds (4, 0.10), Molecular Weight (3, 0.08) |
> | | Llama 3 | LogP (5, 0.19), Topological Polar Surface Area (4, 0.12), Aliphatic Rings Count (4, 0.10), Fraction of Csp3 Carbon Atoms (3, 0.08), Number of Hydrogen Bond Donors (2, 0.07), QED (2, 0.06) |
> | | Gemini 2.0 | Topological Polar Surface Area (5, 0.19), LogP (5, 0.18), Number of Hydrogen Bond Donors (4, 0.12), QED (3, 0.09), Molecular Weight (2, 0.06), Aliphatic Rings Count (1, 0.03) |
> | ESOL | GPT-4 | LogP (5, 0.19), Molecular Weight (5, 0.18), Topological Polar Surface Area (5, 0.17), Molecular Volume in Å³ (5, 0.17), QED (1, 0.05), Fraction of Csp3 Carbon Atoms (1, 0.04), Radius of Gyration (1, 0.03), Eccentricity (1, 0.03), Asphericity (1, 0.03) |
> | | Llama 3 | LogP (5, 0.17), Molecular Weight (5, 0.16), Topological Polar Surface Area (5, 0.15), Molecular Volume in Å³ (4, 0.10), Eccentricity (2, 0.05), Inertial Shape Factor (2, 0.04), QED (1, 0.03) |
> | | Gemini 2.0 | LogP (5, 0.16), Molecular Weight (5, 0.15), Topological Polar Surface Area (4, 0.10), Molecular Volume in Å³ (3, 0.08), Radius of Gyration (2, 0.05), Eccentricity (2, 0.05), Asphericity (2, 0.05), Inertial Shape Factor (1, 0.03), QED (1, 0.03) |

---

> ### Author Response · Authors · 2025-08-04
> **Response to Reviewer 8GVh - 3**
>
> | Dataset | LLM | LLM Selected Auxiliary Tasks |
> | ----- | ----- | ----- |
> | FreeSolv | GPT-4 | LogP (5, 0.19), QED (5, 0.18), Topological Polar Surface Area (4, 0.12), Molecular Volume in Å³ (4, 0.12), Molar Refractivity (4, 0.10), Radius of Gyration (3, 0.08) |
> | | Llama 3 | LogP (5, 0.18), QED (4, 0.12), Topological Polar Surface Area (4, 0.12), Molecular Volume in Å³ (3, 0.09), Molar Refractivity (3, 0.08), Eccentricity (2, 0.05) |
> | | Gemini 2.0 | LogP (5, 0.17), Molecular Volume in Å³ (4, 0.11), QED (3, 0.08), Topological Polar Surface Area (3, 0.08), Fraction of Csp3 Carbon Atoms (2, 0.06), Molar Refractivity (2, 0.06), Radius of Gyration (1, 0.03) |
> | Lipo | GPT-4 | LogP (5, 0.18), Molecular Weight (5, 0.17), Topological Polar Surface Area (4, 0.13), Number of Hydrogen Bond Donors (4, 0.13), Molar Refractivity (4, 0.12), Fraction of Csp3 Carbon (2, 0.06), Radius of Gyration (1, 0.03) |
> | | Llama 3 | LogP (5, 0.16), Molecular Weight (5, 0.16), Molar Refractivity (4, 0.11), Topological Polar Surface Area (3, 0.09), Number of Hydrogen Bond Donors (2, 0.06), Heavy Atom Molecular Weight (2, 0.05), Fraction of Csp3 Carbon Atoms (1, 0.03) |
> | | Gemini 2.0 | LogP (5, 0.15), Molecular Weight (5, 0.15), Molar Refractivity (4, 0.11), Topological Polar Surface Area (3, 0.08), Number of Hydrogen Bond Donors (2, 0.05), Fraction of Csp3 Carbon Atoms (2, 0.05), Eccentricity (1, 0.03) |
>
> This comprehensive reporting demonstrates that our findings are robust and not limited to selected examples. As shown, core physicochemical properties—such as LogP, Topological Polar Surface Area, and hydrogen bond donors—are consistently selected and assigned substantial weights across all runs and datasets, regardless of the LLM used. In contrast, less relevant descriptors (e.g., Eccentricity, Asphericity, Inertial Shape Factor, Radius of Gyration) are either rarely selected or, when chosen, appear in only a small fraction of runs with much lower average weights.
>
> These results provide direct statistical evidence that our auxiliary-task weighting mechanism is both stable and reproducible, and that the most useful auxiliary tasks are consistently prioritized by the framework. This robustness holds across datasets, LLMs, and random seeds, supporting the validity of our main findings.
>
> **Q6**
> In our experiments, we used the following LLM versions: GPT-4 (OpenAI, “gpt-4-2024-04-09”), Gemini 2.0 (Google Cloud, June 2024), and Llama-3 70B (Meta, released May 2024).
>
> Regarding small language models, we have already begun evaluating our method with Qwen3-8B, an open-source 8B-parameter model. (Access to Llama-3.1-8B is currently under application, and we will report results as soon as possible.) For full transparency, we also provide the compact code and a complete example interaction with Qwen3-8B.
>
> The table below summarizes the auxiliary tasks selected by Qwen3-8B across datasets, with (k, w) indicating the number of selections in 5 runs and the average learned weight. Performance is shown as ROC-AUC (classification) or RMSE (regression). These results indicate that *even small, recent models can reliably select relevant auxiliary tasks*, achieving performance comparable to much larger LLMs. We thank the reviewer for encouraging us to explore this direction, as it demonstrates our method’s broad applicability to real-world, resource-constrained scenarios
>
> | Dataset | LLM Selected Auxiliary Tasks | Performance (ROC-AUC for classification tasks) / RMSE for regression tasks |
> | ----- | ----- | ----- |
> | BBBP | Topological Polar Surface Area (5, 0.25), LogP (5, 0.24), Number of Hydrogen Bond Donors (5, 0.22), QED (5, 0.20), Molecular Weight (4, 0.14), Heavy Atom Molecular Weight (1, 0.04) | 0.812 |
> | BACE | LogP (5, 0.26), Topological Polar Surface Area (5, 0.25), Molar Refractivity (5, 0.22), Heavy Atom Molecular Weight (3, 0.11), EState VSA1 (2, 0.08), NO Group Count (2, 0.07), QED (1, 0.03) | 0.873 |
> | ClinTox | LogP (5, 0.23), Topological Polar Surface Area (4, 0.15), QED (3, 0.10), Bertz Complexity (3, 0.09), Aliphatic Rings Count (3, 0.08), Molecular Weight (1, 0.03), NHOH Group Count (1, 0.03) | 0.825 |
> | Tox21 | LogP (5, 0.22), Topological Polar Surface Area (5, 0.21), NO Group Count (5, 0.20), Number of Hydrogen Bond Donors (4, 0.13), EState VSA1 (3, 0.09), Bertz Complexity (2, 0.05) | 0.784 |
> | ToxCast | LogP (5, 0.18), Balaban J Index (4, 0.11), Labute Approximate Surface Area (4, 0.10), EState VSA1 (3, 0.09), Topological Polar Surface Area (2, 0.07), QED (2, 0.07), Rotatable Bonds (1, 0.03) | 0.703 |
> | SIDER | Topological Polar Surface Area (5, 0.17), LogP (5, 0.16), Number of Hydrogen Bond Donors (4, 0.11), QED (3, 0.09), Molecular Weight (2, 0.06), Aliphatic Rings Count (1, 0.03) | 0.672 |

---

> ### Author Response · Authors · 2025-08-04
> **Response to Reviewer 8GVh - 4**
>
> | Dataset | LLM Selected Auxiliary Tasks | Performance (ROC-AUC for classification tasks) / RMSE for regression tasks |
> | ----- | ----- | ----- |
> | ESOL | LogP (5, 0.20), Molecular Weight (5, 0.19), Topological Polar Surface Area (5, 0.17), Molecular Volume in Å³ (5, 0.16), QED (1, 0.03), Fraction of Csp3 Carbon Atoms (1, 0.03), Radius of Gyration (1, 0.03), Eccentricity (1, 0.03), Asphericity (1, 0.03) | 0.453 |
> | FreeSolv | LogP (5, 0.18), Molecular Volume in Å³ (4, 0.12), QED (3, 0.10), Topological Polar Surface Area (3, 0.09), Fraction of Csp3 Carbon Atoms (2, 0.06), Molar Refractivity (2, 0.06), Radius of Gyration (1, 0.03) | 1.217 |
> | Lipo | LogP (5, 0.17), Molecular Weight (5, 0.16), Molar Refractivity (4, 0.11), Topological Polar Surface Area (3, 0.09), Number of Hydrogen Bond Donors (2, 0.05), Fraction of Csp3 Carbon Atoms (2, 0.05), Eccentricity (1, 0.03) | 0.671 |
>
> ---
>
> [1] Tree of Thoughts: Deliberate Problem Solving with Large Language Models, NeurIPS, 2023.
> [2] Chain-of-Thought Prompting Elicits Reasoning in Large Language Models. NeurIPS 2022.
> [3] Toolformer: Language Models Can Teach Themselves to Use Tools, ICLR, 2023.
> [4] GPT-4 Technical Report, 2024.
> [5] Large Language Models as Optimizers, ICLR, 2023.
> [6] MoleculeNet: A Benchmark for Molecular Machine Learning, Chemical Science, 2018.
> [7] Best Practices for QSAR Model Development, Validation, and Exploitation, Molecular Informatics, 2010.
> [8] QSAR Modeling: Where Have You Been? Where Are You Going To?, Journal of Medicinal Chemistry, 2014.
>
> ---
>
> **Code:**
>
> from transformers import pipeline
>
> pipe = pipeline("text-generation", model="Qwen/Qwen3-8B")
> messages = [
>  {
>  "role": "system",
>  "content": "You are now an expert in computational chemistry and machine learning for molecular property prediction. \
>  Your task is to provide detailed and accurate information about computable molecular properties.",
>  },
>  {
>  "role": "user",
>  "content": "Search available information about these properties and write a brief summary for each property. \
>  Molecular Weight, Heavy Atom Molecular Weight, Number of Valence Electrons, Total Formal Charge, \
>  Topological Polar Surface Area, Labute Approximate Surface Area, Balaban J Index, Bertz Complexity, \
>  LogP, Molar Refractivity, EState VSA1, Number of Hydrogen Bond Donors, NHOH Group Count, NO Group Count, \
>  Number of Aromatic Rings, Aliphatic Rings Count, Heterocycles Count, Number of Rotatable Bonds, \
>  Fraction of Csp3 Carbon Atoms, Molecular Volume, Asphericity, Eccentricity, Radius of Gyration, \
>  Inertial Shape Factor, and QED.",
>  },
>  {
>  'role': 'assistant',
>  'content': "..."
>  },
>  {
>  "role": "user",
>  "content": "For the primary task - predict aqueous solubility. (Given a small organic molecule, predict its solubility in water, reported as the logarithm of molar concentration (log S, mol/L). Accurate prediction of solubility supports drug formulation and molecular design.), you assess the relevance between retrieved properties and the primary task with affinity scores, based on domain-specific knowledge and the summarised information. ",
>  },
>  {
>  'role': 'assistant',
>  'content': "..."
>  },
>  {
>  "role": "user",
>  "content": 'For the primary task - predict aqueous solubility (Given a small organic molecule, predict its solubility in water, reported as the logarithm of molar concentration (log S, mol/L). Accurate prediction of solubility supports drug formulation and molecular design.), you recommend which [$K$] properties as auxiliary tasks to improve the machine learning model performance on the primary task. You provide: a list of 5 selected auxiliary tasks and the affinity score of each auxiliary task to the primary task - predict aqueous solubility.',
>  },
> ]
>
> outputs = pipe(
>  messages,
>  max_new_tokens=2048,
>  do_sample=True,
>  temperature=0.2,
>  top_p=0.95,
>  repetition_penalty=1.1,
>  pad_token_id=pipe.tokenizer.eos_token_id,
> )
> print(outputs)

---

> ### Author Response · Authors · 2025-08-04
> **Response to Reviewer 8GVh - 5**
>
> **A complete conversation with Qwen3-8B:**
>
> {'generated_text': [{'role': 'system', 'content': 'You are now an expert in computational chemistry and machine learning for molecular property prediction. Your task is to provide detailed and accurate information about computable molecular properties.'},
> {'role': 'user', 'content': 'Search available information about these properties and write a brief summary for each property. Molecular Weight, Heavy Atom Molecular Weight, Number of Valence Electrons, Total Formal Charge, Topological Polar Surface Area, Labute Approximate Surface Area, Balaban J Index, Bertz Complexity, LogP, Molar Refractivity, EState VSA1, Number of Hydrogen Bond Donors, NHOH Group Count, NO Group Count, Number of Aromatic Rings, Aliphatic Rings Count, Heterocycles Count, Number of Rotatable Bonds, Fraction of Csp3 Carbon Atoms, Molecular Volume, Asphericity, Eccentricity, Radius of Gyration, Inertial Shape Factor, and QED.'},
> {'role': 'assistant', 'content': "<think>\nOkay, the user wants me to act as an expert in computational chemistry and machine learning for molecular property prediction. They need summaries of various molecular properties. Let me start by recalling what each of these properties means.\n\nFirst, Molecular Weight is straightforward—the sum of atomic weights. But wait, there's also Heavy Atom Molecular Weight, which excludes hydrogen. I should mention that difference. Then Valence Electrons: total valence electrons from all atoms. Formal Charge is calculated using the formula, right? Need to explain how it's determined.\n\nTopological Polar Surface Area (TPSA) and Labute Approximate Surface Area both relate to surface area but use different methods. TPSA uses atom types, while Labute might be more empirical. Balaban J Index is a topological descriptor based on graph theory. Bertz Complexity measures synthetic difficulty, so maybe mention factors like functional groups and ring systems.\n\nLogP is octanol-water partition coefficient, important for drug absorption. Molar Refractivity relates to polarizability. EState VSA1 is part of the E-State descriptors, focusing on hydrophobicity. Hydrogen Bond Donors and Acceptors count specific groups like -OH or NH. NHOH and NO groups are specific functional groups contributing to polarity.\n\nNumber of Aromatic Rings, Aliphatic Rings, and Heterocycles help in understanding ring structures. Rotatable bonds affect conformational flexibility. Fraction of Csp3 carbons indicates saturation level. Molecular Volume and shape parameters like Asphericity, Eccentricity, Radius of Gyration describe 3D structure. Inertial Shape Factor relates to density and solubility. QED is a measure of drug likeness, combining multiple properties.\n\nWait, did I miss any properties? Let me check the list again. Yes, all are covered. Now, need to ensure each summary is concise but informative. Also, note if some properties are similar or have overlapping purposes. For example, TPSA and Labute both estimate surface area but with different approaches. Highlight their applications, like in permeability or toxicity predictions. Make sure to mention relevance in ML models—like how LogP affects pharmacokinetics, or QED for virtual screening. Avoid jargon where possible, but since the user is an expert, technical terms are okay. Double-check definitions for accuracy, especially for less common ones like Balaban J Index and Bertz Complexity. Alright, time to structure each point clearly.\n</think>\n\nHere’s a detailed summary of the requested molecular properties, including their definitions, significance, and applications in computational chemistry and machine learning:\n\n---\n\n### **1. Molecular Weight** \n- **Definition**: Sum of the atomic weights of all atoms in a molecule. \n- **Significance**: Influences physical properties (e.g., boiling point, solubility), drug metabolism, and transport across biological membranes. \n- **Applications**: Used in QSAR models, molecular docking, and physicochemical profiling. \n\n---\n\n### **2. Heavy Atom Molecular Weight** \n- **Definition**: Molecular weight excluding hydrogen atoms. \n- **Significance**: Reflects the contribution of heavier atoms (C, O, N, etc.) to molecular interactions and stability. \n- **Applications**: Useful in predicting lipophilicity and molecular complexity. \n\n---\n\n### **3. Number of Valence Electrons** \n- **Definition**: Total valence electrons contributed by all atoms in the molecule. \n- **Significance**: Determines reactivity, bonding behavior, and electronic properties. \n- **Applications**: Critical for quantum mechanical calculations and predicting redox potentials. \n\n---\n\n### **4. Total Formal Charge** \n- **Definition**: Sum of formal charges on individual atoms, calculated as: \n $ \\text{Formal Charge} = \\text{Valence Electrons} - (\\text{Non-bonding Electrons} + \\frac{\\text{Bonding Electrons}}{2}) $. \n- **Significance**: Indicates charge distribution; deviations from neut

---

> ### Author Response · Authors · 2025-08-04
> **Response to Reviewer 8GVh - 6**
>
> rality may affect ionization states and reactivity. \n- **Applications**: Important for modeling charged species (e.g., ions, zwitterions). \n\n---\n\n### **5. Topological Polar Surface Area (TPSA)** \n- **Definition**: Estimates polar surface area using atom-type contributions (e.g., oxygen, nitrogen). \n- **Significance**: Predicts membrane permeability (e.g., PAMPA assay) and hydrogen bond capacity. \n- **Applications**: Used in drug design to optimize bioavailability and reduce toxicity. \n\n---\n\n### **6. Labute Approximate Surface Area** \n- **Definition**: Empirical method estimating molecular surface area based on atom connectivity. \n- **Significance**: Correlates with solubility and permeability. \n- **Applications**: Complementary to TPSA in assessing molecular size and shape effects. \n\n---\n\n### **7. Balaban J Index** \n- **Definition**: Topological descriptor derived from graph theory, quantifying branching and cyclicity. \n- **Significance**: Captures structural complexity and scaffold diversity. \n- **Applications**: Used in QSAR studies to correlate structure with activity or toxicity. \n\n---\n\n### **8. Bertz Complexity** \n- **Definition**: Measures synthetic accessibility based on functional group counts, ring systems, and stereochemistry. \n- **Significance**: Predicts ease of synthesis and likelihood of chemical stability. \n- **Applications**: Guides lead optimization in drug discovery. \n\n---\n\n### **9. LogP (Octanol-Water Partition Coefficient)** \n- **Definition**: Logarithm of the ratio of concentrations of a compound in octanol and water. \n- **Significance**: Reflects hydrophobicity/lipophilicity, critical for absorption, distribution, and toxicity. \n- **Applications**: Key parameter in ADMET (Absorption, Distribution, Metabolism, Excretion, Toxicity) predictions. \n\n---\n\n### **10. Molar Refractivity** \n- **Definition**: Measure of polarizability, calculated from molecular volume and refractive index. \n- **Significance**: Related to molecular size and intermolecular forces. \n- **Applications**: Used in QSAR models and predicting optical properties. \n\n---\n\n### **11. EState VSA1** \n- **Definition**: Part of the E-State descriptors, representing hydrophobicity (volume/shape). \n- **Significance**: Contributes to predicting solubility and membrane permeability. \n- **Applications**: Integrated into predictive models for drug-like molecules. \n\n---\n\n### **12. Number of Hydrogen Bond Donors** \n- **Definition**: Counts OH, NH, or SH groups capable of donating hydrogen bonds. \n- **Significance**: Influences solubility, protein binding, and bioavailability. \n- **Applications**: Critical for optimizing drug efficacy and reducing off-target interactions. \n\n---\n\n### **13. NHOH Group Count** \n- **Definition**: Quantifies the number of NH-OH groups (e.g., in amino alcohols). \n- **Significance**: Affects hydrogen bonding capacity and polarity. \n- **Applications**: Relevant for designing molecules with specific interaction profiles. \n\n---\n\n### **14. NO Group Count** \n- **Definition**: Counts nitro (-NO₂) or other nitrogen-oxygen groups. \n- **Significance**: Impacts oxidation state, reactivity, and toxicity. \n- **Applications**: Used in assessing metabolic stability and potential side effects. \n\n---\n\n### **15. Number of Aromatic Rings** \n- **Definition**: Counts benzene-like rings with conjugated π-electron systems. \n- **Significance**: Influences aromaticity, stability, and reactivity. \n- **Applications**: Critical for predicting UV absorption and electrophilic substitution reactions. \n\n---\n\n### **16. Aliphatic Rings Count** \n- **Definition**: Counts non-aromatic cyclic structures (e.g., cyclohexane). \n- **Significance**: Affects rigidity, conformational flexibility, and metabolic pathways. \n- **Applications**: Used in analyzing scaffold diversity and synthetic feasibility. \n\n---\n\n### **17. Heterocycles Count** \n- **Definition**: Counts rings containing at least one heteroatom (N, O, S). \n- **Significance**: Enhances molecular diversity and biological activity. \n- **Applications**: Key in drug design for targeting specific enzymes/proteins. \n\n---\n\n### **18. Number of Rotatable Bonds** \n- **Definition**: Counts single bonds between two heavy atoms not involved in ring systems. \n- **Significance**: Determines conformational flexibility and diffusion rates. \n- **Applications**: Predicts permeability and drug-likeness (e.g., Rule of Five). \n\n---\n\n### **19. Fraction of Csp³ Carbon Atoms** \n- **Definition**: Proportion of carbon atoms in sp³ hybridization (non-planar, saturated). \n- **Significance**: Reflects molecular saturation and steric bulk. \n- **Applications**: Used in assessing metabolic stability and lipophilicity. \n\n---\n\n### **20. Molecular Volume** \n- **Definition**: Estimated 3D volume of the molecule. \n- **Significance**: Influences packing efficiency, solubility, and molecular recognition. \n- **Applications**

---

> ### Author Response · Authors · 2025-08-04
> **Response to Reviewer 8GVh - 7**
>
> : Critical for predicting crystallization behavior and drug delivery. \n\n---\n\n### **21. Asphericity** \n- **Definition**: Measures deviation from spherical shape (based on 3D geometry). \n- **Significance**: Affects intermolecular interactions and solvation. \n- **Applications**: Used in studying molecular shape effects on biological activity. \n\n---\n\n### **22. Eccentricity** \n- **Definition**: Quantifies elongation of molecular shape (ratio of longest to shortest axis). \n- **Significance**: Influences diffusivity and membrane penetration. \n- **Applications**: Relevant for optimizing drug delivery and permeability. \n\n---\n\n### **2"},
> {'role': 'user', 'content': 'For the primary task - predict aqueous solubility. (Given a small organic molecule, predict its solubility in water, reported as the logarithm of molar concentration (log S, mol/L). Accurate prediction of solubility supports drug formulation and molecular design.), you assess the relevance between retrieved properties and the primary task with affinity scores, based on domain-specific knowledge and the summarised information. '},
> {'role': 'assistant', 'content': "<think>\nOkay, so I need to figure out which of these molecular properties are most relevant for predicting aqueous solubility. The user mentioned that they want to know the relevance of each property with affinity scores based on domain knowledge. Let me start by recalling what each property means and how it might relate to solubility.\n\nFirst, Molecular Weight. I remember that smaller molecules tend to be more soluble because they can interact better with water. But there's also a limit—very large molecules might be less soluble. So maybe Molecular Weight has a moderate relevance. But wait, sometimes larger molecules can have high solubility if they have lots of polar groups. Hmm, but generally, lower MW might be more favorable. Maybe assign a medium score here.\n\nHeavy Atom Molecular Weight. This excludes hydrogens. If the molecule has fewer heavy atoms, maybe it's simpler? Not sure. Wait, heavy atoms like oxygen and nitrogen contribute to polarity. So higher heavy atom weight could mean more polar groups, which might increase solubility. But again, depends on the context. Maybe this is related but not as directly as other factors. Maybe a low to medium score?\n\nNumber of Valence Electrons. Valence electrons relate to electron density and bonding. More valence electrons might mean more polarizable or different bonding types. But I'm not sure how directly this affects solubility. Maybe not very relevant. Low score.\n\nTotal Formal Charge. Charged molecules can be more soluble if they're ionic. For example, salts are highly soluble. However, neutral molecules might have different solubility behaviors. So formal charge could be important, especially if the molecule is ionized. Maybe medium to high relevance. High score?\n\nTopological Polar Surface Area (TPSA). Oh right! TPSA is a key factor in solubility. Higher TPSA usually means more hydrogen bonding capability, which increases solubility. This is a well-known descriptor in QSAR models. Definitely high relevance. High score.\n\nLabute Approximate Surface Area. Similar to TPSA, but empirical. It's used to estimate surface area, which relates to how much the molecule can interact with water. So this is also important. High score.\n\nBalaban J Index. This measures structural complexity. More complex molecules might have varied solubility depending on their structure. Not sure. Maybe some correlation, but not as direct as others. Medium score.\n\nBertz Complexity. Synthetic complexity might influence solubility indirectly, like through functional groups. But not sure. Maybe medium.\n\nLogP. LogP is inversely related to solubility. Higher logP means more hydrophobic, so lower solubility. So this is a strong negative predictor. High relevance. High score.\n\nMolar Refractivity. Relates to polarizability. More polarizable molecules might have different solubility characteristics. Not sure. Maybe medium.\n\nEState VSA1. This is part of the E-State descriptors, focusing on hydrophobicity. Since VSA1 is about volume and shape, it might relate to how the molecule fits in water. Maybe medium relevance.\n\nNumber of Hydrogen Bond Donors. More HBDs mean more ability to form hydrogen bonds with water, increasing solubility. High relevance. High score.\n\nNHOH Group Count. These groups can act as both donors and acceptors. So similar to HBDs, this would increase solubility. High score.\n\nNO Group Count. Nitro groups are electron-withdrawing and might make the molecule less soluble due to increased hydrophobicity or reduced hydrogen bonding. Lower relevance. Maybe medium or low.\n\nNumber of Aromatic Rings. Aromatics can be either hydrophilic or hydrophobic depending on substituents. Some aromatics are soluble, others aren't. Not sure. Maybe medium.\n\nAliphatic Rings Count. Aliphatic rings might be more rigid, affecting solubility. Not

---

> ### Author Response · Authors · 2025-08-04
> **Response to Reviewer 8GVh - 8**
>
> sure. Medium.\n\nHeterocycles Count. Heterocycles can be polar or not. Depends on the heteroatoms. Maybe medium.\n\nNumber of Rotatable Bonds. More rotatable bonds might indicate more flexible molecules, which could affect solubility. Not sure. Maybe medium.\n\nFraction of Csp³ Carbon Atoms. Sp³ carbons are more sterically hindered. Could affect solubility. Maybe medium.\n\nMolecular Volume. Larger molecules might be less soluble. So inverse relationship. High relevance. High score.\n\nAsphericity. More aspherical shapes might fit better in water. Maybe medium.\n\nEccentricity. Elongated shapes might have different solubility compared to spherical ones. Medium.\n\nRadius of Gyration. Measures overall size. Larger radius might mean lower solubility. High relevance. High score.\n\nInertial Shape Factor. Related to shape, which affects solubility. Medium.\n\nQED. Quantitative Estimate of Drug likeness. QED includes factors like solubility, so it's a composite measure. High relevance. High score.\n\nWait, let me check again. For example, LogP is definitely a big one. TPSA is another. Number of HBDs and HBA (like NHOH count) are important. Also, Molecular Volume and Radius of Gyration. Maybe Molecular Weight too. But I should prioritize those that are most directly linked. \n\nSo putting it all together, the top properties would be TPSA, LogP, Number of HBDs, Molecular Volume, Radius of Gyration, QED, and possibly Molecular Weight. Others like Heavy Atom Molecular Weight, EState VSA1, etc., might have lower scores. Need to assign scores based on how directly they impact solubility.\n</think>\n\nHere’s a structured assessment of the relevance of each molecular property to **aqueous solubility prediction**, ranked by **affinity scores** (1–5, where 5 = highest relevance):\n\n---\n\n### **1. Topological Polar Surface Area (TPSA)** \n- **Score**: 5 \n- **Relevance**: Directly correlates with hydrogen bonding capacity and polar interactions. Higher TPSA typically indicates greater solubility due to enhanced water-molecule interactions. Widely used in QSAR models for solubility prediction. \n\n---\n\n### **2. LogP (Octanol-Water Partition Coefficient)** \n- **Score**: 5 \n- **Relevance**: Strong inverse relationship with solubility. Lower LogP (more hydrophilic) implies higher aqueous solubility. Crucial for balancing hydrophobic/hydrophilic character. \n\n---\n\n### **3. Number of Hydrogen Bond Donors (HBDs)** \n- **Score**: 5 \n- **Relevance**: Increases solubility via hydrogen bonding with water. Molecules with ≥2 HBDs often exhibit improved solubility, though exceptions exist (e.g., bulky groups). \n\n---\n\n### **4. Molecular Volume** \n- **Score**: 4 \n- **Relevance**: Larger molecules face greater entropy penalties in solution, reducing solubility. Smaller molecules (lower molecular volume) are generally more soluble. \n\n---\n\n### **5. Radius of Gyration** \n- **Score**: 4 \n- **Relevance**: Measures molecular size and shape. Larger radii (bulky molecules) may hinder solubility due to steric hindrance and reduced packing efficiency. \n\n---\n\n### **6. QED (Quantitative Estimate of Drug-Likeness)** \n- **Score**: 4 \n- **Relevance**: Composite metric incorporating solubility-related parameters (e.g., TPSA, LogP, HBD/HBA). High QED values suggest better solubility and drug-like properties. \n\n---\n\n### **7. Number of Hydrogen Bond Acceptors (NHOH Group Count)** \n- **Score**: 4 \n- **Relevance**: Enhances solubility via hydrogen bonding. Groups like –OH, –NH, and –SH improve water interactions, though over-saturation can reduce solubility. \n\n---\n\n### **8. Molecular Weight** \n- **Score**: 3 \n- **Relevance**: Lighter molecules are generally more soluble, but exceptions exist (e.g., large polar molecules). Often used alongside TPSA and LogP for balanced predictions. \n\n---\n\n### **9. Heavy Atom Molecular Weight** \n- **Score**: 3 \n- **Relevance**: Reflects the contribution of polar atoms (C, O, N). Higher values may indicate increased polarity, but this is secondary to TPSA and LogP. \n\n---\n\n### **10. EState VSA1** \n- **Score**: 3 \n- **Relevance**: Estimates hydrophobicity (volume/shape). Lower VSA1 values correlate with higher solubility, but it is less precise than TPSA or LogP. \n\n---\n\n### **11. Number of Rotatable Bonds** \n- **Score**: 3 \n- **Relevance**: Flexible molecules may adopt conformations favoring solubility, but this effect is indirect and context-dependent. \n\n---\n\n### **12. Fraction of Csp³ Carbon Atoms** \n- **Score**: 3 \n- **Relevance**: Sp³-hybridized carbons introduce steric bulk, potentially reducing solubility. However, this is a minor contributor compared to TPSA or LogP. \n\n---\n\n### **13. Labute Approximate Surface Area** \n- **Score**: 3 \n- **Relevance**: Empirically estimates surface area, which influences solubility. Less precise than TPSA but complementary. \n\n---\n\n### **14. Number of Aromatic Rings** \n- **Score**: 2 \n- **Relevance**: Aromatics can en

---

> ### Author Response · Authors · 2025-08-04
> **Response to Reviewer 8GVh - 9**
>
> hance solubility (via π-π stacking) or reduce it (if hydrophobic). Context-dependent, with limited direct impact. \n\n---\n\n###"},
> {'role': 'user', 'content': 'For the primary task - predict aqueous solubility (Given a small organic molecule, predict its solubility in water, reported as the logarithm of molar concentration (log S, mol/L). Accurate prediction of solubility supports drug formulation and molecular design.), you recommend which [$K$] properties as auxiliary tasks to improve the machine learning model performance on the primary task. You provide: a list of 5 selected auxiliary tasks and the affinity score of each auxiliary task to the primary task - predict aqueous solubility.'},
> {'role': 'assistant', 'content': '<think>\nOkay, so the user wants me to recommend five auxiliary tasks that can help improve the ML model\'s performance for predicting aqueous solubility. Let me start by recalling what I know about solubility predictors.\n\nFrom the previous conversation, I listed several molecular properties and their relevance scores. The top ones were TPSA, LogP, HBD, Molecular Volume, and QED. But the user is asking for auxiliary tasks, which means they want properties that aren\'t directly the target but can help the model learn better.\n\nWait, the primary task is log S. So auxiliary tasks should be related but different enough to add value. Maybe properties that influence solubility indirectly? For example, something like pKa could be useful because pH affects solubility. Or maybe something like ionization state, but that might be part of formal charge?\n\nLooking back at the properties mentioned earlier, some like Number of Hydrogen Bond Acceptors (NHOH Group Count) and Molecular Volume have high relevance. Also, the user previously gave scores up to 5. But the question is about selecting auxiliary tasks, so perhaps those with lower scores but still relevant?\n\nWait, the user wants the auxiliary tasks to be separate from the primary task. Since the primary is log S, the auxiliary tasks should be other properties that contribute to understanding factors affecting solubility. \n\nLet me think again. The key factors for solubility include polarity (TPSA, LogP), hydrogen bonding (HBD, HBA), molecular size (Molecular Volume, Radius of Gyration), and maybe something like ionization (pKa, but that wasn\'t listed here). Wait, the original list didn\'t include pKa, but maybe there\'s another property. Alternatively, maybe the Number of Rotatable Bonds? Not sure.\n\nAlternatively, considering that the user has already given a list of properties, I need to pick five from them that are most helpful as auxiliary tasks. From the previous ranking, the top ones were TPSA (score 5), LogP (5), HBD (5), Molecular Volume (4), QED (4), then others. But since the primary task is log S, the auxiliary tasks should be properties that are correlated with solubility but not the same as the target. So maybe TPSA, LogP, HBD, Molecular Volume, and QED. But wait, QED is a composite metric that includes solubility factors. Using QED as an auxiliary might help capture multiple aspects.\n\nBut the user asked for auxiliary tasks, which are separate from the primary. So if the primary is log S, then the auxiliary tasks would be other properties that are related but not the same. So TPSA, LogP, HBD, Molecular Volume, and maybe something else like HBA (Number of Hydrogen Bond Acceptors) which was scored 4. Alternatively, maybe the Number of Rotatable Bonds (score 3) or something else.\n\nWait, looking back at the initial list, the user had properties like Number of Hydrogen Bond Donors (HBD), Number of Hydrogen Bond Acceptors (NHOH Group Count), Molecular Volume, QED, etc. So perhaps the best approach is to select the top-scoring properties that are not the primary target. Since the primary is log S, which isn\'t explicitly listed in the properties, but the properties themselves are features. Wait, no—the primary task is to predict log S, and the properties are the features. So the auxiliary tasks would be other properties that are related to solubility but not the same as log S. Therefore, the auxiliary tasks would be the properties that are highly correlated with solubility, such as TPSA, LogP, HBD, Molecular Volume, and QED. These can serve as additional features that help the model understand the underlying factors influencing solubility.\n\nSo the answer would list these five properties with their scores. But the user wants the auxiliary tasks to be selected from the provided list. Let me check the previous response again. The user had a list of properties, and the assistant ranked them by relevance. The top ones were TPSA (5), LogP (5), HBD (5), Molecular Volume (4), QED (4), then others. So selecting the top five would make sense. However, the user might prefer to avoid redundancy. For example, QED is a composite metric that includes TPSA, LogP, HBD, etc. Including QED as an auxiliary might add value by capturing a broad

---

> ### Author Response · Authors · 2025-08-04
> **Response to Reviewer 8GVh - 10**
>
> er picture. Similarly, Molecular Volume and Radius of Gyration both relate to size, but they\'re different measures. So perhaps the five would be TPSA, LogP, HBD, Molecular Volume, and QED. That gives a good mix of polarity, hydrogen bonding, size, and overall drug likeness.\n</think>\n\nHere’s a curated list of **5 auxiliary tasks** (properties) that are most relevant to improving the performance of a machine learning model for **predicting aqueous solubility (log S)**. These properties are chosen based on their strong correlation with solubility mechanisms and their ability to complement the primary task by providing diverse structural insights:\n\n---\n\n### **1. Topological Polar Surface Area (TPSA)** \n- **Affinity Score**: 5 \n- **Rationale**: Directly captures hydrogen bonding capacity and polar interactions, which are critical for water solubility. High TPSA correlates strongly with increased solubility due to enhanced water-molecule interactions. This property provides explicit structural insight into solubility drivers. \n\n---\n\n### **2. LogP (Octanol-Water Partition Coefficient)** \n- **Affinity Score**: 5 \n- **Rationale**: Inversely related to solubility—lower LogP (more hydrophilic) implies higher aqueous solubility. It balances hydrophobic/hydrophilic character, offering a complementary perspective to TPSA. Combining LogP with TPSA improves the model’s ability to distinguish between polar and non-polar solubility determinants. \n\n---\n\n### **3. Number of Hydrogen Bond Donors (HBDs)** \n- **Affinity Score**: 5 \n- **Rationale**: Hydrogen bonding significantly enhances solubility. Molecules with ≥2 HBDs often exhibit improved water solubility. This property highlights the role of functional groups (e.g., –OH, –NH) in forming favorable interactions with water, adding granularity to the model’s understanding of solubility mechanisms. \n\n---\n\n### **4. Molecular Weight** \n- **Affinity Score**: 4 \n- **Rationale**: Larger molecules face greater entropy penalties in solution, reducing solubility. Molecular weight complements TPSA and LogP by accounting for steric effects and packing efficiency, which are critical for predicting solubility in crowded environments (e.g., biological systems). \n\n---\n\n### **5. QED (Quantitative Estimate of Drug-Likeness)** \n- **Affinity Score**: 4 \n- **Rationale**: A composite metric integrating solubility-related parameters (e.g., TPSA, LogP, HBD/HBA). While not a direct measure of solubility, QED provides a holistic view of molecular properties that influence solubility, such as polarity, size, and hydrogen bonding. It acts as a proxy for "drug-like" behavior, enhancing generalization in solubility prediction. \n\n---\n\n### **Why These Are Effective Auxiliary Tasks** \nThese properties collectively address key solubility determinants: \n- **Hydrophilicity/polarity** (TPSA, LogP, HBD). \n- **Structural complexity** (Molecular Volume, QED). \n- **Functional group contributions** (HBD, HBA). \nBy incorporating these, the model gains a richer representation of molecular characteristics that drive solubility, leading to more robust and interpretable predictions.'}]}

---

> > ### Comment · Reviewer_8GVh · 2025-08-04
> >
> > Thank you to the authors for making every attempt to address my questions and concerns!
> > With all the clarifications provided, I intend to keep my positive evaluation of this work.

---

> > > ### Author Response · Authors · 2025-08-04
> > > **Response to Reviewer 8GVh - 11**
> > >
> > > We sincerely thank Reviewer 8GVh for the thoughtful feedback and follow-up questions. As mentioned previously, we just obtained access to `Llama-3.1-8B`. However, we encountered a technical limitation: this model currently does not support multi-turn conversational prompting, which is required by our framework’s methodology (as detailed in Appendix A.1–A.2). For reference, further information about the model’s chat capabilities can be found on its Hugging Face homepage.
> > >
> > > Given this constraint, we instead report the complete interaction using the closest available alternative, `Llama-3-8B-Instruct`. Upon review, we observed that `Llama-3-8B-Instruct` sometimes produces outputs not aligned with the provided candidate auxiliary task list (for example, generating "Number of Hydrogen Bond Acceptors," which does not exist in our predefined candidates). This behavior is an instance of hallucination, and we did not encounter similar issues with GPT-4, Gemini 2.0, Llama-3 70B, or Qwen3-8B. We believe this is a limitation of the current `Llama-3-8B-Instruct` model, and we will note this as a limitation of our methodology in practice.
> > >
> > > We will include a full discussion of these observations and their implications in Appendix G and H of the revised paper.
> > >
> > > ---
> > >
> > > **A complete conversation with Llama-3-8B-Instruct:**
> > >
> > > [{'generated_text': [{'role': 'system', 'content': 'You are now an expert in computational chemistry and machine learning for molecular property prediction.             Your task is to provide detailed and accurate information about computable molecular properties.'}, {'role': 'user', 'content': 'Search available information about these properties and write a brief summary for each property.             Molecular Weight, Heavy Atom Molecular Weight, Number of Valence Electrons, Total Formal Charge,                 Topological Polar Surface Area, Labute Approximate Surface Area, Balaban J Index, Bertz Complexity,                 LogP, Molar Refractivity, EState VSA1, Number of Hydrogen Bond Donors, NHOH Group Count, NO Group Count,                 Number of Aromatic Rings, Aliphatic Rings Count, Heterocycles Count, Number of Rotatable Bonds,                 Fraction of Csp3 Carbon Atoms, Molecular Volume, Asphericity, Eccentricity, Radius of Gyration,                 Inertial Shape Factor, and QED.'}, {'role': 'assistant', 'content': "Here's a brief summary for each of the computable molecular properties you listed:\n\n**Molecular Weight**: The total weight of all atoms in a molecule, typically measured in daltons (Da). This property is important for understanding pharmacokinetics and pharmacodynamics.\n\n**Heavy Atom Molecular Weight**: Similar to molecular weight, but only considers the weight of heavy atoms (non-hydrogen atoms) in the molecule. This property can be useful for predicting toxicity or metabolic stability.\n\n**Number of Valence Electrons**: The number of electrons that participate in chemical bonding within a molecule. This property can influence reactivity and chemical behavior.\n\n**Total Formal Charge**: The sum of the formal charges on all atoms in a molecule. This property can affect the overall charge distribution and reactivity of the molecule.\n\n**Topological Polar Surface Area (TPSA)**: A measure of the surface area of a molecule that is capable of forming hydrogen bonds. TPSA is an important predictor of drug-like properties and bioavailability.\n\n**Labute Approximate Surface Area (ASA)**: An estimate of the total surface area of a molecule, which can influence its solubility, absorption, and metabolism.\n\n**Balaban J Index**: A topological descriptor that measures the complexity of a molecule's structure. It can be used to predict physicochemical properties like logP and solubility.\n\n**Bertz Complexity**: Another topological descriptor that assesses the complexity of a molecule's structure. It can be used to predict properties like lipophilicity and bioavailability.\n\n**LogP**: The logarithm of the partition coefficient between octanol and water, which measures a molecule's hydrophobicity and ability to cross biological membranes.\n\n**Molar Refractivity**: A measure of a molecule's polarizability, which can influence its interactions with light and other molecules.\n\n**EState VSA1**: A set of descriptors based on the electrotopological state of a molecule, which can capture its electronic and steric properties.\n\n**Number of Hydrogen Bond Donors**: The number of functional groups in a molecule that can donate hydrogen bonds to other molecules.\n\n**NHOH Group Count**: The number of nitroso (-NOOH) groups in a molecule, which can influence its reactivity and toxicity.\n\n**NO Group Count**: The number of nitro (-NO2) groups in a molecule, which can affect its reactivity and toxicity.\n\n**Number of Aromatic Rings**: The number of planar, ring-shaped structures in a molecule, which can influence its reactivity and physical properties.\n\n**Aliphatic Rings Count**: The number

---

> > > > ### Author Response · Authors · 2025-08-04
> > > > **Response to Reviewer 8GVh - 12**
> > > >
> > > > of non-aromatic, chain-like structures in a molecule, which can affect its flexibility and reactivity.\n\n**Heterocycles Count**: The number of rings containing atoms other than carbon and hydrogen, which can influence a molecule's reactivity and biological activity.\n\n**Number of Rotatable Bonds**: The number of single bonds in a molecule that can rotate freely, which can affect its flexibility and conformational space.\n\n**Fraction of Csp3 Carbon Atoms**: The proportion of carbon atoms in a molecule that have three sigma bonds (i.e., not part of a double or triple bond), which can influence its physical properties.\n\n**Molecular Volume**: The volume occupied by a molecule, which can affect its solubility, diffusion, and interactions with other molecules.\n\n**Asphericity**: A measure of how much a molecule deviates from spherical symmetry, which can influence its interactions with other molecules and biological systems.\n\n**Eccentricity**: A measure of how elongated a molecule is, which can affect its interactions with other molecules and biological systems.\n\n**Radius of Gyration**: A measure of the distance from the center of mass of a molecule to its outermost atoms, which can influence its size and shape.\n\n**Inertial Shape Factor**: A measure of how much a molecule's shape influences its inertial properties, such as its moment of inertia.\n\n**QED (Quantitative Estimation of Drug-likeness)**: A score that predicts a molecule's likelihood of being a successful drug candidate based on its structural and physicochemical properties.\n\nThese computable molecular properties can be used to predict various aspects of a molecule's behavior, including its biological activity, pharmacokinetics, and toxicology. By combining multiple properties, researchers can gain a more comprehensive understanding of a molecule's potential and make informed decisions about its development as a therapeutic agent."}, {'role': 'user', 'content': 'For the primary task - predict aqueous solubility. (Given a small organic molecule, predict its solubility in water, reported as the logarithm of molar concentration (log S, mol/L). Accurate prediction of solubility supports drug formulation and molecular design.), you assess the relevance between retrieved properties and the primary task with affinity scores, based on domain-specific knowledge and the summarised information. '}, {'role': 'assistant', 'content': "To predict aqueous solubility (log S, mol/L) using the retrieved properties, I will assess their relevance to the primary task using affinity scores. Here's my assessment:\n\n**High Affinity (Score: 8-10)**\n\n* **Topological Polar Surface Area (TPSA)**: High TPSA values indicate increased solubility due to enhanced hydrogen bonding capabilities.\n* **Number of Hydrogen Bond Donors**: More hydrogen bond donors increase the likelihood of favorable interactions with water, leading to higher solubility.\n* **LogP**: Lower LogP values suggest increased solubility due to reduced hydrophobicity.\n* **Molecular Weight**: Smaller molecular weights often result in higher solubility due to reduced steric hindrance.\n\n**Medium Affinity (Score: 5-7)**\n\n* **Number of Rotatable Bonds**: Increased rotatable bonds can lead to improved solubility by reducing steric hindrance and increasing flexibility.\n* **Fraction of Csp3 Carbon Atoms**: Higher proportions of Csp3 carbon atoms may contribute to increased solubility through improved hydrogen bonding.\n* **Molar Refractivity**: Higher refractivity values can indicate increased solubility due to enhanced polarizability.\n\n**Low Affinity (Score: 0-4)**\n\n* **Number of Aromatic Rings**: Presence of aromatic rings may decrease solubility due to increased hydrophobicity.\n* **Aliphatic Rings Count**: Higher counts of aliphatic rings may also reduce solubility due to increased hydrophobicity.\n* **Heterocycles Count**: Presence of heterocycles may have varying effects on solubility depending on their specific structure and functionality.\n* **QED**: While QED is related to drug-likeness, it has limited direct correlation with aqueous solubility.\n\n**Neutral Affinity (Score: 1-4)**\n\n* **Balaban J Index**, **Bertz Complexity**, and **EState VSA1**: These topological descriptors have some indirect relationships with solubility, but their impact is less clear-cut compared to the high-affinity properties.\n\nBased on this assessment, I recommend focusing on the high-affinity properties (TPSA, Number of Hydrogen Bond Donors, LogP, and Molecular Weight) when building a model for predicting aqueous solubility. These properties are most strongly correlated with solubility and should provide a solid foundation for your predictive model."}, {'role': 'user', 'content': 'For the primary task - predict aqueous solubility (Given a small organic molecule, predict its solubility in water, reported as the logarithm of molar concentration (log S, mol/L). Accurate prediction of solubility supports drug fo

---

> > > > > ### Author Response · Authors · 2025-08-04
> > > > > **Response to Reviewer 8GVh - 13**
> > > > >
> > > > > rmulation and molecular design.), you recommend which [$K$] properties as auxiliary tasks to improve the machine learning model performance on the primary task. You provide: a list of 5 selected auxiliary tasks and the affinity score of each auxiliary task to the primary task - predict aqueous solubility.'}, {'role': 'assistant', 'content': "Based on the previous analysis, I recommend the following 5 auxiliary tasks to improve the machine learning model performance on the primary task of predicting aqueous solubility (log S, mol/L):\n\n**Auxiliary Task 1:** Predicting **Topological Polar Surface Area (TPSA)**\n\nAffinity Score: 9/10\n\nTPSA is highly relevant to aqueous solubility, as it measures the molecule's ability to form hydrogen bonds with water. Improving the accuracy of TPSA predictions can directly benefit the primary task.\n\n**Auxiliary Task 2:** Predicting **Number of Hydrogen Bond Acceptors**\n\nAffinity Score: 8.5/10\n\nThe number of hydrogen bond acceptors is another key factor influencing aqueous solubility. By accurately predicting this property, we can better understand the molecule's ability to interact with water.\n\n**Auxiliary Task 3:** Predicting **LogP**\n\nAffinity Score: 8/10\n\nLogP is a well-established property that correlates with aqueous solubility. Improving the accuracy of LogP predictions can help refine our understanding of the molecule's hydrophobicity and its impact on solubility.\n\n**Auxiliary Task 4:** Predicting **Molecular Weight**\n\nAffinity Score: 7.5/10\n\nMolecular weight is an important factor in determining aqueous solubility, particularly for larger molecules. By accurately predicting molecular weight, we can better understand the molecule's size and shape, which can inform its solubility.\n\n**Auxiliary Task 5:** Predicting **Fraction of Csp3 Carbon Atoms**\n\nAffinity Score: 6.5/10\n\nThe fraction of Csp3 carbon atoms is a subtle but important property that can influence aqueous solubility. By improving the accuracy of this prediction, we can better understand the molecule's electronic properties and their impact on solubility.\n\nBy incorporating these auxiliary tasks into our machine learning model, we can leverage the strong correlations between these properties and aqueous solubility to improve our predictions."}]}]

---

### Official Review · Reviewer_eSg9 · 2025-07-02

**Clarity:** 3
**Significance:** 3
**Originality:** 3
**Rating:** 4
**Confidence:** 3

**Summary:**

This paper aims to solve a critical problem in molecule properties prediction, where selecting effective auxiliary learning tasks are critical but challenging. To address this, this paper introduces Automatic Auxiliary Task Selection (AUTAUT), a fully automated framework that first retrieves auxiliary tasks using large language models and adaptively integrates them through a gradient alignment weighting mechanism. This paper evaluates AUTAUT on 9 molecular property prediction datasets, demonstrating its superiority over 10 auxiliary task-based methods and 18 state-of-the-art property prediction models.

**Questions:**

Some subsets in MoleculeNet are skipped, such as HIV and MUV datasets. What is the reason for this?

**Ethical Concerns:**

["NO or VERY MINOR ethics concerns only"]

**Final Justification:**

I maintain my positive score after author response. It addresses most of my concerns, with only concerns on more baselines and datasets unresolved, which cannot be finished due to the rebuttal time limits.

**Limitations:**

yes

**Quality:**

3

**Strengths And Weaknesses:**

**Strengths**

(1) This paper is well motivated by addressing an important problem in molecular property prediction on auxiliary learning tasks selection. The proposed workflow of LLM based searching and gradient based weighting mechanism are simple yet make sense to me. This paper shed light on a potential new workflow on machine learning and LLM based molecule property prediction.

(2) Results on MoleculeNet demonstrate the effectiveness of AUTAUT as it outperformed 18 state-of-the-art property prediction models.

**Weaknesses**

(1) My concerns for this paper are mainly on evaluation settings. First, AUTAUT can be regarded as a pretrained molecular model, as it is actually extracting information from GPT-4. This makes the comparison to lots of baselines in this paper unfair, especially those GNN-like methods that are trained from scratch. Some baselines, such as InstructMol, have leveraged large-scale molecular data. But there should be more baseline methods of its kind, such as “*Enhancing Activity Prediction Models in Drug Discovery with the Ability to Understand Human Language*” in ICML. A literature review on these large-scale pretrained molecular models is necessary.

(2) The evaluation of AUTAUT is solely based on the MoleculeNet benchmark, making the evaluation very limited. This is a famous yet relatively old benchmark, so it is possible that GPT-4 has information on even the test set of this benchmark. Although the LLM is only doing task selection in AUTAUT, it still could lead to potential information leakage. It will be more convincing to evaluate this workflow on more recent high quality molecular datasets.

---

> ### Author Rebuttal · Authors · 2025-07-30
>
> We sincerely thank Reviewer eSg9 for the thoughtful and insightful comments, which have significantly helped clarify our contributions and strengthen the evaluation rigor of our paper. Below we systematically address the concerns and questions raised by the reviewer and remain open to providing further clarifications if needed.
>
> > AUTAUT can be regarded as a pretrained molecular model, as it is actually extracting information from GPT-4.
>
> We fully agree that methods such as [1] are relevant. Indeed, our experiments already include a wide range of pretraining-based baselines, including PretrainGNN, GPT-GNN, GROVER, 3D-INFOMAX, GraphMVP, Uni-Mol, GEM, Pin-Tuning, LAC, and InstructMol (see Tables 2 and 3 in the manuscript). We will include further results with [1], though it may not be feasible to report these before the rebuttal deadline. In the revised manuscript, we will expand Section 2 to include a dedicated discussion of these large-scale pretrained molecular models and more clearly distinguish their training strategies from the retrieval-only, joint-training approach used in AUTAUT.
>
> We also briefly clarify that AUTAUT itself does **not** perform pretraining. As illustrated in Figure 1, the base ML predictive models (including GCN, GIN, and Graphformer) are always trained from scratch on the target dataset, just like most baselines in Table 2. The automatically selected auxiliary tasks serve as additional sources of supervision in a joint training process, not in a separate pretraining stage.
>
> > The evaluation of AUTAUT is solely based on the MoleculeNet benchmark,
>
> We selected MoleculeNet as our primary benchmark due to its broad adoption and comprehensive coverage of molecular property prediction tasks [1,2,3], which enables rigorous and standardized comparison with a wide range of prior work.
>
> > Although the LLM is only doing task selection in AUTAUT, it still could lead to potential information leakage.
>
> LLMs in AUTAUT are **never** exposed to any dataset description, data splits, or molecular features from the benchmarks. The LLMs are used exclusively to suggest and prioritize types of auxiliary tasks (e.g., “Molecular Weight”, “LogP”) based on the name of the primary prediction task. At no point are they exposed to, or queried with, dataset content, actual molecules, or their measured properties. The downstream predictive models are always trained and evaluated strictly following standard train/validation/test splits, ensuring that test molecules remain entirely unseen during both LLM prompting and model training.
>
> > Some subsets in MoleculeNet, like HIV and MUV, were skipped.
>
> Our dataset selection followed recent relevant work [3], focusing on the most commonly evaluated MoleculeNet benchmarks for fair and standardized comparison. Consequently, we adopted the same set of datasets as [3], which excludes HIV and MUV.
>
> Regarding these two datasets:
> - HIV: This dataset suffers from severe class imbalance and noisy labels, which complicate the application of auxiliary supervision and often result in unstable or non-representative outcomes. For these reasons, several recent works (e.g., MolGroup [2], InstructMol [3]) also omit HIV from their evaluations.
> - MUV: The MUV dataset is highly sparse, with very few positive samples per task. This makes it particularly challenging for both single- and multi-task learning frameworks, and recent literature on auxiliary task selection or molecular property prediction rarely reports results on MUV.
>
> Nevertheless, to address the reviewer’s concern, we have conducted supplementary experiments on both HIV and MUV. The results, summarized below, show that AUTAUT, even in these problematic datasets, achieves better results.
>
> | Model | Dataset | Performance (ROC-AUC) Baseline | Performance (ROC-AUC) +AUTAUT |
> | ---------- | -------- | --------- | ------------ |
> | GCN | HIV| 71.58 | 75.64 |
> | | HUV| 69.64 | 71.15 |
> | GIN | HIV| 78.23 | 78.43 |
> | | HUV| 71.03 | 71.61 |
> | Graphformer | HIV| 77.80 | 78.16 |
> | | HUV| 69.26 | 70.96 |
>
> We will include this table and additional analysis in Appendix G of the revised manuscript.
>
> [1] Enhancing Activity Prediction Models in Drug Discovery with the Ability to Understand Human Language, ICML, 2023.
> [2] Learning to Group Auxiliary Datasets for Molecule, NeurIPS, 2023.
> [3] Instructor-inspired Machine Learning for Robust Molecular Property Prediction, NeurIPS, 2024.

---

> > ### Author Response · Authors · 2025-08-04
> >
> > We sincerely thank you for highlighting the motivation and workflow of our method, as well as for your detailed comments regarding evaluation settings, potential information leakage, and benchmark coverage. Your suggestions on relevant baselines and concerns about the evaluation scope have been extremely helpful in clarifying our contributions and improving the rigor of our work.
> >
> > We believe that we have addressed all of your concerns regarding our submission in our previous response. As the discussion period is closing soon, we would highly appreciate any further feedback at this stage, as this would allow us to address any remaining issues in a timely manner.
> >
> > We look forward to hearing from you. Thank you again for your time and constructive comments.

---

> > ### Comment · Reviewer_eSg9 · 2025-08-05
> >
> > Thank you for the response, which addresses most of my concerns. I will maintain my original positive assessment of this paper.

---

### Official Review · Reviewer_mprS · 2025-07-03

**Clarity:** 2
**Significance:** 3
**Originality:** 2
**Rating:** 5
**Confidence:** 4

**Summary:**

This paper proposes an automatic framework for auxiliary task selection in molecular property prediction. The method aims to automatically identify the top k auxiliary tasks that are related to a given primary task in order to improve its performance, and then apply them during training to assist the learning of the primary task. An adaptive weighting scheme is used to assign different weights to the auxiliary tasks based on their gradient alignment with the primary task. The authors evaluate their method on several molecular property datasets and show performance gains over various baselines.

**Questions:**

See above.

**Ethical Concerns:**

["NO or VERY MINOR ethics concerns only"]

**Final Justification:**

This work points to a meaningful direction for leveraging and exploring LLMs in property prediction tasks. The experiments presented in both the paper and the rebuttal are extensive and could offer useful insights to the community. Although there are some discussions regarding the validity of the selected tasks, this remains open for further exploration, as automatic auxiliary task selection is a relatively new topic. Overall, I remain positive about the contributions.

**Limitations:**

See above.

**Quality:**

3

**Strengths And Weaknesses:**

The paper is generally well written and easy to follow. The method is clearly described, and the implementation details are sufficient to support reproducibility. The tables and results are presented clearly, and the comparisons across different model backbones make the findings convincing.

Weaknesses:
- It is not entirely clear how the authors validate that the selected auxiliary tasks are truly meaningful beyond their quantitative effect. For example, when presenting a generated summary of the auxiliary-primary task relationship, it is unclear how one can assess whether the generated summary is accurate or potentially hallucinated.

- The size and content of the auxiliary task pool could be better specified. It is important to understand how many auxiliary labels are available in each dataset and what their sources are.

- Since the method involves using LLMs and potentially expensive search over tasks, some discussion of computational cost would be valuable. How long does the search take? How much GPU time or memory is needed? How much does it cost?

- The results, while consistent, show only modest gains. Given the complexity of the approach and the use of large models, a deeper analysis of when and why auxiliary tasks provide significant benefit would help strengthen the impact.

- It would be helpful to include more qualitative or quantitative analysis of the selected auxiliary tasks. For example, do the selected tasks correspond to chemically meaningful groupings? Do they vary across different primary tasks?

---

> ### Author Rebuttal · Authors · 2025-07-30
>
> We sincerely thank Reviewer mprS for the thoughtful comments and valuable feedback on our submission. We appreciate the recognition of our method’s clarity, reproducibility, and comprehensive evaluation across models and datasets. Below, we address the main questions and remain open to providing further clarifications if needed.
>
> > It is not entirely clear how the authors validate that the selected auxiliary tasks are truly meaningful beyond their quantitative effect.
>
> To validate that the auxiliary tasks selected by AUTAUT are chemically meaningful, we conducted qualitative analyses based on domain knowledge and established chemical literature. As detailed in Section 4.4 (line 254-262), Appendix A.3 (line 530-551), and Table 5, repeatedly selected tasks such as LogP, Topological Polar Surface Area (TPSA), and Hydrogen Bond Donors align strongly with recognized chemical principles for predicting molecular properties.
>
> To further reduce the risk of hallucinated or inconsistent outputs, we set the LLM temperature to 0.2 (Section 4.4), which promotes stable and reproducible task selection. Across multiple runs and diverse primary tasks, the selected auxiliary tasks remained both stable and domain-relevant, supporting the robustness of our method.
>
> However, such manual assessments are not practical at scale, given the limited availability of domain expertise in many real-world applications. This limitation is a central motivation for our work - to provide an automatic and scalable solution for auxiliary task selection.
>
> > The size and content of the auxiliary task pool could be better specified.
>
> Our auxiliary task pool, referenced in Section 3.3, line 145 and described in Table 4, Appendix A, comprises 25 computable molecular descriptors, all retrieved from the internet and filtered using RDKit [1] for benchmarking and reproducibility. The descriptors cover a broad range of chemical property categories, including constitutional, topological, electronic, hydrogen bonding, ring descriptors, molecular flexibility, 3D descriptors, and drug-likeness. Importantly, none of these auxiliary labels are present in the original benchmark datasets (see Section 4.1).
>
> To further improve clarity, we will explicitly state the total number and source of auxiliary tasks in Section 3.3 and add more direct cross-references to Table 4 in the revised manuscript.
>
> > Some discussion of computational cost would be valuable.
>
> We agree that understanding computational costs is essential. The overall cost consists of two components: (1) LLM usage for auxiliary task selection and the subsequent (2) model training.
>
> **LLM usage:** The LLM-based retrieval and selection steps incurred approximately 21.3 seconds per GPT-4 request and 7.8 seconds per Gemini 2.0 request, resulting in a total expense of approximately \$32 across all five training runs (details in Appendix A.3). Search time is affected by the  API traffic situation. Llama 3 instead runs on our GPUs, so it incurs no additional cost beyond electricity.
>
> **Model training:** We provide an analysis of computational complexity in Appendix D. To summarize: during training, the adaptive weighting adds <5% computational overhead due to gradient alignment computation, as it is amortized across mini-batches. All experiments were conducted on 8 NVIDIA A100 GPUs. The GPU time and memory requirements vary across datasets and models.
>
> Below, we report the number of model parameters, GPU memory usage, and average execution time of ML models with and without AUTAUT across 5 runs. In practice, the additional computational burden from auxiliary task integration is minimal. We will add this table and further clarify these points in Appendix D of the next revision.
>
> | Model | Dataset | #Params | GPU Mem (MB) | Time (min) Baseline | Time (min) +AUTAUT |
> | ---------- | -------- | --------- | ------------ | ------------------- | ------------------ |
> | GCN | BACE| 296,833| 534 | 2.37 | 2.47 |
> | | BBBP | 296,833 | 536 | 3.14 | 3.23 |
> | | ClinTox | 296,962 | 536 | 2.30 | 2.37 |
> | | Tox21 | 298,252 | 532 | 12.17 | 12.64 |
> | | ToxCast | 376,297 | 538 | 18.75 | 19.44 |
> | | SIDER | 300,187 | 562 | 2.42 | 2.50 |
> | | ESOL | 296,833 | 508 | 1.77 | 1.84 |
> | | FreeSolv | 296,833 | 506 | 1.00 | 1.04 |
> | | Lipo | 296,833 | 536 | 6.44 | 6.68 |
> | GIN | BACE | 496,005 | 546 | 2.31 | 2.39 |
> | | BBBP | 496,005 | 548 | 3.08 | 3.19 |
> | | ClinTox | 496,134 | 548 | 2.28 | 2.36 |
> | | Tox21 | 497,424 | 542 | 12.00 | 12.47 |
> | | ToxCast | 575,469 | 546 | 18.69 | 19.34 |
> | | SIDER | 499,359 | 570 | 2.41 | 2.53 |
> | | ESOL | 496,005 | 518 | 1.76 | 1.83 |
> | | FreeSolv | 496,005 | 512 | 0.99 | 1.03 |
> | | Lipo | 496,005 | 550 | 6.38 | 6.64 |
> | Graphormer | BACE | 7,944,160 | 10,957 | 10.59 | 11.12 |
> | | BBBP | 7,944,160 | 10,997 | 14.48 | 15.07 |
> | | ClinTox | 7,946,740 | 10,997 | 10.45 | 10.88 |
> || Tox21 | 7,972,540 | 10,898 | 59.31 | 60.56 |
> || ToxCast | 9,533,440 | 10,998 | 92.52 | 94.28 |
> | | SIDER | 8,011,240 | 11,478 | 11.02 | 11.57 |
> | | ESOL | 7,944,160 | 10,417 | 7.81 | 7.96 |
> | | FreeSolv | 7,944,160 | 10,338 | 3.90 | 4.05 |
> | | Lipo | 7,944,160 | 11,018 | 31.02 | 32.50 |
>
> > a deeper analysis of when and why auxiliary tasks provide significant benefit would help strengthen the impact.
>
> We emphasize that these gains are consistent across all nine datasets, three backbone architectures, and multiple LLM variants (see Tables 2 and 3, and Figure 3c). Importantly, in regression tasks such as ESOL and FreeSolv, AUTAUT achieves RMSE reductions up to 47% in some settings.
>
> Our ablation studies (Figures 2 and 3a,b) indicate that both auxiliary task selection and adaptive weighting are essential for stable improvements. Gains are most pronounced in scenarios where the primary task is underdetermined (e.g., limited training data or noisy labels) and when auxiliary tasks exhibit strong gradient alignment with the main objective. In particular, regression tasks with sparse or complex labels benefit more from targeted auxiliary supervision. This is because the adaptive weighting mechanism emphasizes auxiliary tasks that provide non-redundant, mechanistically relevant signals and filters out noisy or less relevant gradients, resulting in faster convergence and lower training loss.
>
> We will incorporate this discussion in Section 4.4..
>
> > include more qualitative or quantitative analysis of the selected auxiliary tasks.
>
> Our analysis confirms that the selected auxiliary tasks consistently form chemically meaningful groupings tailored to each primary task. For example, polarity-related descriptors such as LogP and TPSA are repeatedly prioritized for solubility prediction, while toxicity and drug-likeness descriptors (e.g., Hydrogen Bond Donors, QED) are emphasized for toxicity-related endpoints. These patterns are highly stable across multiple runs and clearly reflect underlying chemical knowledge.
>
> As demonstrated in Section 4.4 and Appendix A (Table 5), the selected auxiliary tasks show strong domain alignment: for instance, descriptors related to molecular polarity (e.g., TPSA, Dipole Moment) are preferred for solubility tasks, while LogP and QED are frequently selected for toxicity prediction. The groupings also vary appropriately across different primary tasks, highlighting the adaptability of our approach.
>
> [1] RDKit: Open-source cheminformatics.

---

> > ### Author Response · Authors · 2025-08-04
> >
> > Thank you for your thoughtful and constructive review, particularly for highlighting the importance of validating the meaningfulness of selected auxiliary tasks, specifying the auxiliary task pool, clarifying computational costs, and encouraging deeper qualitative and quantitative analysis.
> >
> > We believe that we have addressed all of your concerns regarding our submission. As the discussion period is closing soon, we would highly appreciate any further feedback at this stage, as this would allow us to address any remaining issues in a timely manner.
> >
> > We look forward to hearing from you. Thank you again for your time and valuable comments.

---

> > > ### Comment · Reviewer_mprS · 2025-08-05
> > >
> > > Thank the authors for the rebuttal, which provides additional information that helps clarify the paper. This work points to a meaningful direction for leveraging and exploring LLMs in property prediction tasks. The experiments presented in both the paper and the rebuttal are extensive and could offer useful insights to the community. Although there are some discussions regarding the validity of the selected tasks, this remains open for further exploration, as automatic auxiliary task selection is a relatively new topic. Overall, I remain positive about the contributions.

---

### Note · Authors · 2025-08-12

We thank **Reviewers mprS, eSg9, and 8GVh**, whose comments helped strengthen our paper.
As a summary of our discussions with the reviewers:

* **mprS**: We clarified the auxiliary task design (25 descriptors), expanded on when and why auxiliary tasks help (especially in low-data/noisy-label settings), and added detailed cost/resource analysis.

* **eSg9**: We clarified that AUTAUT does not involve molecular pretraining—base models are always trained from scratch, included new pretraining-based baselines, and added results on HIV and MUV, omitted previously. We also addressed concerns about potential LLM leakage and clarified our prompting process.

* **8GVh**: We added results with smaller models (Qwen3-8B), clarified prompt and descriptor choices, demonstrated reproducibility, and provided correlation analysis between task selection frequency and adaptive weights across 5 runs.

**D2QM**: The review rates our work **borderline reject** despite listing **four clear strengths** and **only one weakness**, already addressed in the original submission (Section 4.4, Table 6) by directly comparing static feature concatenation with auxiliary-task training, where the latter clearly outperforms. We reinforced this with **new ablations**, **theoretical justification**, and **support from recent studies**, yet after an initial brief exchange, the reviewer did not re-engage.
This left the sole concern unresolved and unacknowledged, creating a mismatch between the review’s content and rating. We respectfully ask the AC and engaged reviewers to take this into account when forming their final recommendation.

Thank you again for your consideration.

— Authors of Submission 18116

---

### Decision · Program_Chairs · 2025-09-17

**Decision:**

Accept (poster)

**Comment:**

The paper introduces a novel task-selection framework to improve molecular property prediction. The pipeline queries an LLM to identify auxiliary tasks, which are then adaptively weighted during training.

The core experiments are conducted on the MoleculeNet benchmark using RDKit-derived features.

The main strength of the paper is that the framework delivers competitive performance compared to a wide range of other task-selection methods. This outcome is somewhat surprising and was noted positively by the reviewers.

However, as several reviewers observed, the experiments are conducted in a relatively contrived setting. The RDKit-based auxiliary tasks may not carry substantial information given the simplicity of many RDKit functions, and MoleculeNet has known issues, including overlap between training and test splits.

Furthermore, it is not clear whether the observed gains are primarily due to the LLM-based selection or the adaptive weighting.

These issues were discussed during the decision-making phase but were not deemed sufficient to warrant rejection.

Overall, it is my pleasure to recommend accepting the work. For the camera-ready version, please include an ablation that isolates the effect of the LLM from that of the optimization procedure.